# COVARIATE-GUIDED CLUSTERWISE LINEAR REGRESSION FOR GENERALIZATION TO UNSEEN DATA

**Dohyun Bu    Hyunho Kim    Jong-Seok Lee**[*]
Korea Advanced Institute of Science and Technology
{dohyun.bu, hyunho.kim, jongseok.lee}@kaist.ac.kr

## ABSTRACT

In many tabular regression tasks, the relationships between covariates and response can often be approximated as linear only within localized regions of the input space; a single global linear model therefore fails to capture these local relationships. Conventional Clusterwise Linear Regression (CLR) mitigates this issue by learning $K$ local regressors. However, existing algorithms either optimize latent binary indicators, (i) providing no explicit rule for assigning an *unseen* covariate vector to a cluster at test time, or rely on heuristic mixture of experts approaches, (ii) lacking convergence guarantees. To address these limitations, we propose *covariate-guided* CLR, an end-to-end framework that jointly learns an assignment function and $K$ linear regressors within a single gradient-based optimization loop. During training, a proxy network iteratively predicts coefficient vectors for inputs, and hard vector quantization assigns samples to their nearest codebook regressors. This alternating minimization procedure yields monotone descent of the empirical risk, converges under mild assumptions, and enjoys a PAC-style excess-risk bound. By treating the covariate data from all clusters as a single concatenated design matrix, we derive an $F$-test statistic from a nested linear model, quantitatively characterizing the effective model complexity. As $K$ varies, our method spans the spectrum from a single global linear model to instance-wise fits. Experimental results show that our method exactly reconstructs synthetic piecewise-linear surfaces, achieves accuracy comparable to strong black-box models on standard tabular benchmarks, and consistently outperforms existing CLR and mixture-of-experts approaches.

## 1 INTRODUCTION

Empirical studies suggest that, in many tabular data regression problems, the relationship between covariates $\boldsymbol{x}_i \in \mathbb{R}^p$ and the response $y_i \in \mathbb{R}$ can often be *approximately* linear only within localized regions of the input space; outside these regions, pronounced heterogeneity motivates modeling with several local linear components (Zhong et al., 2016; Devijver, 2017; Yu et al., 2017; Li & Liang, 2018; Klusowski et al., 2019; Diamandis et al., 2021; Kong et al., 2020). A single global linear model lacks the flexibility to represent local structures, whereas fully nonlinear black-box methods impose complexity that diminishes the applicability of conventional statistical diagnostics (Lipton, 2018; Petch et al., 2022; Chen & Zhang, 2023). Clusterwise Linear Regression (CLR) offers a middle ground: given $N$ samples $\{(\boldsymbol{x}_i, y_i)\}_{i=1}^N$, it learns $K$ linear regressors $\{\tilde{\mathbb{w}}_j \in \mathbb{R}^{p+1}\}_{j=1}^K$ and $NK$ binary indicators $[\alpha_{i,1}, \alpha_{i,2}, \ldots, \alpha_{i,K}]_{i=1}^N$ with $[\alpha_{i,1}, \alpha_{i,2}, \ldots, \alpha_{i,K}] \subset \{0,1\}^K$ and $\sum_{j=1}^K \alpha_{i,j} = 1$. With the augmented covariate vector $\mathbb{x}_i := [\boldsymbol{x}_i^\top, 1]^\top \in \mathbb{R}^{p+1}$, CLR minimizes

$$\frac{1}{N} \sum_{i=1}^N \big(y_i - \mathbb{x}_i^\top \big(\sum_{j=1}^K \alpha_{i,j} \tilde{\mathbb{w}}_j\big)\big)^2.$$

thereby retaining the simplicity of local linear models while accommodating heterogeneity through clustering. CLR has been applied successfully in market segmentation (Preda & Saporta, 2005), welding-process control (Ganjigatti et al., 2007), pavement management Khadka & Paz (2017), and rainfall forecasting (Bagirov et al., 2017). However, its classical form assumes a generative mixture of linear functions or relies on post-hoc clustering of covariates–response pair—assumptions misaligned

---

[*]Corresponding author.

with predictive tasks, where one must assign a new covariate vector to the appropriate local model without access to an underlying mixture distribution (Manwani & Sastry, 2015; Gitman et al., 2018).

Existing CLR-based and related methods face two fundamental limitations in the standard "single-point" prediction task, particularly under a *non-realizable* setting—where the data are not assumed to be probabilistically generated by exactly one of $K$ linear models. First, most algorithms decouple clustering and regression—via mixed programming based approach (Bertsimas & Shioda, 2007; Carbonneau et al., 2011), column generation based approach (Carbonneau et al., 2014), or algorithm based approach (Pal et al., 2022; Ghosh & Mazumdar, 2024)—yielding no explicit assignment rule for unseen covariates, which leads to overfitting on training dataset and degraded out-of-sample performance (Long et al., 2023). Second, Sparse Mixture-of-Experts (MoE) schemes, which integrate gating and regression (e.g., MoE-based (Ismail et al., 2023) or tree-based splits (Ahmed et al., 2018; Raymaekers et al., 2024)), suffer from unstable convergence, rely heavily on heuristics, and—when using axis-aligned partitions—cannot capture true assignment boundaries. As a result, the problem of learning and generalizing multiple local linear regressors for new covariates in the non-realizable setting remains largely unresolved.

To address these challenges, we introduce **CG-CLR** (Covariate-Guided Clusterwise Linear Regression), an end-to-end approach that unifies clustering and regression under a dual loss. CG-CLR maintains a *codebook* of $K$ linear regressors $\tilde{\mathbb{W}} := [\tilde{w}_j]_{j=1}^K$ and an $M$-hidden-layer *proxy network* $W_\phi$, parameterized by $\phi \in \mathbb{R}^d$, which predicts for each input $x_i$ the regressor that best fits it. Concretely, the proxy network outputs a proxy vector $\hat{w}_i := W_\phi(x_i)$, and each input $x_i$ is assigned to the regressor whose prediction $x_i^\top \tilde{w}_j$ is closest to $x_i^\top \hat{w}_i$. Training proceeds by alternating between (i) assigning each sample according to an explicit assignment rule and (ii) updating both $\tilde{\mathbb{W}}$ and $\phi$ to minimize the overall dual loss. This simple vector-quantized formulation yields a fully differentiable pipeline, admits a convergence proof for the empirical risk, and—by treating the concatenated design matrix of all $K$ regressors as a single linear model—allows us to compute the classical F-statistic to quantitatively characterize the effective model complexity.

Our main contributions are:

- **End-to-end assignment and regression:** CG-CLR is the first CLR framework that trains both the data-driven assignment rule and the $K$ local regressors jointly in a single gradient loop, thereby providing a principled way to assign *unseen* covariates at test time.

- **Convergence of alternating minimization:** By reformulating CLR as a dual loss $V_\lambda$, we show that our alternating updates yield a monotone descent of $V_\lambda$ and that CG-CLR exhibits linear convergence towards optimal parameters $(\tilde{\mathbb{W}}^\star, \phi^\star)$.

- **PAC-style generalization bounds:** In the sense of PAC learning for functional mixtures, we derive excess-risk bounds for single-point prediction under non-realizable settings, scaling as $O(\max_j \|\tilde{w}_j\| \sqrt{dM \log d \log 2N / N})$.

- **Model complexity quantification via F-test:** The joint design-matrix viewpoint enables embedding the classical F-statistic as a transparent indicator of the effective degrees of freedom when varying $K$.

- **Fine-grained control over model complexity:** Adjusting the cluster count $K$ lets CG-CLR range from a single global linear model ($K=1$) to nearly one regressor per instance $\left(K \approx N/(p+1)\right)$, giving users a smoothly adjustable dial between model simplicity and predictive flexibility.

## 2 RELATED WORKS

Methods that learn *local* linear models for tabular regression can be broadly grouped into three families: *clusterwise linear regression*, which assigns each sample to one of several linear models; *piecewise linear regression*, which constructs explicit partitions of the input space and fits a linear model in each partition; and *hypernetwork-driven local experts*, which employ a learned hypernetwork to either select a small set of linear experts or generate an input-specific linear model. We review key ideas and representative works in each family and then summarize their common limitations.

**Clusterwise linear regression** Conventional CLR fits the $NK$ binary assignment indicators $[\alpha_{i,1}, \ldots, \alpha_{i,K}]_{i=1}^N$ and the associated linear models $[\tilde{w}_j]_{j=1}^K$ by solving a mixed–integer program (Bertsimas & Shioda, 2007; Carbonneau et al., 2011; 2014; D'Urso et al., 2010; Joki et al., 2020;

Table 1: Comparison of representative approaches against the four desiderata: availability of *single-point* prediction rule, a rigorous *convergence analysis*, applicability to the *agnostic* setting, and ability to model *flexible (assigned) regions* (✓ = satisfied, ✗ = not satisfied).

| Method Class | Single-Point | Convergence Analysis | Agnostic | Flexible Regions |
|---|:---:|:---:|:---:|:---:|
| Conventional CLR | ✗ | ✓ | ✗ | High |
| MLR | ✗ | ✓ | ✓ | High |
| LDT / PILOT | ✓ | ✗ | ✓ | Low |
| DC | ✓ | ✓ | ✓ | Moderate |
| S-IME$_d$ | ✓ | ✗ | ✓ | High |
| **CG-CLR** | ✓ | ✓ | ✓ | High |

Klusowski et al., 2019). Most of these studies carry out their analysis in the *realizable* setting, where each sample is assumed to be generated by one of exactly $K$ linear models. Although this assumption facilitates coefficient-recovery analysis, it fails to account for more general *non-realizable* cases and convergence guarantees cannot be established if the generative-mixture assumption is even slightly violated. More recent works, such as MLR, adopts an *agnostic* (non-realizable) setting, optimizing a hard-assignment min-loss via alternating minimization or a soft-assignment min-loss via expectation maximization (Pal et al., 2022; Ghosh & Mazumdar, 2024). Such "min-loss" guarantees hold for *list decoding*: given a test pair $(\boldsymbol{x}_{i'}, y_{i'})$ with the response $y_{i'}$ in hand, one can retrospectively select the linear model that attains the smallest error $|y_{i'} - \mathbf{x}_{i'}^{\top} \tilde{\mathbf{w}}_j|^2$ among the learned list.

**Piecewise linear regression** One strategy is to build decision trees or rule lists that partition the covariate space and then fit a separate linear model in each leaf (e.g., LDT (Ahmed et al., 2018), PILOT (Raymaekers et al., 2024)). Another branch of research realizes piecewise linear functions with shallow neural networks whose activations are absolute or maxima (Kahlert & Chua, 1990; Wang & Sun, 2005), and deeper nested architectures extend this idea to cover all piecewise linear functions (Lin et al., 1994). More recently, difference-of-convex (DC) formulations describe the response as the gap between two max-linear envelopes and estimate the underlying hyperplanes through a single optimization step (Siahkamari et al., 2020).

**Hypernetwork-driven local experts** Sparse Mixture-of-Experts models pair a fixed set of linear experts with a learned *gating network* that assigns each input a vector of weights over those experts, thereby producing an input-specific convex combination or hard selection (e.g., S-IME$_d$ (Ismail et al., 2023)). A complementary direction dispenses with the expert pool entirely and employs a *hypernetwork* that maps the covariate vector directly to an instance-wise coefficient vector: attention-style TabNet (Arik & Pfister, 2021) variants focus on feature selection (Yang et al., 2022), while contextual lasso and other linear hypernetworks generate full local regressors (Thompson et al., 2023; Kadra et al., 2024).

**Limitations** Despite their diverse mechanics, the three families face limitations (Table 1). Conventional and agnostic CLR variants optimize only the fitting of $K$ linear models and require a *post-hoc* assignment rule to map unseen $\boldsymbol{x}_{i'}$ to a linear model (Manwani & Sastry, 2015; Gitman et al., 2018; Long et al., 2023; Vicari & Vichi, 2013), creating an assignment bias that can degrade generalization (Tao et al., 2022). Tree-based and DC piecewise linear methods hinge on axis-aligned or locally continuous partitions, which can restrict the geometric flexibility of the resulting regions. MoE frameworks train with soft gating but predict with hard selections, while instance-wise hypernetworks often exhibit highly varying coefficients and impose excessive complexity, rendering them nearly non-transparent as black-box models. Among the existing cluster-oriented methods, no single approach simultaneously achieves (i) a built-in single-point predictor, (ii) provable convergence in the agnostic regime, and (iii) assignment flexibility that is not restricted to locally continuous partitions.

## 3 METHOD

We formalize the task of learning *multiple* local linear regressors together with an *assignment* rule as a single optimization problem. This section presents the global problem formulation (Section 3.1), implementation details (Section 3.2), and theoretical guarantees (Section 3.3).

### 3.1 PROBLEM FORMULATION

**Setting** We are given $N$ i.i.d. samples $\{(\boldsymbol{x}_i, y_i)\}_{i=1}^N \subset \mathbb{R}^p \times \mathbb{R}$ drawn from an *unknown* distribution $D$. No generative mixture is assumed—a fully *agnostic* setting (Pal et al., 2022; Ghosh & Mazumdar, 2024). Our goal is to

(i) learn a *response-free routing rule* that maps any unseen covariate vector $\boldsymbol{x}_{i'}$ to an index $z_{i'}$ of the linear regressors; and

(ii) jointly fit the corresponding $K$ linear regressors $[\tilde{\mathrm{w}}_j]_{j=1}^K$ so that the test-time prediction $\tilde{y}_{i'} = \mathrm{x}_{i'}^\top \tilde{\mathrm{w}}_{z_{i'}}$ is accurate *without ever observing the true response $y_{i'}$*

This single-point test scenario precludes any *response-dependent* objective–such as the empirical min-loss in equation 1, which chooses a regressor *after* seeing the response $y_{i'}$. We therefore introduce and optimize the surrogate described below.

**Response-aware risk (infeasible)**  If the response $y_i$ were available *before* choosing one of the $K$ linear models, the oracle risk $L^\star$ would be

$$L^\star(\tilde{\mathbb{W}}) := \mathbb{E}_{(\boldsymbol{x}_i, y_i) \sim D} \left[ \min_{j \in \{1, \ldots, K\}} \left( y_i - \mathrm{x}_i^\top \tilde{\mathrm{w}}_j \right)^2 \right], \tag{1}$$

and minimizing its empirical analogue $\widehat{L}^\star$ is NP-hard even for moderate $K$ (Yi et al., 2014). More importantly, the inner minimization over $j$ is response-dependent: it selects the best regressor after observing $y_i$. This makes the objective fundamentally infeasible for CLR, whose test-time scenario requires selecting a regressor for a new covariate $\boldsymbol{x}_{i'}$ without access to $y_{i'}$. For this reason, a response-free surrogate is required.

**Proxy-based assignment**  CG-CLR eliminates these obstacles by first generating, for each covariate vector, a *proxy* $\hat{\mathrm{w}}_{i'} := W_{\boldsymbol{\phi}}(\boldsymbol{x}_{i'}) \in \mathbb{R}^{p+1}$ through the proxy network. The sample is then *hard-routed* to the linear regressor (codebook regressor) whose predicted value is closest to the proxy prediction:

$$z_{i'} := \underset{j \in [1, \ldots, K]}{\operatorname{argmin}} \left( \mathrm{x}_{i'}^\top \hat{\mathrm{w}}_{i'} - \mathrm{x}_{i'}^\top \tilde{\mathrm{w}}_j \right)^2. \tag{2}$$

Let $S_j := \{ i \mid z_i = j \}$ denote the set of indices of samples assigned to the $j$th regressor. This decision depends only on the covariate vector $\boldsymbol{x}_{i'}$ and therefore remains valid at test time when the response $y_{i'}$ is unavailable. Equation 2 can be viewed as a vector-quantization step that tightly couples the proxy space and the codebook, allowing the gradients of the downstream objective to jointly update both the proxy network parameters $\boldsymbol{\phi}$ and the linear regressors $\tilde{\mathbb{W}}$.

**Prediction-fit loss**  When the proxy network is updated, we want to *evaluate* the current codebook in a fixed state, yet still let the gradient flow *only* to the proxy parameters $\boldsymbol{\phi}$. Let $\hat{\mathrm{w}}_i^{\mathrm{stop}}$, $\tilde{\mathbb{W}}^{\mathrm{stop}}$, and $\tilde{\mathrm{w}}_{z_i}^{\mathrm{stop}}$ denote the stop-gradient copies of $\hat{\mathrm{w}}_i$, $\tilde{\mathbb{W}}$, and $\tilde{\mathrm{w}}_{z_i}$, respectively. We define

$$R^{\mathrm{fit}}(\boldsymbol{\phi}, \tilde{\mathbb{W}}^{\mathrm{stop}}) := \frac{1}{N} \sum_{i=1}^N \left( y_i - \mathrm{x}_i^\top \left( \hat{\mathrm{w}}_i - \underbrace{\hat{\mathrm{w}}_i^{\mathrm{stop}} + \tilde{\mathrm{w}}_{z_i}^{\mathrm{stop}}}_{\text{no gradient}} \right) \right)^2. \tag{3}$$

Because the term $\hat{\mathrm{w}}_i - \hat{\mathrm{w}}_i^{\mathrm{stop}}$ vanishes in the forward pass, equation 3 simply measures the squared error of the *frozen* prediction $\mathrm{x}_i^\top \tilde{\mathrm{w}}_{z_i}^{\mathrm{stop}}$. In the backward pass, however, $\nabla_{\tilde{\mathbb{W}}} R^{\mathrm{fit}} = 0$ while $\nabla_{\boldsymbol{\phi}} R^{\mathrm{fit}} \neq 0$; thus, the proxy network receives the usual regression gradient without ever increasing the current clusterwise loss.

**Alignment loss**  To encourage the proxy output vectors and their assigned codebook vectors to coincide, we add a symmetric alignment term

$$R^{\mathrm{align}}(\boldsymbol{\phi}, \tilde{\mathbb{W}}) := \frac{1}{N} \sum_{i=1}^N \left( \mathrm{x}_i^\top (\hat{\mathrm{w}}_i - \tilde{\mathrm{w}}_{z_i}) \right)^2, \tag{4}$$

whose gradient propagates to *both* the proxy network parameters $\boldsymbol{\phi}$ and the codebook $\tilde{\mathbb{W}}$, progressively shrinking the prediction gap between $\mathrm{x}_i^\top \hat{\mathrm{w}}_i$ and $\mathrm{x}_i^\top \tilde{\mathrm{w}}_{z_i}$ induced by the routing rule in equation 2.

**Composite objective**  The two losses play complementary roles: $R^{\mathrm{fit}}$ keeps each regressor faithful to its currently assigned samples, whereas $R^{\mathrm{align}}$ pushes the proxy predictions toward their codebook counterparts so that future assignments are improved. We blend them with a non-negative weight $\lambda \geq 0$:

$$V_\lambda(\boldsymbol{\phi}, \tilde{\mathbb{W}}) := R^{\mathrm{fit}}(\boldsymbol{\phi}, \tilde{\mathbb{W}}^{\mathrm{stop}}) + (1 + \lambda) R^{\mathrm{align}}(\boldsymbol{\phi}, \tilde{\mathbb{W}}). \tag{5}$$

The overall learning problem is

$$\min_{\boldsymbol{\phi}, \tilde{\mathbb{W}}} \ V_\lambda(\boldsymbol{\phi}, \tilde{\mathbb{W}}) \tag{6}$$

A larger $\lambda$ places more emphasis on closing the proxy–codebook gap, whereas $\lambda = 0$ assigns equal weight to fitting and alignment. In practice, we treat $\lambda$ as a hyperparameter that trades immediate prediction accuracy for faster long-term convergence of the joint model.

**Rationale for surrogate objective** When the alignment term vanishes ($R^{\mathrm{align}} \to 0$), every proxy prediction converges to its assigned codebook vector prediction ($\mathbb{x}_i^\top \hat{\mathbb{w}}_i \to \mathbb{x}_i^\top \tilde{\mathbb{w}}_{z_i}$). In that limit, the prediction–fit loss reduces to the *response-aware* min–loss in equation 1 evaluated on the current assignments, yet the routing rule itself remains response–free. Hence, minimizing equation 6 steers the system toward the oracle target while respecting the single–point test constraint. Because $V_\lambda$ is fully differentiable, we can optimize it using standard gradient methods. The stop-gradient copies in $R^{\mathrm{fit}}$ ensure a clean role separation: this loss *measures* each regressor's current error, whereas the alignment term updates *both* the proxy network and the codebook. Together, they yield a monotone descent of $V_\lambda$ (proved in Section 3.3) under the alternating updates described in Section 3.2.

## 3.2 IMPLEMENTATION DETAILS

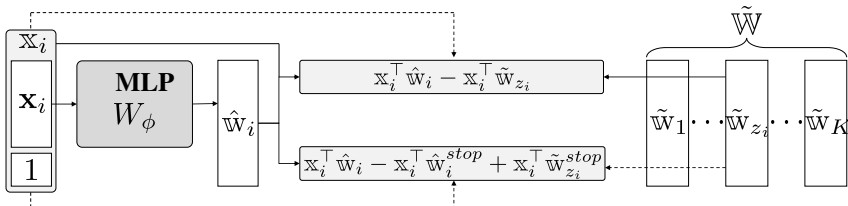

Figure 1: **CG-CLR architecture.** The proxy network $W_\phi$ maps a covariate $\boldsymbol{x}_i$ to a proxy output vector $\hat{\mathbb{w}}_i \in \mathbb{R}^{p+1}$. The *vector–quantizer* compares the proxy prediction $\hat{y}_i = \mathbb{x}_i^\top \hat{\mathbb{w}}_i$ with the $K$ codebook vector predictions $\{\mathbb{x}_i^\top \tilde{\mathbb{w}}_j\}_{j=1}^K$ and outputs the index $z_i$. The selected codebook vector $\tilde{\mathbb{w}}_{z_i}$ produces the final prediction $\tilde{y}_i = \mathbb{x}_i^\top \tilde{\mathbb{w}}_{z_i}$. Two losses are computed: the *prediction–fit* term $R^{\mathrm{fit}}$ equation 3 and the *alignment* term $R^{\mathrm{align}}$ equation 4. During back-propagation, the stop-gradient copies (dashed arrows) prevent the proxy path from negatively affecting the current clusterwise fit.

Figure 1 illustrates the forward and backward passes; the paragraphs below describe the two learnable components and detail the alternating minimization schedule used during training.

**Proxy network ($W_\phi$)** Unless stated otherwise, we instantiate $W_\phi$ as a depth-$M$ ReLU multilayer perceptron

$$\mathbb{R}^p \xrightarrow{\mathrm{FC}(p,h_1),\,\mathrm{ReLU}} \mathbb{R}_+^{h_1} \to \cdots \xrightarrow{\mathrm{FC}(h_{M-1},h_M),\,\mathrm{ReLU}} \mathbb{R}_+^{h_M} \xrightarrow{\mathrm{FC}(h_M,p+1)} \mathbb{R}^{p+1},$$

where $h_1, \ldots, h_M$ denote the hidden dimensions. The final layer is left *linear* so that each coordinate of the proxy vector $\hat{\mathbb{w}}_i = W_\phi(\boldsymbol{x}_i)$ can take any real value.

**Codebook of linear regressors ($\tilde{\mathbb{W}}$)** The $K$ regressors are stored as a matrix $\tilde{\mathbb{W}} = [\tilde{\mathbb{w}}_1, \ldots, \tilde{\mathbb{w}}_K] \in \mathbb{R}^{(p+1) \times K}$, with the last row representing the bias terms. We initialize $\tilde{\mathbb{W}}^{(0)}$ entry-wise, where each element is sampled independently from $\mathrm{Unif}(-1/K, 1/K)$, and standardize each raw feature so that the slope and bias coefficients operate on comparable scales throughout training.

**Training algorithm** Although the updates for $\phi$ and $\tilde{\mathbb{W}}$ are written separately for clarity, both are optimized jointly through the single composite loss $V_\lambda$; the stop-gradient mechanism only controls which part of the model receives gradient flow at each step, analogous to VQ-VAE (van den Oord et al., 2017) training. Optimization proceeds iteratively for epochs $t = 0, 1, \ldots, T-1$. Each epoch consists of the following two-step block–coordinate update (also refer to Figure 1):

- **Assignment step:** For every sample in the current mini-batch, we compute the proxy $\hat{\mathbb{w}}_i = W_{\phi^{(t)}}(\boldsymbol{x}_i)$, and assign

$$z_i \leftarrow \operatorname*{argmin}_{j \in [K]} \big(\mathbb{x}_i^\top (\hat{\mathbb{w}}_i - \tilde{\mathbb{w}}_j)\big)^2$$

  as in equation 2. The resulting mini-batch partitions $\{S_j\}_{j=1}^K$ are cached for the two parameter updates below.

- **Proxy network update:** With assignments fixed, we back-propagate the composite loss $V_\lambda(\phi, \tilde{\mathbb{W}}^{(t)})$ in equation 5. Stop-gradient copies ensure that the gradient flows *only* to $\phi$; the codebook remains frozen. We take one gradient descent step with learning rate $\eta$:

$$\phi^{(t+1)} = \phi^{(t)} - \eta \nabla_\phi V_\lambda(\phi^{(t)}, \tilde{\mathbb{W}}^{(t)}).$$

- **Codebook update:** Holding the updated proxy network fixed, we minimize the composite loss $V_\lambda$ with respect to $\tilde{\mathbb{W}}$. Since $R^{\text{fit}}$ carries a stop-gradient on $\tilde{\mathbb{W}}$, the update reduces to an *alignment* step:

$$\tilde{\mathbb{W}}^{(t+1)} = \tilde{\mathbb{W}}^{(t)} - \eta \nabla_{\tilde{\mathbb{W}}} R^{\text{align}}(\phi^{(t)}, \tilde{\mathbb{W}}^{(t)}).$$

Algorithm 1 is presented in full-batch form for clarity, although in practice we recommend performing mini-batch updates.

---

**Algorithm 1** CG-CLR: Alternating Minimization Algorithm

---

**Require:** Dataset $\{(\boldsymbol{x}_i, y_i)\}_{i=1}^N$, cluster number $K$, learning rate $\eta > 0$, weight $\lambda \geq 0$, number of epochs $T$.
1: **Initialize:** $\phi^{(0)}, \tilde{\mathbb{W}}^{(0)}$.
2: **for** $t = 0, 1, \ldots, T-1$ **do**
3:     **Assignment step:** Compute $z_i^{(t)}$ and construct partitions $S_j^{(t)}, j \in [K]$, according to equation 2.
4:     **Gradient updates:**

$$\phi^{(t+1)} \leftarrow \phi^{(t)} + \frac{2\eta(1+\lambda)}{N} \sum_{i=1}^N \Big( \frac{y_i + \lambda \mathbb{x}_i^\top \tilde{\mathbb{w}}_{z_i}^{(t)}}{1+\lambda} - \mathbb{x}_i^\top \hat{\mathbb{w}}_i^{(t)} \Big) (\nabla_\phi \hat{\mathbb{w}}_i^{(t)})^\top \mathbb{x}_i,$$

$$\tilde{\mathbb{w}}_j^{(t+1)} \leftarrow \tilde{\mathbb{w}}_j^{(t)} + \frac{2\eta(1+\lambda)}{N} \sum_{i \in S_j^{(t)}} \mathbb{x}_i \mathbb{x}_i^\top \big( \hat{\mathbb{w}}_i^{(t)} - \tilde{\mathbb{w}}_j^{(t)} \big), \quad \forall j \in [K].$$

5: **end for**
6: **return** $\phi^{(T)}, \tilde{\mathbb{W}}^{(T)}$.

---

**Test-time prediction.** Given a new covariate vector $\boldsymbol{x}_{i'}$, the proxy network produces an instance-specific coefficient estimate $W_\phi(\boldsymbol{x}_{i'})$. We assign the input to the closest codebook regressor via equation 2 and make the final prediction using the selected codebook coefficient, $\tilde{y}(\boldsymbol{x}_{i'}) = \mathbb{x}_{i'}^\top \tilde{\mathbb{w}}_{z_{i'}}$. As an alternative inference mode, one may also use the proxy coefficients directly, $\hat{y}(\boldsymbol{x}_{i'}) = \mathbb{x}_{i'}^\top W_\phi(\boldsymbol{x}_{i'})$.

### 3.3 THEORETICAL ANALYSIS

We now establish rigorous theoretical guarantees on the convergence and generalization of CG-CLR. We clearly state our assumptions, explicitly define necessary notation, and formally link these theoretical insights to the proposed optimization algorithm.

**Definitions and notation** To clearly state the theoretical guarantees, we define the following notation. The empirical risk on the training set is denoted by $\mathfrak{R}_{\text{train}}$, while the expected risk on an unseen test sample is $\mathfrak{R}_{\text{test}}$. The empirical quantity actually minimized during optimization is represented by $\hat{\mathfrak{R}}$. The family of predictors realized by CG-CLR forms the hypothesis class $\mathcal{H} = \{(\phi, \tilde{\mathbb{W}})\}$, parameterized by the proxy network weights $\phi$ and the codebook $\tilde{\mathbb{W}}$. Finally, we assume bounded support: there exist constants $Y_{\max}, \mathbb{X}_{\min}, \mathbb{X}_{\max} > 0$ such that $|y| \leq Y_{\max}$ and $\mathbb{X}_{\min} \leq \|\mathbb{x}\| \leq \mathbb{X}_{\max}$ almost surely for $(\boldsymbol{x}, y) \sim D$. More explicit definitions of the expected risk, empirical risk, and empirical quantity appear in Appendix C.1.

**Key assumptions** We collect here the regularity conditions used in our analysis. They play a technical role in guaranteeing that the Lyapunov argument applies and that the clusterwise regressions are well-posed: the first assumption concerns the smoothness of the proxy network, the second the curvature of the alignment loss in the codebook parameters, the third the separability of distinct linear rules, and the fourth a minimal expressivity requirement on the proxy network.

**Assumption 1** (Lipschitz continuity and Jacobian lower bound of proxy network). *For each fixed $\boldsymbol{x}$, the mapping $\phi \mapsto W_\phi(\boldsymbol{x})$ is $L_x$-Lipschitz continuous and its Jacobian satisfies $\|\nabla_\phi W_\phi(\boldsymbol{x}) v\| \geq m_x \|v\| > 0$ for all $v \in \mathbb{R}^d$.*

Intuitively, Assumption 1 rules out flat or infinitely steep directions in the proxy parameter space, so that gradient-based updates of $\phi$ change $W_\phi(\boldsymbol{x})$ in a controlled way.

**Assumption 2** (Strong convexity and smoothness). *For a fixed cluster assignment $z$, the alignment loss $R^{align}(\phi, \tilde{\mathbb{W}})$ is strongly convex and smooth with respect to $\tilde{\mathbb{W}}$, satisfying $\mu I \preceq \nabla^2_{\tilde{\mathbb{W}}} R^{align}(\phi, \tilde{\mathbb{W}}) \preceq L I$, for constants $\mu, L > 0$.*

Assumption 2 ensures that, once cluster assignments are fixed, the codebook update corresponds to a strongly convex quadratic problem, as in standard well-posed least-squares regression for each cluster.

**Assumption 3** (Cluster separation (Ghosh & Mazumdar (2024, 1.1 Setup))). *There exists a minimum gap $\Delta$ between predictions from distinct optimal regressors. Formally, for the optimal $\tilde{\mathbb{W}}^*$, $\min_{j \neq k} \min_i |\mathrm{x}_i^\top (\tilde{\mathrm{w}}_j^\star - \tilde{\mathrm{w}}_k^\star)| \geq \Delta > 0$.*

This separation condition guarantees that distinct optimal regressors remain distinguishable in terms of their predictions, so that the hard-assignment rule does not oscillate between clusters in regions where the underlying linear rules genuinely differ.

**Assumption 4** (Expressivity of the proxy network). *Let $\mathcal{F}_{\mathrm{proxy}} := \{ \boldsymbol{x} \mapsto W_\phi(\boldsymbol{x}) : \phi \in \Phi \}$ denote the function class realized by the proxy network, and let $\mathcal{F}_{\mathrm{codebook}} := \{ \boldsymbol{x} \mapsto \tilde{\mathrm{w}}_{z(\boldsymbol{x})} \}$ be the class of all mappings induced by $K$ fixed linear regressors, where $z(\boldsymbol{x})$ is the hard-assignment rule in equation 2. We require that the pseudo-dimension of $\mathcal{F}_{\mathrm{proxy}}$ satisfies $\mathrm{Pdim}(\mathcal{F}_{\mathrm{proxy}}) \geq C K(p+1)$ for some absolute constant $C > 0$.*

Hence the proxy network is at least as expressive, in a complexity sense, as the class of clusterwise linear mappings represented by the codebook, so it does not become a bottleneck in approximating those $K$ linear rules.

**Lyapunov function and descent guarantee** To rigorously establish the convergence properties of CG-CLR, we explicitly introduce the following Lyapunov function:

$$V_\lambda(\phi, \tilde{\mathbb{W}}) := R^{\mathrm{fit}}(\phi, \tilde{\mathbb{W}}^{\mathrm{stop}}) + (1 + \lambda) R^{\mathrm{align}}(\phi, \tilde{\mathbb{W}}).$$

A Lyapunov function is a scalar-valued function typically used in optimization and control theory to demonstrate convergence. Such a function must satisfy two crucial properties:

  (i) It is nonnegative and attains its minimum at the equilibrium (or optimal) point.

 (ii) It strictly decreases along the trajectories (or iterates) of the algorithm.

In our case, $V_\lambda$ clearly meets these criteria:

  (i) Both $R^{\mathrm{fit}}$ and $R^{\mathrm{align}}$ are nonnegative by definition, ensuring that $V_\lambda$ is nonnegative.

 (ii) Under appropriate step-size conditions, $V_\lambda$ strictly decreases at each iteration (explicitly shown in Proposition 3.1), thereby acting as a Lyapunov function guiding the convergence of CG-CLR.

**Proposition 3.1** (One-epoch descent). *Under Assumptions 1 and 2, suppose the cluster assignment $z^{(t)}$ remains fixed during epoch $t$. Define*

$$L_V := \max\left\{ (1+\lambda)L, \ (4+2\lambda)L_x^2 \mathbb{X}_{\max}^2 \right\}.$$

*For any step-size $0 < \eta \leq 1/L_V$, the simultaneous gradient updates guarantee a strict decrease of the Lyapunov function:*

$$V_\lambda(\phi^{(t+1)}, \tilde{\mathbb{W}}^{(t+1)}) \leq V_\lambda(\phi^{(t)}, \tilde{\mathbb{W}}^{(t)}) - \frac{\eta}{2} \left( \|\phi^{(t+1)} - \phi^{(t)}\|^2 + \|\tilde{\mathbb{W}}^{(t+1)} - \tilde{\mathbb{W}}^{(t)}\|^2 \right).$$

This explicitly verifies that $V_\lambda$ serves as a valid Lyapunov function, rigorously ensuring the convergence of the CG-CLR algorithm.

**Linear convergence** Under Assumptions 3 and 4, the cluster assignments eventually stabilize due to the minimum prediction-gap condition, resulting in fixed cluster indices (see Appendix A). Consequently, CG-CLR exhibits linear convergence toward the optimal parameters.

**Theorem 3.2** (Linear convergence). *Let Assumptions 1–4 hold. Define*

$$\mu_V := \min\left\{ 2m_x^2 \mathbb{X}_{\min}^2, \ (1+\lambda)\mu \right\}, \quad L_V := \max\left\{ (1+\lambda)L, \ (4+2\lambda)L_x^2 \mathbb{X}_{\max}^2 \right\}, \quad q := \frac{L_V - \mu_V}{L_V + \mu_V}.$$

*Gradient descent with the optimal step-size $\eta_\star = 2/(\mu_V + L_V)$ ensures that the stacked parameter vector $\theta = (\phi, \tilde{\mathbb{W}})$ converges linearly at the rate $q$, while the Lyapunov gap contracts at the rate $q^2$.*

**Theorem 3.3** (Generalization bound). *For any $\delta \in (0,1)$, with probability at least $1 - \delta$,*

$$\mathfrak{R}_{test} \leq \mathfrak{R}_{train} + 16c_0c_1 \max_{j \leq K} \|\tilde{\mathtt{w}}_j\| \sqrt{\frac{dM \log d \log 2N}{N}} + \frac{8c_1 \max_{j \leq K} \|\tilde{\mathtt{w}}_j\|}{\sqrt{N}} + Y_{\max}^2 \sqrt{\frac{\log(1/\delta)}{2N}},$$

*where $c_0$ is an absolute constant for standard ReLU MLPs and $c_1 := Y_{\max}\mathbb{X}_{\max}$.*

Detailed proofs for Theorems 3.2 and 3.3 are provided in Appendices B and C, respectively.

**Model complexity selection via F-test** Finally, we propose a theoretically grounded criterion for selecting model complexity based on nested-model F-tests.

**Proposition 3.4** (Sequential F-test). *The necessity of adding another cluster is tested sequentially using nested-model F-statistics:*

$$F_{K \to K+1} = \frac{(\mathrm{SSE}_K - \mathrm{SSE}_{K+1})/(p+1)}{\mathrm{SSE}_{K+1}/(N - (K+1)(p+1))} \sim F_{p+1, N-(K+1)(p+1)},$$

*and the number of clusters $K$ is selected based on statistical significance at a given significance level $\alpha$.*

A detailed derivation and proof of this criterion appear in Appendix D. The proposed test offers a statistically principled mechanism for precise control of model complexity.

## 4 EVALUATION

In this section, we empirically validate CG-CLR. We first consider a *synthetic dataset* whose underlying piecewise-linear function is exactly known. We also evaluate our method on *real-world datasets*, with detailed descriptions provided in Appendix F.1.

### 4.1 SYNTHETIC DATASET

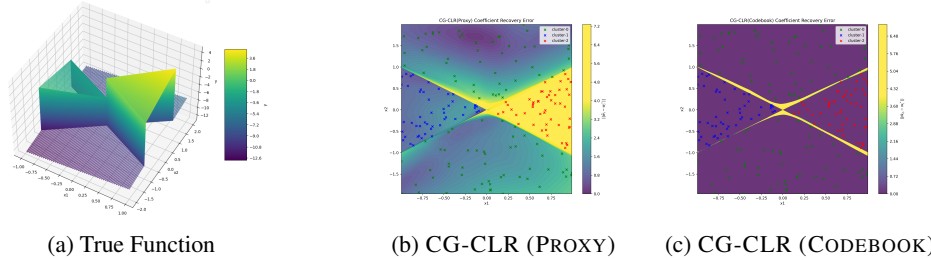

(a) True Function  (b) CG-CLR (PROXY)  (c) CG-CLR (CODEBOOK)

Figure 2: (2a) Ground-truth piecewise-linear function. (2b, 2c) Heat maps of coefficient recovery error norm on the evaluation grid, for (2b) proxy network and (2c) codebook regressors. Colors indicate error magnitude, from dark purple (zero error) to bright yellow (maximum error).

**Experimental setup** We sample four tilted wedges inside the rectangle $[-1, 1] \times [-2, 2]$ and generate responses from *exactly* one of three linear rules (also see Figure 2a.):

$$f^\star(x_1, x_2) = \begin{cases} x_1 + x_2 - 10, & x_2 > |x_1| \text{ or } x_2 < -|x_1|, \\ -x_1 - x_2 - 3, & x_1 < -|x_2|, \\ x_1 + 2x_2 + 2, & x_1 > |x_2|. \end{cases}$$

Thus, every sample $(\boldsymbol{x}_i, y_i)$ is *realizable*: the label is generated by a true piecewise-linear surface, with added i.i.d. Gaussian noise $\varepsilon_i \sim \mathcal{N}(0, 0.1^2)$. We draw 50 points per region ($N = 200$ in total) and standardize both the covariates and responses before training. As illustrated in Figure 2a, the true underlying model is composed of three clearly distinct linear regions, creating a challenging fitting scenario, especially since one region is completely separated by the other two. For evaluation, we create a $1000 \times 1000$ grid and predict the *noise-free* $f^\star$; therefore, all reported errors reflect model bias alone. In addition to prediction errors, we compute the distance between the recovered and ground-truth coefficient vectors on the same grid to assess how well each method reconstructs the underlying linear rules. We compare four models trained with $K = 3$, *except* LDT, whose tree must grow to depth 2 (effectively $K = 4$) to avoid early pruning: (i) LDT (Ahmed et al., 2018), (ii) S–IME$_d$ (Ismail et al., 2023), (iii) CG-CLR (PROXY), and (iv) CG-CLR (CODEBOOK).

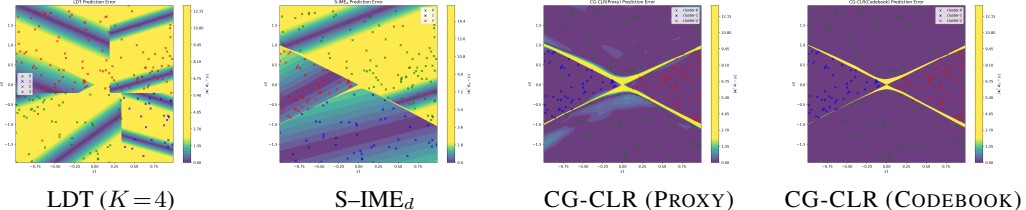

| LDT ($K=4$) | S–IME$_d$ | CG-CLR (PROXY) | CG-CLR (CODEBOOK) |

Figure 3: Absolute-error heat maps evaluated on a grid. Colors range from dark purple (zero error) to bright yellow (highest error).

Here, CG-CLR (PROXY) denotes the inference mode that predicts $\hat{y}(\boldsymbol{x}_i) = \mathrm{x}_i^\top W_\phi(\boldsymbol{x}_i)$ using the instance-wise coefficients output by the proxy network, whereas CG-CLR (CODEBOOK) uses the cluster-specific coefficients $\tilde{\mathrm{w}}_{z_i}$ selected by the hard-assignment rule in equation 2, producing the codebook prediction $\tilde{y}(\boldsymbol{x}_i) = \mathrm{x}_i^\top \tilde{\mathrm{w}}_{z_i}$. The heat maps in Figure 3 visualize the resulting *prediction* errors, while Figures 2b–2c compare the underlying regression coefficients themselves.

**Qualitative results**  Figure 3 visualizes the *prediction errors* on the evaluation grid, while the full prediction surfaces are shown in Figure 4 in the appendix. Only CG-CLR achieves (near-)zero error everywhere: the *codebook* predictions are exact, except for a thin artifact along the data-free cluster boundaries, and even the *proxy* predictions deviate only slightly due to fine-grained noise induced by the noisy responses. By contrast, LDT—constrained to axis-aligned splits—misses the oblique boundaries and produces large biased regions, while S-IME$_d$ incorrectly merges the northeast wedges, creating a broad region of high error.

**Coefficient recovery analysis**  Although the proxy and the codebook yield nearly identical *prediction* errors, their coefficient recovery differ significantly. The codebook, consisting of only three shared linear regressors, almost perfectly recovers the ground-truth coefficients; consequently, the error norm heat map is nearly blank (Figure 2c). In contrast, the proxy network generates distinct coefficient vectors at each point. While these proxy estimates are unbiased on average, they exhibit visible fluctuations, resulting in a speckled error norm map (Figure 2b). Therefore, the shared codebook provides a stable and accurate reconstruction of the underlying linear rules, whereas the instance-wise coefficients remain noisy—even when predictive accuracy is high.

**Selection of $K$ using the $F$-test**  Figure 5 in the appendix shows that models with $K = 1, 2$ underfit the data, whereas models with $K \geq 4$ over-segment the surface. A nested-model $F$-test on the training set quantitatively confirms this visual intuition: the step from $K = 2$ to $3$ is significant at the $\alpha = 0.01$ level ($p$-value $< 0.001$), but the increase from $K = 3$ to $4$ is not ($p$-value $= 0.038 > 0.01$). Thus, $K = 3$ is the smallest model that passes the test, aligning quantitatively with the ground truth.

## 4.2 REAL-WORLD DATASETS

**Experimental setup**  Table 4 in the appendix lists the seven tabular benchmarks for regression. Categorical columns are one-hot encoded, and all numeric features—including the response—are standardized. We perform *nested* 5-fold cross-validation: each outer fold is held out for testing, and the remaining data are split 3:1 into training and validation sets, resulting in 20 independent test estimates per dataset. All baselines are tuned via grid-search on the validation fold. For CG-CLR, we *fix* all hyperparameters across datasets—proxy network architecture, optimizer settings, and regularization parameter $\lambda = 1$ (Appendix F.3)—such that the only dataset-dependent parameter is the coverage budget $K = \lfloor N_{\mathrm{tr}}/(10p+10) \rfloor$, used for the "large-coverage" group, where $N_{\mathrm{tr}}$ is the size of the training set.

**Model groups**  We group baseline models by their *coverage*, defined as the average fraction of training samples explained by a single local regressor. **Small-coverage:** Methods that have either *no explicit partition* (RF (Breiman, 2001), XGBoost (Chen & Guestrin, 2016), CatBoost (Prokhorenkova et al., 2018), vanilla DNN) or an *instance-level partition* with $K > \lceil N_{\mathrm{tr}}/(p+1) \rceil$, represented by DC (Siahkamari et al., 2020) and the proxy prediction of CG-CLR (CG-CLR (PROXY), which uses $\mathrm{x}_i^\top \hat{\mathrm{w}}_i$). **Large-coverage:** Methods that maintain a small expert count $K \leq \lfloor N_{\mathrm{tr}}/(10p+10) \rfloor$: MLR with post-hoc nearest-neighbor cluster assignment (MLR*) (Pal et al., 2022), its EM-based counterpart (EM-MLR*) (Ghosh & Mazumdar, 2024), CART (Breiman et al., 2017), PILOT (Raymaekers et al., 2024), LDT (Ahmed et al., 2018), S-IME$_d$ (Ismail et al., 2023), and our CG-CLR (CODEBOOK). This grouping illustrates how distinguishes between methods that can regulate model complexity through clustering and those that cannot.

Table 2: Test RMSE (95% confidence intervals) across the seven datasets (lower is better). Within each model group, results are shown in **bold** if they are statistically indistinguishable from the group best. The overall best per dataset is **underlined**, with the same rule applied.

| Model | CONDUCT | HOUSING | BIKE | ELECTRICAL | PLANT | WINE | CONCRETE |
|---|---|---|---|---|---|---|---|
| RF | $[\mathbf{9.74}, \mathbf{9.89}]$ | $[0.507, 0.515]$ | $[55.89, 57.01]$ | $[0.012, 0.012]$ | $[3.420, 3.512]$ | $[\mathbf{0.615}, \mathbf{0.639}]$ | $[5.185, 5.577]$ |
| XGBoost | $[\mathbf{9.73}, \mathbf{9.85}]$ | $[0.453, 0.461]$ | $[45.51, 46.27]$ | $[0.008, 0.008]$ | $[\mathbf{3.156}, \mathbf{3.244}]$ | $[\mathbf{0.622}, \mathbf{0.646}]$ | $[\mathbf{4.348}, \mathbf{4.666}]$ |
| CatBoost | $[\mathbf{9.62}, \mathbf{9.76}]$ | $[\mathbf{0.440}, \mathbf{0.446}]$ | $[44.69, 45.29]$ | $[0.007, 0.007]$ | $[\mathbf{3.108}, \mathbf{3.216}]$ | $[\mathbf{0.621}, \mathbf{0.643}]$ | $[\mathbf{4.057}, \mathbf{4.393}]$ |
| DNN | $[10.35, 10.53]$ | $[0.502, 0.508]$ | $[41.76, 42.70]$ | $[0.007, 0.007]$ | $[3.738, 3.832]$ | $[0.659, 0.681]$ | $[4.760, 5.108]$ |
| DC | $[11.38, 11.94]$ | $[0.586, 0.608]$ | $[55.02, 55.70]$ | $[0.012, 0.012]$ | $[3.696, 3.796]$ | $[\mathbf{0.638}, \mathbf{0.664}]$ | $[5.454, 5.854]$ |
| CG-CLR (PROXY) | $[10.36, 10.50]$ | $[0.487, 0.498]$ | $[\underline{\mathbf{39.96}, \mathbf{40.86}}]$ | $[\underline{\mathbf{0.006}, \mathbf{0.006}}]$ | $[3.583, 3.680]$ | $[0.654, 0.679]$ | $[4.663, 5.096]$ |
| MLR* | $[33.77, 33.95]$ | $[1.159, 1.169]$ | $[177.57, 178.91]$ | $[0.035, 0.035]$ | $[13.717, 13.877]$ | $[0.827, 0.843]$ | $[15.037, 15.449]$ |
| EM-MLR* | $[18.75, 19.11]$ | $[0.748, 0.768]$ | $[102.28, 103.62]$ | $[0.022, 0.022]$ | $[5.208, 5.406]$ | $[0.726, 0.747]$ | $[10.404, 10.938]$ |
| CART | $[16.56, 16.86]$ | $[0.645, 0.659]$ | $[120.34, 122.30]$ | $[0.021, 0.021]$ | $[4.217, 4.305]$ | $[0.734, 0.756]$ | $[9.316, 9.822]$ |
| PILOT | $[15.31, 15.74]$ | $[0.816, 0.826]$ | $[165.26, 166.48]$ | $[0.033, 0.033]$ | $[4.711, 4.837]$ | $[0.746, 0.766]$ | $[14.236, 14.824]$ |
| LDT | $[13.36, 15.62]$ | $[0.603, 0.743]$ | $[59.44, 60.02]$ | $[0.015, 0.015]$ | $[4.076, 4.162]$ | $[0.698, 0.718]$ | $[6.110, 6.418]$ |
| S-IME$_d$ | $[12.68, 13.04]$ | $[0.560, 0.570]$ | $[56.13, 58.01]$ | $[0.010, 0.010]$ | $[4.183, 4.259]$ | $[0.693, 0.715]$ | $[8.853, 9.511]$ |
| CG-CLR (CODEBOOK) | $[\mathbf{10.50}, \mathbf{10.62}]$ | $[\mathbf{0.485}, \mathbf{0.497}]$ | $[\mathbf{40.77}, \mathbf{41.71}]$ | $[\mathbf{0.006}, \mathbf{0.006}]$ | $[\mathbf{3.573}, \mathbf{3.675}]$ | $[\mathbf{0.652}, \mathbf{0.676}]$ | $[\mathbf{5.193}, \mathbf{5.537}]$ |

**Results**   Given the *same* coverage budget $K = \lfloor N_{\mathrm{tr}}/(10p+10) \rfloor$, CG-CLR achieves the best RMSE among all large-coverage methods and even attains the overall best performance on BIKE and ELECTRICAL. On the remaining five datasets, its RMSE is only marginally higher than the best gradient-boosted ensemble, despite using just $K$ shared regressors instead of thousands of trees. MLR* illustrates the danger of post-hoc assignment: while its local experts adequately fit the training data, the absence of a learned assignment rule severely hurts generalization, resulting in the worst scores within its group. CART, LDT, and PILOT are constrained by axis-aligned or single-split partitions, and S-IME$_d$ exhibits optimization instability. In contrast, the vector-quantized alignment of CG-CLR provides both a stable global objective and a built-in single-point predictor, matching or surpassing black-box accuracy while maintaining a compact codebook that enables principled control over model complexity. Additional evaluations—including sensitivity analyses for $\lambda$ and $K$, computational cost comparisons, an empirical check of Assumption 1-3, and a small-$K$ case study illustrating interpretability via local linear models—are provided in Appendix F.

## 5   CONCLUSION

We introduced **CG-CLR**, an end-to-end framework that jointly learns cluster assignment and local linear regressors via a vector-quantized dual loss. The resulting algorithm (i) provides a single-point prediction rule for unseen covariates through a covariate-guided pipeline, (ii) enjoys monotone empirical-risk descent with formal convergence guarantees, (iii) admits PAC-style excess-risk bounds, and (iv) includes an $F$-test criterion to quantify effective model complexity. Empirically, CG-CLR accurately reconstructs piecewise-linear functions on challenging synthetic data and achieves performance comparable to strong black-box methods on real-world tabular benchmarks, while maintaining a compact codebook that enables principled control over model complexity and, for small $K$, interpretability through local linear models. It also outperforms existing approaches designed for the same purpose.

For future work, relaxing the prediction-gap assumption (Assumption 3) by introducing input-space separation criteria or adaptive margins is a promising direction to further broaden the applicability of the proposed convergence analysis. Another compelling extension is to incorporate cluster-specific feature selection or sparsity constraints, enabling more stable and interpretable local linear models in high-dimensional settings. Finally, extending CG-CLR to shallow nonlinear experts—such as kernelized or spline-based regressors—could capture richer local structures while retaining explicit assignment and controlled model complexity.

## 6   REPRODUCTIBILITY STATEMENT

The proofs of the main results are given in Appendix A–D. Details on figure generation are provided in Section 4 and Appendix E. Appendix F describes the benchmarks used in our experiments, the corresponding experimental settings, and the implementation details of the baseline methods for comparison. In addition, the supplementary material includes the benchmark dataset and fully reproducible code.

## 7 ACKNOWLEDGEMENTS

This research was supported in part by the National Research Foundation of Korea grant funded by the Korea government (MSIT) (Grant number RS-2020-NR049544, RS-2022-NR068758, and RS-2024-00361377) and in part by the Korea Evaluation Institute of Industrial Technology grant funded by the Korea government (MOTIR) (Grant number RS-2025-25458052).

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

# A    CLUSTER ASSIGNMENT STABILIZATION

## A.1    ABSTRACT ERROR CONTRACTION UNDER IDEALIZED ALTERNATING MINIMIZATION

This section presents a high-level analysis of how the alternating minimization procedure drives the stabilization of cluster assignments. Here, we abstract away from the details of gradient-based optimization, and instead assume that each update within a high-level epoch brings the proxy predictions $\hat{y}$ and codebook predictions $\tilde{y}$ to the minimizers of their respective loss functions, given the current assignments. This idealized analysis serves to illustrate the mechanism by which the alternating minimization scheme contracts prediction errors and ultimately leads to stable assignments.

We emphasize that this argument is schematic: it isolates the error contraction and stabilization logic of the alternating minimization framework, while rigorous convergence proofs under gradient-based optimization are provided separately in Appendix B.

For a given sample $i \in [1, \ldots, N]$ and high-level epoch $t' \in \{t_0, t_1, \ldots\}$, we let $z_i^{t'}$ denote the cluster assignment and define the corresponding codebook and proxy predictions as

$$\tilde{y}_i^{(t')} := \mathrm{x}_i^\top \, \tilde{\mathrm{w}}_{z_i^{t'}}^{(t')}, \quad \hat{y}_i^{(t')} := \mathrm{x}_i^\top \, \hat{\mathrm{w}}_i^{(t')}.$$

The dual loss minimized by the alternating updates takes the following two-term form:

$$V_\lambda(\phi, \tilde{\mathrm{W}}) = \frac{1}{N} \sum_{i=1}^N \left( y_i - \hat{y}_i + y_i^{\mathrm{stop}} - \tilde{y}_i^{\mathrm{stop}} \right)^2 + (1+\lambda) \frac{1}{N} \sum_{i=1}^N (\tilde{y}_i - \hat{y}_i)^2.$$

Each iteration consists of the following updates:

$$\tilde{y}_i^{(t_{n+1})} = \mathrm{x}_i \tilde{\mathrm{w}}_{z_i}^{(t_{n+1})} = \mathrm{x}_i^\top \left( \underset{\tilde{\mathrm{w}}_{z_i}}{\mathrm{argmin}} \sum_{i \in S_{z_i}^{(t_n)}} (\tilde{y}_i - \hat{y}_i^{(t_n)})^2 \right), \qquad \hat{y}_i^{(t_{n+1})} = \underset{\hat{y}}{\mathrm{argmin}} \left( \frac{y_i + \lambda \tilde{y}_i^{(t_n)}}{1+\lambda} - \hat{y} \right)^2.$$

This yields the explicit updates:

$$\tilde{y}_i^{(t_{n+1})} = \mathrm{x}_i^\top \left( X_{z_i^{t_n}}^{(t_n)\top} X_{z_i^{t_n}}^{(t_n)} \right)^{-1} X_{z_i^{t_n}}^{(t_n)\top} \hat{\mathbf{y}}_{z_i^{t_n}}^{(t_n)}, \qquad \hat{y}_i^{(t_{n+1})} = \frac{y_i + \lambda \tilde{y}_i^{(t_n)}}{1+\lambda},$$

where $X_j^{(t')} := [\mathrm{x}_i]_{i \in S_j^{(t')}}^\top \in \mathbb{R}^{|S_j^{(t')}| \times (p+1)}$ denotes the design matrix for cluster $j$ at epoch $t'$ and $\hat{\mathbf{y}}_j^{(t')} := [\hat{y}_i]_{i \in S_j^{(t')}}^\top$ is the vector of proxy predictions assigned to cluster $j$.

Each high-level epoch in this analysis refers to a single alternating cycle in which the proxy and codebook regressors are assumed to be fully optimized with respect to their losses for the current assignments. This abstraction removes the effect of partial or stochastic optimization, allowing us to focus on the error contraction logic of the idealized alternating minimization scheme.

To characterize the contraction per iteration, we establish the following bound:

**Lemma A.1** (Projector gain $\kappa$). *Under Assumption 2 (strong convexity $\mu$ and smoothness $L$ of $R^{\mathrm{align}}$) holds, for every epoch $t'$, cluster $j$, and sample $i \in S_j^{(t')}$,*

$$\kappa_{i,j}^{(t')} := \left\| \mathrm{x}_i^\top (X_j^{(t')\top} X_j^{(t')})^{-1} X_j^{(t')\top} \right\|_2$$

*satisfies $\kappa_{i,j}^{(t)} \leq \sqrt{L/\mu}$.*

*Proof.*
Applying the Rayleigh quotient to $\mathrm{x}_i^\top (X_j^\top X_j)^{-1} \mathrm{x}_i$ and the eigenvalue bounds $\mu N I \preceq X_j^\top X_j \preceq LNI$ from Assumption 2 gives result.  □

From Lemma A.1, the error for the codebook prediction at each step is bounded by

$$|\tilde{y}_i^{(t_{n+1})} - y_i| \leq \kappa \max_{i' \in S_j^{(t_n)}} |\hat{y}_{i'}^{(t_n)} - y_{i'}| \leq \kappa \max_{i'} |\hat{y}_{i'}^{(t_n)} - y_{i'}|,$$

while for the proxy prediction, we have

$$|\hat{y}_i^{(t_{n+1})} - y_i| = \frac{\lambda}{1+\lambda}|\tilde{y}_i^{(t_n)} - y_i|,$$

where $\kappa := \sqrt{L/\mu}$.

Combining these yields the overall contraction factor:

$$\rho := \kappa\,\frac{\lambda}{1+\lambda}, \qquad \text{with the assumption } \rho < 1 \ (\Leftrightarrow \lambda < 1/(\kappa - 1)).$$

Consequently, the maximum error contracts geometrically as follows:

$$\max_{i'} |\hat{y}_{i'}^{(t_n)} - y_{i'}| \le \rho^n \max_{i'} |\hat{y}_{i'}^{(0)} - y_{i'}|.$$

**Proposition A.2** (Geometric contraction). *Given the contraction rate $\rho$ above, for every $i$ and $n \ge 0$,*

$$|\hat{y}_i^{(t_n)} - y_i| \le \rho^n \max_{i'} |\hat{y}_{i'}^{(0)} - y_{i'}|, \qquad |\tilde{y}_i^{(t_n)} - y_i| \le \kappa\rho^{n-1} \max_{i'} |\hat{y}_{i'}^{(0)} - y_{i'}|.$$

## A.2 FROM PREDICTION CONVERGENCE TO STABLE ASSIGNMENT

Next, we relate the convergence of predictions to the stabilization of cluster assignments. Let $\Delta$ denote the minimum prediction gap defined in Assumption 3:

$$\forall i : \min_{j \ne z_i^*} |\mathbf{x}_i^\top (\tilde{\mathbf{w}}_{z_i^*}^* - \tilde{\mathbf{w}}_j^*)| \ge \Delta > 0.$$

If we define $M := \max_i |\hat{y}_i^{(0)} - y_i|$, then by Proposition A.2,

$$|\tilde{y}_i^{(t_n)} - y_i|, \ |\hat{y}_i^{(t_n)} - y_i| \le \kappa\rho^{n-1}M.$$

Thus, the maximum possible deviation in any score difference is at most $\kappa\rho^{n-1}M$. By selecting

$$n_0 := \left\lceil \frac{\log(2\kappa M/\Delta)}{\log(1/\rho)} \right\rceil + 1,$$

we guarantee that for all $t' \ge t_{n_0}$, the gap between the predicted value and each cluster's optimal regressor is at least $\Delta/2$, ensuring correct and stable assignments to $z_i^*$. Formally:

**Proposition A.3** (Assignment stabilization). *Suppose Assumptions 2–4 hold and $\rho < 1$. Then all cluster assignments stabilize after $t_{n_0}$ high-level epochs:*

$$\forall t' \ge t_{n_0} : \quad z_i^{(t')} = z_i^*, \ \forall i.$$

To summarize, this section provides a high-level view of the error contraction and assignment stabilization mechanism, assuming each alternating update fully minimizes the relevant loss. The formal proof of convergence under practical gradient-based optimization can be found in Appendix B.

# B  LYAPUNOV ANALYSIS

This appendix provides the theoretical proofs for the Lyapunov analysis outlined in Section 3.3. The main results—coercivity, one-epoch descent, and linear convergence—are restated here as in the main text, followed by detailed proofs.

## B.1  COERCIVITY OF OBJECTIVE FUNCTION

We first show that the Lyapunov function $V_\lambda(\phi, \tilde{\mathbb{W}})$ is coercive, meaning that it grows without bound as $(\phi, \tilde{\mathbb{W}})$ moves away from any bounded set. Owing to the structure of our objective, this result follows directly:

**Lemma B.1** (Coercivity). *Let Assumptions 1–2 hold and recall the Lyapunov function $V_\lambda(\phi, \tilde{\mathbb{W}}) = R^{\mathrm{fit}}(\phi, \tilde{\mathbb{W}}^{\mathrm{stop}}) + (1 + \lambda) R^{\mathrm{align}}(\phi, \tilde{\mathbb{W}})$. Then there exist positive constants $\mu_\phi > 0$ and $\mu_{\tilde{\mathbb{W}}} := (1 + \lambda)\mu/N$ such that*
$$V_\lambda(\phi, \tilde{\mathbb{W}}) \geq \tfrac{\mu_\phi}{2} \|\phi\|^2 + \tfrac{\mu_{\tilde{\mathbb{W}}}}{2} \|\tilde{\mathbb{W}}\|^2.$$
*Consequently $V_\lambda(\phi, \tilde{\mathbb{W}}) \to \infty$ whenever $\|(\phi, \tilde{\mathbb{W}})\| \to \infty$.*

*Proof.*
Write $J_\phi(x) = \frac{\partial W_\phi(x)}{\partial \phi}$. By Assumption 1, $m_x \leq \|J_\phi(x)\| \leq L_x$. Using the Gauss-Newton approximation, we have $\nabla_\phi^2 R^{\mathrm{fit}} \succeq 2m_x^2 \mathbb{X}_{min}^2 I$. Set $\mu_\phi := 2m_x^2 \mathbb{X}_{min}^2$; then $R^{\mathrm{fit}} \geq \frac{\mu_\phi}{2} \|\phi\|^2$.

The term $R^{\mathrm{align}}$ is quadratic in $\tilde{\mathbb{W}}$, and its Hessian equals $\frac{1}{N} \sum_{i \in S_j} \mathbb{x}_i \mathbb{x}_i^\top \succeq \mu I$ (by Assumption 2); hence $R^{\mathrm{align}} \geq \frac{\mu}{2} \|\tilde{\mathbb{W}}\|^2$.

Adding the two bounds and multiplying the $R^{\mathrm{align}}$ term by $1 + \lambda$ yields:
$$V_\lambda(\phi, \tilde{\mathbb{W}}) = R^{\mathrm{fit}}(\phi, \tilde{\mathbb{W}}^{\mathrm{stop}}) + (1 + \lambda) R^{\mathrm{align}}(\phi, \tilde{\mathbb{W}}) \geq \frac{\mu_\phi}{2} \|\phi\|^2 + \frac{(1+\lambda)\mu}{2} \|\tilde{\mathbb{W}}\|^2$$
$\square$

## B.2  SMOOTHNESS OF OBJECTIVE FUNCTION

**Lemma B.2** (Gradient Lipschitz continuity of $V_\lambda(\phi, \tilde{\mathbb{W}})$). *Under Assumptions 1–2, the gradient of $V_\lambda(\phi, \tilde{\mathbb{W}})$ is $L_V$–Lipschitz, where*
$$L_V = \max\Big\{(1 + \lambda) L, \ (4 + 2\lambda) L_x^2 \mathbb{X}_{max}^2\Big\}.$$

*Proof.*
Define $\mathbf{X}_j := [\mathbb{x}_i]_{i \in S_j}^\top$. The alignment part splits cluster–wise:
$$R^{\mathrm{align}}(\phi, \tilde{\mathbb{W}}) = \frac{1}{N} \sum_{j=1}^K \big\|X_j \tilde{\mathbb{w}}_j - r_j(\phi)\big\|^2.$$

where $r_j(\phi) := [\mathbb{x}_i^\top \hat{\mathbb{w}}_i]_{i \in S_j}^\top$

Hence
$$\nabla_{\tilde{\mathbb{W}}}^2 R^{\mathrm{align}}(\phi, \tilde{\mathbb{W}}) = \mathrm{diag}\big(\tfrac{1}{N} X_j^\top X_j\big)_{j=1}^K.$$

By Assumption 2, $\frac{1}{N} X_j^\top X_j \preceq LI$ for every $j$. Therefore, $\|\nabla_{\tilde{\mathbb{W}}}^2 V_\lambda(\phi, \tilde{\mathbb{W}})\| \leq (1 + \lambda)L$.

For each sample $i$, $\big\|\nabla_\phi \mathbb{x}_i^\top \hat{\mathbb{w}}_i\big\| \leq L_x \mathbb{X}_{\max}$ (by Assumption 1). Therefore,
$$\|\nabla^2 R^{\mathrm{fit}}(\phi, \tilde{\mathbb{W}})\| \leq 2L_x^2 \mathbb{X}_{max}^2, \qquad \|\nabla_\phi^2 R^{\mathrm{align}}(\phi, \tilde{\mathbb{W}})\| \leq (1 + \lambda) 2L_x^2 \mathbb{X}_{max}^2.$$

Add the two contributions gives $\|\nabla_\phi^2 V_\lambda(\phi, \tilde{\mathbb{W}})\| \leq (4 + 2\lambda) L_x^2 \mathbb{X}_{max}^2$.

The full Hessian of $V_\lambda(\phi, \tilde{\mathbb{W}})$ is block-diagonal, so its operator norm is the maximum of the two block norms computed above. Hence,
$$L_V = \max\{(1 + \lambda)L, \ (4 + 2\lambda)L_x^2 \mathbb{X}_{max}^2\},$$
which proves the claim. $\square$

**Proposition 3.1** (One-epoch descent). Under Assumptions 1 and 2, suppose the cluster assignment $z^{(t)}$ remains fixed during epoch $t$. Define

$$L_V := \max\Big\{(1+\lambda)L,\ (4+2\lambda)L_x^2\mathbb{X}_{\max}^2\Big\}.$$

For any step-size $0 < \eta \leq 1/L_V$, the simultaneous gradient updates guarantee a strict decrease of the Lyapunov function:

$$V_\lambda\big(\phi^{(t+1)}, \tilde{\mathbb{W}}^{(t+1)}\big) \ \leq\ V_\lambda\big(\phi^{(t)}, \tilde{\mathbb{W}}^{(t)}\big) - \frac{\eta}{2}\Big(\|\phi^{(t+1)} - \phi^{(t)}\|^2 + \|\tilde{\mathbb{W}}^{(t+1)} - \tilde{\mathbb{W}}^{(t)}\|^2\Big).$$

*Proof.*
Because $V_\lambda(\phi, \tilde{\mathbb{W}})$ is $L_V$-smooth (by Lemma B.2), for any update vector $\theta_\Delta := (\phi^{(t+1)} - \phi^{(t)}, \tilde{\mathbb{W}}^{(t+1)} - \tilde{\mathbb{W}}^{(t)})$ and current vector $\theta^{(t)} := (\phi^{(t)}, \tilde{\mathbb{W}}^{(t)})$, the descent theorem (Nesterov et al. (2018, Theorem 2.1.5)) gives

$$V_\lambda(\theta^{(t)} + \theta_\Delta) \ \leq\ V_\lambda(\theta^{(t)}) + \langle \nabla V_\lambda(\theta^{(t)}), \theta_\Delta \rangle + \frac{L_V}{2}\|\theta_\Delta\|^2.$$

Choose $\theta_\Delta = -\eta \nabla V_\lambda(\phi^{(t)}, \tilde{\mathbb{W}}^{(t)})$ with $0 < \eta \leq 1/L_V$.

Then,

$$V_\lambda(\phi^{(t+1)}, \tilde{\mathbb{W}}^{(t+1)}) \leq V_\lambda(\phi^{(t)}, \tilde{\mathbb{W}}^{(t)}) - \eta\|\nabla V_\lambda(\phi^{(t)}, \tilde{\mathbb{W}}^{(t)})\|^2 + \frac{\eta^2 L_V}{2}\|\nabla V_\lambda(\phi^{(t)}, \tilde{\mathbb{W}}^{(t)})\|^2$$

$$= V_\lambda(\phi^{(t)}, \tilde{\mathbb{W}}^{(t)}) - \eta\Big(1 - \frac{\eta L_V}{2}\Big)\|\nabla V_\lambda(\phi^{(t)}, \tilde{\mathbb{W}}^{(t)})\|^2.$$

Since $\eta L_V \leq 1$, the parenthesis is at least $\frac{1}{2}$, yielding the stated decrease: $V_\lambda(\phi^{(t+1)}, \tilde{\mathbb{W}}^{(t+1)}) \leq V_\lambda(\phi^{(t)}, \tilde{\mathbb{W}}^{(t)}) - \frac{\eta}{2}\|\nabla V_\lambda(\phi^{(t)}, \tilde{\mathbb{W}}^{(t)})\|^2.$ □

**Theorem 3.2** (Linear convergence). Let Assumptions 1–4 hold. Define

$$\mu_V := \min\Big\{2m_x^2\mathbb{X}_{\min}^2,\ (1+\lambda)\mu\Big\}, \quad L_V := \max\Big\{(1+\lambda)L,\ (4+2\lambda)L_x^2\mathbb{X}_{\max}^2\Big\}, \quad q := \frac{L_V - \mu_V}{L_V + \mu_V}.$$

Gradient descent with the optimal step-size $\eta_\star = 2/(\mu_V + L_V)$ ensures that the stacked parameter vector $\theta = (\phi, \tilde{\mathbb{W}})$ converges linearly at the rate $q$, while the Lyapunov gap contracts at the rate $q^2$.

*Proof.*
Lemma B.1 shows that, under Assumptions 1–2, the Hessian satisfies

$$\nabla^2 V_\lambda \ \succeq\ \mu_V I, \qquad \mu_V := \min\Big\{2m_x^2\mathbb{X}_{min}^2,\ (1+\lambda)\mu\Big\},$$

Lemma B.2 gives a global Lipschitz bound $\|\nabla^2 V_\lambda\| \leq L_V$ with $L_V = \max\{(1+\lambda)L,\ (4+2\lambda)L_x^2\mathbb{X}_{max}^2\}$.

For epoch $t$, the update $\theta^{(t+1)} = \theta^{(t)} - \eta\nabla V_\lambda(\theta^{(t)})$, $\theta^{(t)} := (\phi^{(t)}, \tilde{\mathbb{W}}^{(t)})$ is standard gradient descent on an $L_V$-smooth, $\mu_V$-strongly convex function.

Choosing any stepsize $0 < \eta \leq 2/(\mu_V + L_V)$ gives the classical contraction (Nesterov et al. (2018, Theorem 2.1.15)):

$$V_\lambda(\theta^{(t)}) - V_\lambda^\star \ \leq\ \frac{L_V}{2}\Big(\frac{L_V/\mu_V - 1}{L_V/\mu_V + 1}\Big)^{2t}\|\theta^{(0)} - \theta^\star\|^2,$$

where $\theta^\star$ is the optimal point of $V_\lambda(\theta)$ and $V_\lambda^\star := V_\lambda(\theta^\star)$.

The same theorem gives the one-step contraction, yielding the geometric bound:

$$\|\theta^{(t)} - \theta^\star\|^2 \leq \left(\frac{L_V/\mu_V - 1}{L_V/\mu_V + 1}\right)^t \|\theta^{(0)} - \theta^\star\|^2.$$

In other words, the stacked parameter vector $\theta = (\phi, \tilde{\mathbb{W}})$ converges linearly with ratio $\frac{L_V/\mu_V - 1}{L_V/\mu_V + 1}$, while the objective gap contracts with ratio $\left(\frac{L_V/\mu_V - 1}{L_V/\mu_V + 1}\right)^2$.

$\square$

These results together provide a rigorous foundation for the Lyapunov-based convergence guarantees stated in the main text.

# C PAC-STYLE GENERALIZATION BOUND

This appendix provides the full proof of the PAC-style generalization bound stated in Theorem 3.3 of the main text. We begin with a general Rademacher-complexity-based risk bound for squared loss, then specialize it to the structure of CG-CLR, resulting in a bound that is uniform over all model parameters.

## C.1 FORMAL DEFINITION OF EMPIRICAL AND EXPECTED RISKS

We formally define the empirical quantities used in the theoretical analysis in Section 3.3. Throughout, let $\ell(\hat{y}, y) := (\hat{y} - y)^2$ denote the squared loss.

**Predictor induced by CG-CLR.** Given parameters $(\boldsymbol{\phi}, \tilde{\mathbb{W}})$, the prediction of CG-CLR on an input $\boldsymbol{x}$ is

$$h_{\boldsymbol{\phi}, \tilde{\mathbb{W}}}(\boldsymbol{x}) := \mathbb{x}^\top \tilde{\mathbb{w}}_{z_{\boldsymbol{\phi}}(\boldsymbol{x})},$$

where $z_{\boldsymbol{\phi}}(\boldsymbol{x})$ is the hard-assignment rule defined in equation 2.

**Expected risk.** For a predictor $h_{\boldsymbol{\phi}, \tilde{\mathbb{W}}}$, the expected risk under the data distribution $D$ is

$$\mathfrak{R}_{\text{test}} := \mathbb{E}_{(\boldsymbol{x}, y) \sim D} \big[ \ell(h_{\boldsymbol{\phi}, \tilde{\mathbb{W}}}(\boldsymbol{x}), y) \big].$$

**Empirical risk on the training sample.** Given i.i.d. training data $\{(\boldsymbol{x}_i, y_i)\}_{i=1}^N$, the empirical risk is

$$\mathfrak{R}_{\text{train}} := \frac{1}{N} \sum_{i=1}^N \ell\big(h_{\boldsymbol{\phi}, \tilde{\mathbb{W}}}(\boldsymbol{x}_i), y_i\big).$$

**Objective minimized during training.** CG-CLR minimizes the composite empirical objective

$$\hat{\mathfrak{R}} := R^{\text{fit}}(\boldsymbol{\phi}, \tilde{\mathbb{W}}^{\text{stop}}) + (1 + \lambda) R^{\text{align}}(\boldsymbol{\phi}, \tilde{\mathbb{W}}),$$

where $R^{\text{fit}}$ measures the prediction error between the codebook predictions and the responses, and $R^{\text{align}}$ encourages agreement between the proxy predictions and the quantized codebook predictions.

## C.2 GENERALIZATION VIA RADEMACHER COMPLEXITY

For any measurable function $f$, we denote the true and empirical risks as

$$L(f) = \mathbb{E}_{(\boldsymbol{x}, y) \sim \mathcal{D}}\big[\ell(f(\boldsymbol{x}), y)\big], \quad \hat{L}(f) = \frac{1}{N} \sum_{i=1}^N \ell\big(f(\boldsymbol{x}_i), y_i\big)$$

where the loss function $\ell : \mathbb{R} \times \mathbb{R} \to [0, Y_{\max}^2]$ is $4Y_{\max}$-Lipschitz in its first argument:

$$|\ell(u, y) - \ell(v, y)| = |(u - y)^2 - (v - y)^2| = |u - v||u + v - 2y| \leq 4Y_{\max}|u - v|.$$

**Lemma C.1** (Rademacher-based Risk Bound). *Let $\mathcal{F}$ be a class of real-valued functions such that $\sup_{\boldsymbol{x}} |f(\boldsymbol{x})| \leq Y_{\max}$ for all $f \in \mathcal{F}$. Then for any $\delta \in (0, 1)$, with probability at least $1 - \delta$ over an i.i.d. sample of size $N$, every $f \in \mathcal{F}$ satisfies*

$$L(f) \leq \hat{L}(f) + 8Y_{\max} \widehat{\mathcal{R}}_N(\mathcal{F}) + Y_{\max}^2 \sqrt{\frac{\ln(1/\delta)}{2N}},$$

*where*

$$\widehat{\mathcal{R}}_N(\mathcal{F}) = \mathbb{E}_\sigma \Big[ \sup_{f \in \mathcal{F}} \frac{1}{N} \sum_{i=1}^N \sigma_i f(\boldsymbol{x}_i) \Big],$$

*and $\sigma_i \in \{\pm 1\}$ are independent Rademacher random variables.*

*Proof.*

Let $S = (d_1, \ldots, d_N)$ with $d_i = (\boldsymbol{x}_i, y_i)$, and define

$$\epsilon(S) = \sup_{f \in \mathcal{F}} \Big( L(f) - \hat{L}(f) \Big) = \sup_{f \in \mathcal{F}} \Big( \mathbb{E}[\ell(f(\boldsymbol{x}), y)] - \frac{1}{N} \sum_{i=1}^{N} \ell(f(\boldsymbol{x}_i), y_i) \Big).$$

By the symmetrization argument (Bartlett & Mendelson (2002, Appendix B)),

$$\mathbb{E}[\epsilon(S)] \leq \mathbb{E}_\sigma \Big[ \sup_{f \in \mathcal{F}} \frac{2}{N} \sum_{i=1}^{N} \sigma_i \, \ell\big(f(\boldsymbol{x}_i), y_i\big) \Big].$$

For each fixed $y$, define the centered loss

$$\psi_y(u) = \ell(u, y) - \ell(0, y).$$

Then $\psi_y$ is $4Y_{\max}$-Lipschitz and satisfies $\psi_y(0) = 0$.

By Talagrand's Lemma (Mohri (2018, Lemma 5.7)),

$$\mathbb{E}_\sigma \Big[ \sup_{f \in \mathcal{F}} \frac{1}{N} \sum_{i=1}^{N} \sigma_i \, \psi_{y_i}\big(f(\boldsymbol{x}_i)\big) \Big] \leq 4Y_{\max} \, \mathbb{E}_\sigma \Big[ \sup_{f \in \mathcal{F}} \frac{1}{N} \sum_{i=1}^{N} \sigma_i \, f(\boldsymbol{x}_i) \Big] = 4Y_{\max} \, \mathcal{R}_N(\mathcal{F}).$$

Subtracting the constant $\ell(0, y_i)$ does not affect the supremum, so

$$\mathbb{E}_\sigma \Big[ \sup_{f \in \mathcal{F}} \frac{1}{N} \sum_{i=1}^{N} \sigma_i \, \ell\big(f(\boldsymbol{x}_i), y_i\big) \Big] = \mathbb{E}_\sigma \Big[ \sup_{f \in \mathcal{F}} \frac{1}{N} \sum_{i=1}^{N} \sigma_i \, \psi_{y_i}\big(f(\boldsymbol{x}_i)\big) \Big].$$

Combining the above yields

$$\mathbb{E}[\epsilon(S)] \leq 8 \, Y_{\max} \, \mathcal{R}_N(\mathcal{F}).$$

Now, suppose $S'$ differs from $S$ only in the $i$-th sample, $d_i \to d_i'$. For any fixed $f$,

$$\Big| \frac{1}{N} \ell\big(f(\boldsymbol{x}_i), y_i\big) - \frac{1}{N} \ell\big(f(\boldsymbol{x}_i'), y_i'\big) \Big| \leq \frac{Y_{\max}^2}{N}.$$

Thus, taking the supremum over $f$,

$$\big| \epsilon(S) - \epsilon(S') \big| \leq \frac{Y_{\max}^2}{N}.$$

Therefore, $\epsilon(S)$ satisfies the bounded difference property with constants $c_i = Y_{\max}^2 / N$.

By McDiarmid's inequality (Mohri (2018, Theorem D.8)), for any $\epsilon_0 > 0$,

$$\Pr\big(\epsilon(S) - \mathbb{E}[\epsilon(S)] > \epsilon_0\big) \leq \exp\Big(-\frac{2\epsilon_0^2}{\sum_{i=1}^{N} c_i^2}\Big) = \exp\Big(-\frac{2N \, \epsilon_0^2}{Y_{\max}^4}\Big).$$

Setting the right-hand side equal to $\delta$ gives

$$\epsilon_0 = Y_{\max}^2 \sqrt{\frac{\ln(1/\delta)}{2N}},$$

so with probability at least $1 - \delta$,

$$\epsilon(S) \leq \mathbb{E}[\epsilon(S)] + Y_{\max}^2 \sqrt{\frac{\ln(1/\delta)}{2N}}.$$

Combining the above bounds, for every $f \in \mathcal{F}$,

$$L(f) \leq \hat{L}(f) + 8Y_{\max} \widehat{\mathcal{R}}_N(\mathcal{F}) + Y_{\max}^2 \sqrt{\frac{\ln(1/\delta)}{2N}}.$$

$\square$

### C.3 CG-CLR-SPECIFIC DECOMPOSITION

We now apply the general bound to our clusterwise model class.

**Theorem 3.3** (Generalization bound). *For any $\delta \in (0, 1)$, with probability at least $1 - \delta$,*

$$\mathfrak{R}_{\text{test}} \leq \mathfrak{R}_{\text{train}} + 16c_0 c_1 \max_{j \leq K} \|\tilde{w}_j\| \sqrt{\frac{dM \log d \log 2N}{N}} + \frac{8c_1 \max_{j \leq K} \|\tilde{w}_j\|}{\sqrt{N}} + Y_{\max}^2 \sqrt{\frac{\log(1/\delta)}{2N}},$$

*where $c_0$ is an absolute constant for standard ReLU MLPs and $c_1 := Y_{\max} \mathbb{X}_{\max}$.*

*Proof.*
We bound the empirical Rademacher complexity

$$\widehat{\mathfrak{R}}_N(\mathcal{H}) = \mathbb{E}_\sigma \Big[ \sup_{(\phi, \tilde{W})} \frac{1}{N} \sum_{i=1}^N \sigma_i h_{\phi, \tilde{W}}(\boldsymbol{x}_i) \Big],$$

where

$$h_{\phi, \tilde{W}}(\boldsymbol{x}) = \mathbb{x}^\top \tilde{w}_{z_\phi(\boldsymbol{x})}, \quad z_\phi(\boldsymbol{x}) = \operatorname*{argmin}_{j \leq K} \big( \mathbb{x}^\top (\hat{w}_i - \tilde{w}_j) \big)^2.$$

Define

$$\mathcal{G} = \big\{ \boldsymbol{x} \mapsto z_\phi(\boldsymbol{x}) \big\} \subset \{ \mathcal{X} \to \{j\}_{j=1}^K \}, \qquad \mathcal{C} = \big\{ (\boldsymbol{x}, j) \mapsto \mathbb{x}^\top \tilde{w}_j \big\} \subset \{ \mathcal{X} \times \{j\}_{j=1}^K \to \mathbb{R} \}.$$

Then $\mathcal{H} = \mathcal{C} \circ \mathcal{G}$.

**Lemma C.2** (Codebook Lipschitz). *Assume $\forall j \ \|\tilde{w}_j\| \leq \mathbb{W}_{\max}$. Define*

$$\psi_{\mathbb{x}}(j) = \mathbb{x}^\top \tilde{w}_j.$$

*With the discrete metric $d(j, j') = \mathbb{1}\{j \neq j'\}$, we have*

$$|\psi_{\mathbb{x}}(j) - \psi_{\mathbb{x}}(j')| \leq 2 \mathbb{X}_{\max} \mathbb{W}_{\max} \, d(j, j').$$

*Thus, each $\psi_{\mathbb{x}}$ is $(2\mathbb{X}_{\max}\mathbb{W}_{\max})$-Lipschitz.*

For any fixed code vectors $\{\tilde{w}_j^*\}$, consider

$$\widehat{\mathfrak{R}}_N(\mathcal{C} \circ \mathcal{G}) = \mathbb{E}_\sigma \Big[ \sup_{g \in \mathcal{G}} \frac{1}{N} \sum_i \sigma_i \mathbb{x}_i^\top \tilde{w}_{g(\boldsymbol{x}_i)}^* \Big].$$

Since the map $j \mapsto \mathbb{x}_i^\top \tilde{w}_j^*$ is $(2\mathbb{X}_{\max}\mathbb{W}_{\max})$-Lipschitz on the discrete set, the Ledoux–Talagrand contraction lemma (Ledoux & Talagrand, 2013, Theorem 4.4) gives

$$\widehat{\mathfrak{R}}_N(\mathcal{C} \circ \mathcal{G}) \leq 2\mathbb{X}_{\max}\mathbb{W}_{\max} \, \widehat{\mathfrak{R}}_N(\mathcal{G}).$$

Allowing the codebook to vary introduces an additional term. By sup-subadditivity, for any fixed $(\phi^*, \tilde{W}^*)$,

$$\sup_{\phi, \tilde{W}} \sum_i \sigma_i \mathbb{x}_i^\top \tilde{w}_{z_\phi(\boldsymbol{x}_i)} \leq \underbrace{\sup_\phi \sum_i \sigma_i \mathbb{x}_i^\top \tilde{w}_{z_\phi(\boldsymbol{x}_i)}^*}_{A} + \underbrace{\sup_{\tilde{W}} \sum_i \sigma_i \mathbb{x}_i^\top \tilde{w}_{z_{\phi^*}(\boldsymbol{x}_i)}}_{B}.$$

Term $A$ is bounded as above, and term $B$ is the Rademacher sum over the linear class $\mathcal{C}$:

$$A \leq 2N\mathbb{X}_{\max}\mathbb{W}_{\max}\widehat{\mathfrak{R}}_N(\mathcal{G}), \qquad B = N\widehat{\mathfrak{R}}_N(\mathcal{C}).$$

Dividing by $N$ and taking expectation in $\sigma$ yields

$$\widehat{\mathfrak{R}}_N(\mathcal{H}) \leq \widehat{\mathfrak{R}}_N(\mathcal{C}) + 2\mathbb{X}_{\max}\mathbb{W}_{\max} \widehat{\mathfrak{R}}_N(\mathcal{G}).$$

Now, for

$$\mathcal{C} = \Big\{ (\boldsymbol{x}, j) \mapsto \mathrm{x}^\top \tilde{\mathrm{w}}_j \, : \, \|\tilde{\mathrm{w}}_j\| \leq \mathbb{W}_{\max} \Big\},$$

we have

$$\widehat{\mathfrak{R}}_N(\mathcal{C}) = \mathbb{E}_\sigma \Big[ \sup_{j, \|\tilde{\mathrm{w}}_j\| \leq \mathbb{W}_{\max}} \frac{1}{N} \sum_{i=1}^N \sigma_i \, \mathrm{x}_i^\top \tilde{\mathrm{w}}_j \Big]$$

$$= \mathbb{W}_{\max} \, \mathbb{E}_\sigma \Big\| \frac{1}{N} \sum_{i=1}^N \sigma_i \, \mathrm{x}_i \Big\| \quad \text{(by the Cauchy–Schwarz duality)}$$

$$\leq \frac{\mathbb{W}_{\max} \, \mathbb{X}_{\max}}{\sqrt{N}},$$

where the last inequality is the standard Rademacher complexity bound for a Euclidean ball in $\mathbb{R}^p$ (Mohri (2018, Theorem 11.11)).

Next,

$$\mathcal{G} = \big\{ \boldsymbol{x} \mapsto W_\phi(\boldsymbol{x}) \big\}$$

is the hypothesis class implemented by a depth-$M$ ReLU MLP with $d$ trainable parameters.

By the nearly-tight VC and pseudo-dimension bounds of Bartlett et al. (2019, Theorem 3 and 7), its pseudo-dimension satisfies

$$\mathrm{Pdim}(\mathcal{G}) = O\big(d \, M \log d\big).$$

By the standard conversion from pseudo-dimension to Rademacher complexity (Mohri (2018, Theorem 11.6)),

$$\widehat{\mathfrak{R}}_N(\mathcal{G}) \, \leq \, \sqrt{\frac{2 \, \mathrm{Pdim}(\mathcal{G}) \, \ln(2N)}{N}} \, = \, O\Big( \sqrt{\frac{d \, M \ln d \, \ln(2N)}{N}} \Big).$$

Combining Lemma C.2 with the decomposition

$$\widehat{\mathfrak{R}}_N(\mathcal{H}) \leq \widehat{\mathfrak{R}}_N(\mathcal{C}) + 2 \mathbb{X}_{\max} \mathbb{W}_{\max} \widehat{\mathfrak{R}}_N(\mathcal{G}),$$

and using

$$\widehat{\mathfrak{R}}_N(\mathcal{C}) \leq \frac{\mathbb{W}_{\max} \, \mathbb{X}_{\max}}{\sqrt{N}}, \qquad \widehat{\mathfrak{R}}_N(\mathcal{G}) \leq c_0 \sqrt{\frac{dM \ln d \ln(2N)}{N}},$$

where $c_0$ is an absolute constant for standard ReLU MLPs, we conclude that, for any $\delta \in (0, 1)$, with probability at least $1 - \delta$,

$$L\big(h_{\boldsymbol{\phi}, \tilde{\mathbb{W}}}\big) \, \leq \, \hat{L}\big(h_{\boldsymbol{\phi}, \tilde{\mathbb{W}}}\big) + 8 Y_{\max} \Big( \tfrac{\mathbb{W}_{\max} \, \mathbb{X}_{\max}}{\sqrt{N}} + 2 \mathbb{X}_{\max} \mathbb{W}_{\max} c_0 \sqrt{\tfrac{dM \ln d \ln(2N)}{N}} \Big) + Y_{\max}^2 \sqrt{\tfrac{\ln(1/\delta)}{2N}}.$$

In particular,

$$\mathfrak{R}_{\text{test}} \, \leq \, \mathfrak{R}_{\text{train}} + 16 c_0 c_1 \max_{j \leq K} \|\tilde{\mathrm{w}}_j\| \sqrt{\frac{dM \log d \log 2N}{N}} + \frac{8 c_1 \max_{j \leq K} \|\tilde{\mathrm{w}}_j\|}{\sqrt{N}} + Y_{\max}^2 \sqrt{\frac{\log(1/\delta)}{2N}},$$

where $c_1 := Y_{\max} \mathbb{X}_{\max}$, which recovers Theorem 3.3. $\qquad\square$

**Connection to the training objective.** Note that our training objective upper-bounds the empirical prediction risk:

$$\mathfrak{R}_{\text{train}}(\boldsymbol{\phi}, \tilde{\mathbb{W}}) \, = \, R^{\text{fit}}(\boldsymbol{\phi}, \tilde{\mathbb{W}}^{\text{stop}}) \, \leq \, \hat{\mathfrak{R}}(\boldsymbol{\phi}, \tilde{\mathbb{W}}).$$

Therefore, Theorem 3.3 also yields a bound in terms of the optimization objective:

$$\mathfrak{R}_{\text{test}} \, \leq \, \hat{\mathfrak{R}} + 16 c_0 c_1 \max_{j \leq K} \|\tilde{\mathrm{w}}_j\| \sqrt{\frac{dM \log d \log(2N)}{N}} + \frac{8 c_1 \max_{j \leq K} \|\tilde{\mathrm{w}}_j\|}{\sqrt{N}} + Y_{\max}^2 \sqrt{\frac{\log(1/\delta)}{2N}}.$$

# D  $F-$TEST DERIVATION

This appendix derives the sequential $F$-test procedure used to determine the appropriate number of clusters in our model. The test evaluates, at each step, whether introducing an additional cluster leads to a statistically significant reduction in the residual sum of squares.

**Proposition 3.4** (Sequential $F$-test). The necessity of adding another cluster is tested sequentially using nested-model F-statistics:

$$F_{K\to K+1} = \frac{(\text{SSE}_K - \text{SSE}_{K+1})/(p+1)}{\text{SSE}_{K+1}/(N-(K+1)(p+1))} \sim F_{p+1,N-(K+1)(p+1)},$$

and the number of clusters $K$ is selected according to statistical significance at a given level $\alpha$.

*Proof.*
We consider a fixed assignment vector $z = (z_1, \ldots, z_N) \in [1, \ldots, K]^N$. For each cluster $j \in [1, \ldots, K]$, let

$$\mathbf{X}_j := \text{diag}(\mathbf{1}_{\mathcal{S}_j})[\mathbb{x}_1 : \cdots : \mathbb{x}_N]^\top \in \mathbb{R}^{N\times(p+1)},$$

where $\mathcal{S}_j := \{i|z_i = j\}$ is the index set for cluster $j$.

The *restricted* (null) model with $K$ clusters is then given by

$$\boldsymbol{y} = X_{(K)}\boldsymbol{\beta}_{(K)} + \boldsymbol{\varepsilon}, \qquad X_{(K)} := [\mathbf{X}_1 : \cdots : \mathbf{X}_K] \in \mathbb{R}^{N\times K(p+1)}, \quad \boldsymbol{\beta}_{(K)} := [\tilde{\mathbb{w}}_1^\top, \ldots, \tilde{\mathbb{w}}_K^\top]^\top,$$

where $\boldsymbol{\varepsilon} \sim N(0, \sigma^2)$.

To test whether adding a $(K+1)$th cluster is necessary, we consider a new partition of the data in which one or more samples are reassigned from their current clusters to form the candidate new cluster. This induces a new set of cluster assignments

$$z' = (z'_1, \ldots, z'_N) \in [1, \ldots, K+1]^N,$$

where there exists at least one $i$ such that $z'_i = K+1$ and $z'_i \neq z_i$.

The new partition gives rise to a modified block structure for the design matrix, where

$$X'_{(K+1)} := [\mathbf{X}'_1, \ldots, \mathbf{X}'_{K+1}] \in \mathbb{R}^{N\times(K+1)(p+1)},$$

with

$$\mathbf{X}'_j := \text{diag}(\mathbf{1}_{\mathcal{S}'_j})[\mathbb{x}_1 : \cdots : \mathbb{x}_N]^\top, \quad \text{for } j = 1, \ldots, K+1,$$

and the updated index set for cluster $j$ is defined as $\mathcal{S}'_j := \{i|z'_i = j\}$.

The extended (alternative) model is now

$$\boldsymbol{y} = X'_{(K+1)}\boldsymbol{\beta}'_{(K+1)} + \boldsymbol{\varepsilon},$$

where $\boldsymbol{\beta}'_{(K+1)} := [(\tilde{\mathbb{w}}'_1)^\top, \ldots, (\tilde{\mathbb{w}}'_{K+1})^\top]^\top$.

This model introduces exactly $p+1$ additional parameters relative to the restricted model, due to the addition of the new cluster regressor $\tilde{\mathbb{w}}'_{K+1}$, while allowing the other cluster assignments to be updated accordingly.

Let

$$\widehat{\boldsymbol{\beta}}_{(K)} := (X_{(K)}^\top X_{(K)})^{-1} X_{(K)}^\top \boldsymbol{y}, \qquad \widehat{\boldsymbol{\beta}'}_{(K+1)} := (X_{(K+1)}'^\top X'_{(K+1)})^{-1} X_{(K+1)}'^\top \boldsymbol{y},$$

be the least-squares solutions for each model, and define

$$\text{SSE}_K := \left\|\boldsymbol{y} - X_{(K)}\widehat{\boldsymbol{\beta}}_{(K)}\right\|_2^2, \qquad \text{SSE}_{K+1} := \left\|\boldsymbol{y} - X'_{(K+1)}\widehat{\boldsymbol{\beta}'}_{(K+1)}\right\|_2^2.$$

Since the restricted model is nested within the full model, the standard lack-of-fit statistic is given by

$$F_{K\to K+1} := \frac{(\text{SSE}_K - \text{SSE}_{K+1})/(p+1)}{\text{SSE}_{K+1}/(N-(K+1)(p+1))} \sim F_{p+1,\, N-(K+1)(p+1)} \quad \text{under } H_0,$$

where $H_0$ denotes the null hypothesis that the additional cluster does not provide a significant improvement in fit. $\qquad\square$

**Sequential decision rule.** The sequential testing procedure proceeds as follows: Starting with $K = 1$ (the global linear fit), we increment $K$ as long as the statistic

$$F_{K \to K+1} > F_{p+1,\, N-(K+1)(p+1)}^{(\alpha/2)}$$

(where the superscript denotes the upper $\alpha/2$-quantile of the $F$-distribution). The process stops at the smallest $K$ where the null hypothesis cannot be rejected at level $\alpha$.

The final selected number of clusters,

$$\widehat{K} \in \left\{ 1, \ldots, \left\lceil \frac{N}{p+1} \right\rceil - 1 \right\},$$

thus provides a data-driven estimate of the number of linear components supported by the observed data. This sequential rule controls the risk of overfitting by requiring each added cluster to achieve a statistically significant reduction in residual variance.

# E SIMULATION STUDY

## E.1 LOW-DIMENSIONAL PIECEWISE-LINEAR FUNCTION

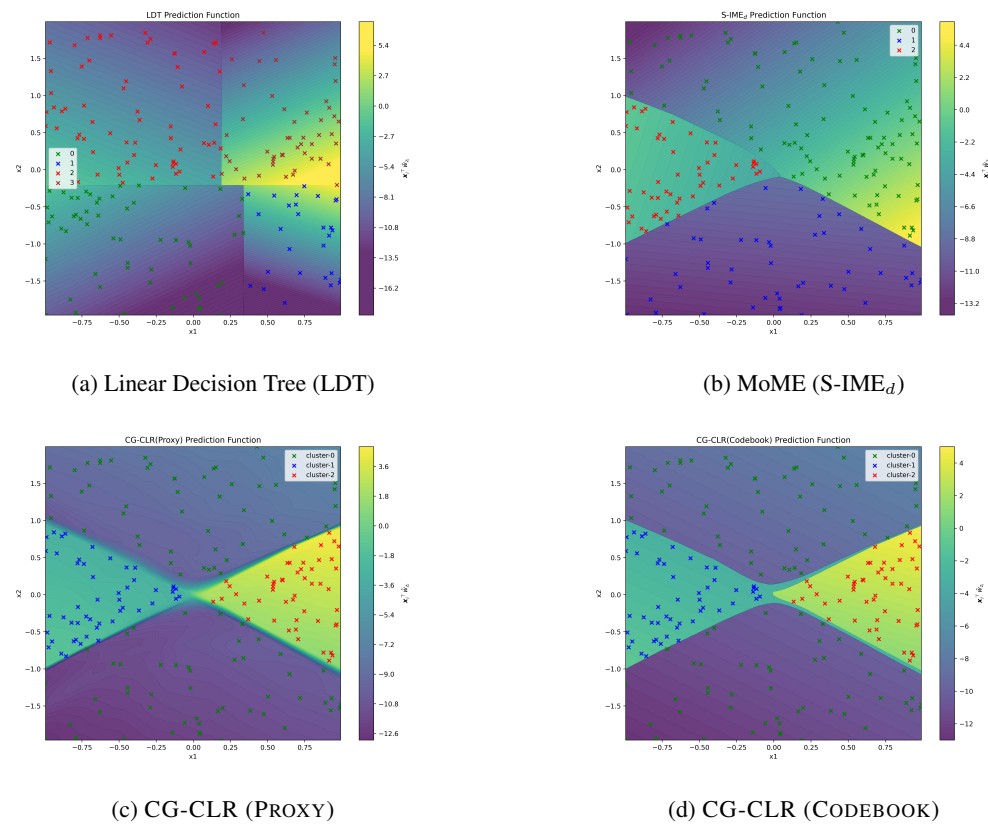

(a) Linear Decision Tree (LDT)

(b) MoME (S-IME$_d$)

(c) CG-CLR (PROXY)

(d) CG-CLR (CODEBOOK)

Figure 4: Function approximation and Clustering results: (a) LDT, (b) S-IME$_d$, (c) proxy network, (d) codebook regressors

Table 3: F-test Results under $H_0 : K \leq k_0$ and $H_1 : K > k_0$ where $p = 2$, $N = 200$, and true $K = 3$.

| $\mathbf{H_0}$ | $\mathbf{SSE_{H_0}}$ | $\mathbf{SSE_{H_1}}$ | $\mathbf{F}$ | **p-value** |
|---|---|---|---|---|
| $K \leq 1$ | 5398.91 | 139.56 | 2436.9 | 0.000 |
| $K \leq 2$ | 139.56 | 1.94 | 4507.2 | 0.000 |
| $K \leq 3$ | 1.94 | 1.84 | 3.3 | **0.038** |
| $K \leq 4$ | 1.84 | 1.80 | 1.2 | **0.586** |

## E.2 HIGH-DIMENSIONAL GAUSSIAN MIXTURE COVARIATES

The main simulation in Section E.1 focuses on a two-dimensional, piecewise-linear function with clearly separated clusters. While this setting is useful for visualizing the behavior of CG-CLR, it is deliberately benign: the covariate clusters are well separated and the underlying linear experts differ substantially. To assess whether CG-CLR also behaves well in more realistic, noisy settings with overlapping clusters and high-dimensional covariates, we conduct an additional experiment based on a Gaussian mixture model in $p = 10$ dimensions.

Conceptually, this experiment is designed to mimic a regime in which (i) the covariates arise from several correlated Gaussian components that partially overlap in $\mathbb{R}^{10}$, (ii) each component is governed by its own linear regression model, and (iii) some regions of the covariate space are intrinsically ambiguous, in the sense that both the covariate distribution and the underlying linear functions are very similar across components. In such regions, any purely covariate-based assignment rule (including

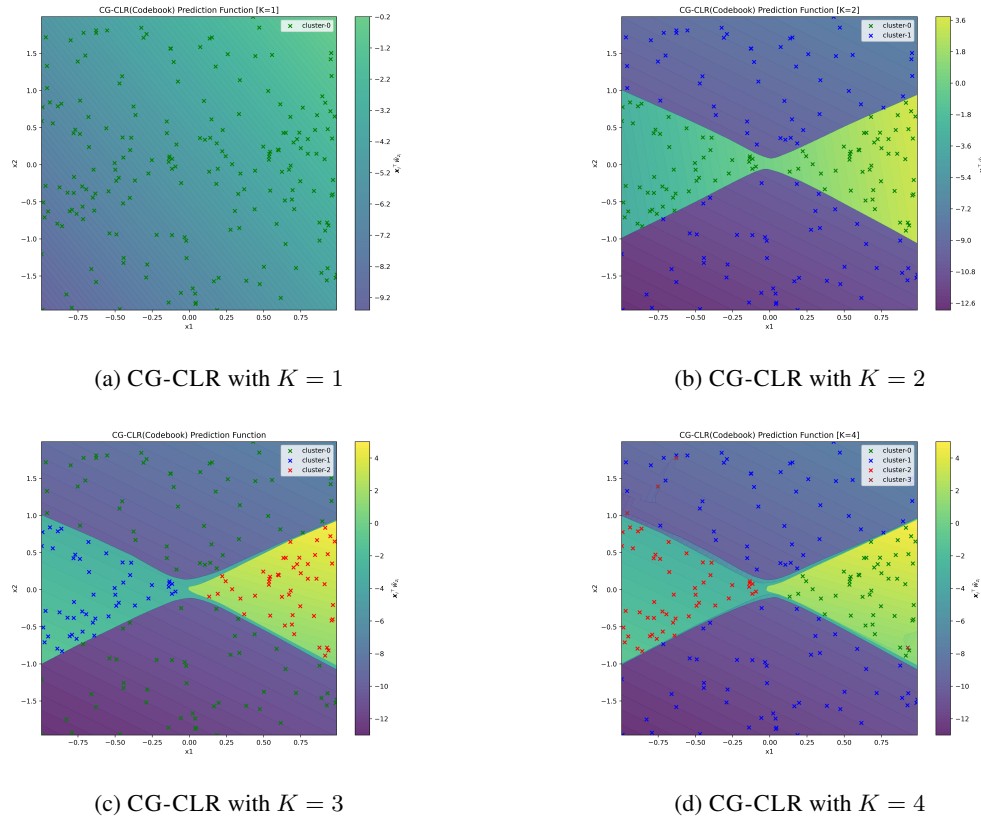

(a) CG-CLR with $K = 1$                        (b) CG-CLR with $K = 2$

(c) CG-CLR with $K = 3$                        (d) CG-CLR with $K = 4$

Figure 5: Function approximation and Clustering results (Ablation): (a) $K = 1$, (b) $K = 2$, (c) $K = 3$, (d) $K = 4$.

CG-CLR) necessarily faces ambiguity in cluster labels, but prediction can still remain accurate if the corresponding linear experts are close. The goal of this simulation is to test whether CG-CLR can (a) recover the cluster structure where it is identifiable from covariates, and (b) accurately reconstruct the component-wise linear mechanisms even under covariate overlap and observation noise.

**Data generating process.** We consider $K = 3$ latent regression components with mixture weights $\pi_1 = \frac{1}{6}$, $\pi_2 = \frac{1}{3}$, and $\pi_3 = \frac{1}{2}$, corresponding to unequal cluster sizes. First, a latent cluster index $z \in \{1, 2, 3\}$ is drawn according to $\{\pi_j\}_{j=1}^3$. Conditioned on $z = j$, the covariate vector $\boldsymbol{x} \in \mathbb{R}^{10}$ is drawn from a full-covariance Gaussian component

$$\boldsymbol{x} \mid (z = j) \sim \mathcal{N}(\boldsymbol{\mu}_j, \boldsymbol{\Sigma}_j), \qquad j = 1, 2, 3,$$

where the means $\boldsymbol{\mu}_j \in \mathbb{R}^{10}$ and covariance matrices $\boldsymbol{\Sigma}_j \in \mathbb{R}^{10 \times 10}$ are sampled once at random and then fixed. Each $\boldsymbol{\Sigma}_j$ is a dense, positive-definite covariance with moderate correlations between coordinates, obtained by drawing a random correlation matrix and rescaling it by random marginal variances.

**Gaussian mixture parameters.** The randomly generated mixture component parameters used in this experiment are

$$\boldsymbol{\mu}_1 = [-0.729, \ 0.776, \ 0.865, \ -0.109, \ -0.224, \ -0.485, \ 0.315, \ -0.015, \ 0.928, \ 0.602],$$

$$\boldsymbol{\mu}_2 = [-0.090, \ 0.602, \ -0.917, \ 0.539, \ -0.994, \ -0.414, \ 0.222, \ 0.826, \ -0.400, \ -0.503],$$

$$\boldsymbol{\mu}_3 = [0.333, \ 0.975, \ -0.063, \ -0.753, \ 0.832, \ 0.892, \ -0.445, \ 0.039, \ -0.691, \ -0.971].$$

The covariance matrices are:

$$\boldsymbol{\Sigma}_1 = \begin{bmatrix} 0.438 & 0.045 & 0.067 & 0.066 & -0.036 & 0.034 & -0.013 & -0.105 & -0.086 & -0.069 \\ 0.045 & 0.991 & 0.008 & 0.020 & -0.100 & 0.044 & -0.057 & -0.124 & 0.146 & 0.041 \\ 0.067 & 0.008 & 0.572 & 0.023 & 0.014 & 0.053 & -0.065 & -0.099 & 0.003 & -0.051 \\ 0.066 & 0.020 & 0.023 & 0.880 & -0.067 & 0.009 & 0.056 & 0.032 & -0.070 & 0.023 \\ -0.036 & -0.100 & 0.014 & -0.067 & 0.285 & 0.027 & -0.024 & 0.007 & -0.037 & -0.021 \\ 0.034 & 0.044 & 0.053 & 0.009 & 0.027 & 0.412 & -0.077 & -0.093 & 0.024 & -0.004 \\ -0.013 & -0.057 & -0.065 & 0.056 & -0.024 & -0.077 & 0.539 & 0.116 & -0.063 & -0.018 \\ -0.105 & -0.124 & -0.099 & 0.032 & 0.007 & -0.093 & 0.116 & 0.776 & -0.024 & 0.038 \\ -0.086 & 0.146 & 0.003 & -0.070 & -0.037 & 0.024 & -0.063 & -0.024 & 0.924 & 0.122 \\ -0.069 & 0.041 & -0.051 & 0.023 & -0.021 & -0.004 & -0.018 & 0.038 & 0.122 & 0.485 \end{bmatrix}.$$

$$\boldsymbol{\Sigma}_2 = \begin{bmatrix} 0.754 & 0.020 & -0.174 & 0.007 & 0.064 & -0.129 & 0.009 & -0.059 & -0.067 & 0.081 \\ 0.020 & 0.311 & -0.004 & -0.023 & 0.049 & 0.078 & -0.003 & 0.006 & 0.028 & -0.026 \\ -0.174 & -0.004 & 0.652 & -0.028 & -0.082 & 0.066 & -0.023 & 0.052 & 0.044 & -0.080 \\ 0.007 & -0.023 & -0.028 & 0.430 & -0.083 & -0.011 & 0.051 & 0.005 & -0.018 & 0.006 \\ 0.064 & 0.049 & -0.082 & -0.083 & 0.911 & 0.112 & 0.086 & -0.032 & -0.056 & 0.003 \\ -0.129 & 0.078 & 0.066 & -0.011 & 0.112 & 0.994 & 0.084 & 0.059 & 0.075 & -0.138 \\ 0.009 & -0.003 & -0.023 & 0.051 & 0.086 & 0.084 & 0.962 & -0.021 & -0.093 & 0.020 \\ -0.059 & 0.006 & 0.052 & 0.005 & -0.032 & 0.059 & -0.021 & 0.382 & 0.024 & -0.133 \\ -0.067 & 0.028 & 0.044 & -0.018 & -0.056 & 0.075 & -0.093 & 0.024 & 0.418 & -0.082 \\ 0.081 & -0.026 & -0.080 & 0.006 & 0.003 & -0.138 & 0.020 & -0.133 & -0.082 & 0.806 \end{bmatrix}.$$

$$\boldsymbol{\Sigma}_3 = \begin{bmatrix} 0.402 & -0.051 & -0.053 & -0.085 & -0.048 & 0.011 & 0.035 & -0.032 & -0.035 & -0.053 \\ -0.051 & 0.659 & 0.017 & 0.126 & -0.054 & -0.019 & -0.084 & -0.094 & 0.065 & -0.037 \\ -0.053 & 0.017 & 0.995 & 0.050 & 0.014 & 0.142 & 0.009 & 0.122 & -0.008 & 0.037 \\ -0.085 & 0.126 & 0.050 & 0.497 & 0.035 & -0.048 & -0.052 & -0.080 & 0.014 & 0.041 \\ -0.048 & -0.054 & 0.014 & 0.035 & 0.949 & -0.090 & 0.035 & 0.030 & -0.026 & 0.039 \\ 0.011 & -0.019 & 0.142 & -0.048 & -0.090 & 0.542 & 0.019 & 0.104 & 0.080 & -0.008 \\ 0.035 & -0.084 & 0.009 & -0.052 & 0.035 & 0.019 & 0.532 & 0.081 & -0.038 & 0.016 \\ -0.032 & -0.094 & 0.122 & -0.080 & 0.030 & 0.104 & 0.081 & 0.906 & -0.008 & -0.008 \\ -0.035 & 0.065 & -0.008 & 0.014 & -0.026 & 0.080 & -0.038 & -0.008 & 0.860 & -0.029 \\ -0.053 & -0.037 & 0.037 & 0.041 & 0.039 & -0.008 & 0.016 & -0.008 & -0.029 & 0.301 \end{bmatrix}.$$

Given the covariates and cluster index, the response is generated by a cluster-specific linear model with additive Gaussian noise:

$$y \mid (\boldsymbol{x}, z = j) = \boldsymbol{x}^\top \boldsymbol{w}_j + b_j + \varepsilon, \qquad \varepsilon \sim \mathcal{N}(0, 0.1^2).$$

The true regression coefficients $\boldsymbol{w}_j \in \mathbb{R}^{10}$ and intercepts $b_j$ are sampled at random and fixed to

$\boldsymbol{w}_1 = [-0.306, 0.809, -0.350, -0.283, 0.457, 0.893, -0.526, 0.887, -0.808, 0.163]$,
$b_1 = -0.755$,

$\boldsymbol{w}_2 = [-0.322, 0.657, -0.311, -0.080, -0.861, -0.307, -0.623, -0.658, -0.633, -0.718]$,
$b_2 = -0.129$,

$\boldsymbol{w}_3 = [-0.847, -0.747, 0.576, -0.134, 0.460, -0.537, -0.037, 0.322, -0.287, 0.299]$,
$b_3 = 0.109$.

From this Gaussian mixture model, we draw 600 training samples and 600 test samples, with cluster sizes proportional to the mixture weights $(\pi_1, \pi_2, \pi_3) = (1/6, 1/3, 1/2)$. Using these data, we train CG-CLR with $K = 3$ under the same proxy architecture and optimization settings as in the main simulation, and evaluate both the learned cluster assignments and the recovered codebook regressors.

**Cluster assignments in a noisy, overlapping regime.** Since $p = 10$, we visualize the learned structure via PCA. Figure 6 plots the first two principal components of the test covariates: colored

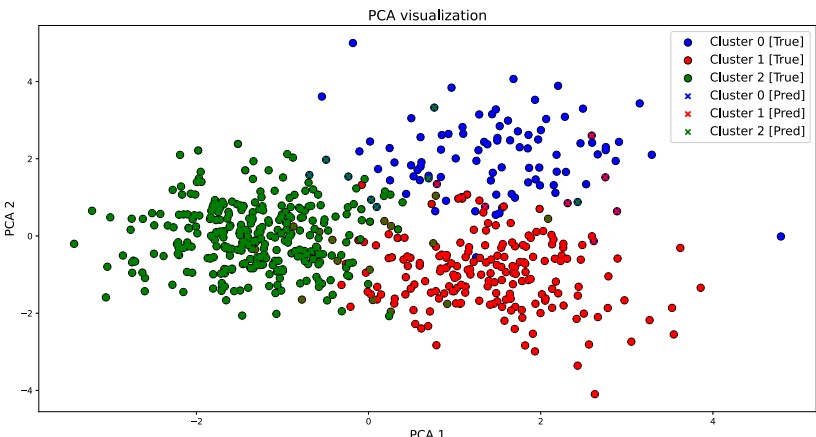

Figure 6: High-dimensional Gaussian mixture experiment ($p = 10$, $K = 3$). PCA projection of the test covariates, colored by the *true* Gaussian component (circles) and annotated with the *predicted* CG-CLR cluster (crosses). In regions where the components are well separated, the predicted clusters align closely with the ground truth; in heavily overlapping regions, assignments follow the local covariate geometry, as expected for a covariate routing rule.

circles indicate the true Gaussian component labels, while crosses indicate the predicted cluster assignments from CG-CLR.

Even though the Gaussian components substantially overlap in the PCA plane, CG-CLR correctly recovers many of the cluster assignments, especially in the better-separated regions. In areas where multiple components are nearly indistinguishable in terms of $\boldsymbol{x}$, the model naturally follows the local covariate trend and may assign some points to a nearby cluster, reflecting the intrinsic ambiguity of a purely covariate-based assignment.

**Prediction accuracy and function values.** To examine the effect of such ambiguous assignments on prediction quality, we compare the true denoised function values $f^{\star}(\boldsymbol{x}_i) = \boldsymbol{x}_i^{\top} \boldsymbol{w}_{z^*} + b_{z^*}$ with the codebook predictions $\tilde{y}_i = \mathbb{x}_i^{\top} \tilde{\mathbb{w}}_{z_i}$. Figure 7 shows both quantities over the same PCA projection; color encodes the function value.

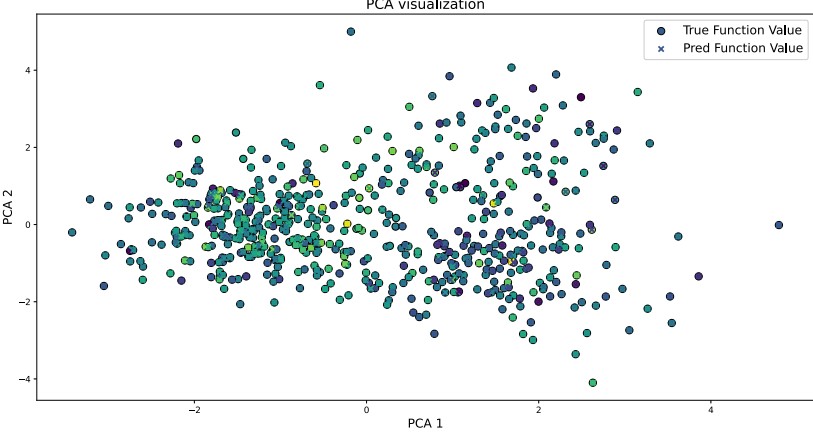

Figure 7: True denoised function values $f^{\star}(\boldsymbol{x}_i)$ (circles) and CG-CLR codebook predictions $\tilde{y}_i$ (crosses) on the PCA projection of the test set. Despite some mismatches in cluster assignments in overlapping regions, the predicted function values closely track the ground truth, indicating that many of these assignment errors occur where the underlying linear functions are themselves very similar.

The two fields are visually indistinguishable in most regions, confirming that the codebook regressors provide accurate predictions even under covariate overlap. This supports the intuition that when both the covariates and the underlying linear functions are nearly indistinguishable across clusters, misassignments are unavoidable but also largely harmless for prediction.

**Recovery of linear coefficients.** Finally, we compare the true regression weights $\boldsymbol{w}_j$ with the learned codebook weights $\tilde{\boldsymbol{w}}_j$ for each cluster. The recovered coefficients are

$$\tilde{\boldsymbol{w}}_1 = [-0.303,\ 0.787,\ -0.346,\ -0.283,\ 0.421,\ 0.854,\ -0.520,\ 0.868,\ -0.819,\ 0.149],$$

$$\tilde{\boldsymbol{w}}_2 = [-0.318,\ 0.645,\ -0.315,\ -0.084,\ -0.864,\ -0.310,\ -0.618,\ -0.638,\ -0.666,\ -0.720],$$

$$\tilde{\boldsymbol{w}}_3 = [-0.854,\ -0.748,\ 0.577,\ -0.152,\ 0.461,\ -0.531,\ -0.045,\ 0.319,\ -0.288,\ 0.278].$$

In all three clusters, the codebook closely matches the ground-truth coefficients: the relative $\ell_2$-errors $\|\boldsymbol{w}_j - \tilde{\boldsymbol{w}}_j\|_2 / \|\boldsymbol{w}_j\|_2$ are approximately 3.3% (cluster 1), 2.3% (cluster 2), and 2.0% (cluster 3), respectively. Thus, even in a high-dimensional Gaussian mixture with observation noise, CG-CLR is able to recover the underlying linear mechanisms to a high degree of accuracy.

# F  REAL DATASET EXPERIMENT

## F.1  DATASET DESCRIPTION

We evaluate our proposed CG-CLR framework on seven diverse real-world regression datasets, as detailed in Table 4. These datasets vary significantly in size and feature composition, allowing comprehensive evaluation across different predictive scenarios. The selected datasets span several domains, including materials science (CONDUCT), real estate (HOUSING), transportation (BIKE), energy (ELECTRICAL, PLANT), agriculture (WINE), and civil engineering (CONCRETE).

Table 4: Overview of the seven real-world regression datasets used in our experiments. Columns give the total number of **Samples** and the counts of **Categorical** and **Numerical** input features. Dataset abbreviations are: CONDUCT (superconductivity; Hamidieh, 2018), HOUSING (California housing; Kelley Pace & Barry, 1997), BIKE (bike-sharing; Fanaee-T, 2013), ELECTRICAL (electrical-grid stability; Arzamasov, 2018), PLANT (combined-cycle power plant; Tfekci & Kaya, 2014), WINE (wine quality; Cortez & Reis, 2009), and CONCRETE (concrete strength; Yeh, 1998).

| Dataset | Samples | Categorical | Numerical |
|---|---|---|---|
| CONDUCT | 21,263 | 0 | 81 |
| HOUSING | 20,640 | 0 | 8 |
| BIKE | 17,379 | 7 | 6 |
| ELECTRICAL | 10,000 | 0 | 12 |
| PLANT | 9,568 | 0 | 4 |
| WINE | 4,898 | 0 | 11 |
| CONCRETE | 1,030 | 0 | 8 |

## F.2  HYPERPARAMETER TUNING

Each method underwent rigorous tuning via grid search to optimize performance on validation folds within a nested cross-validation scheme. Tables 6 summarize the extensive hyperparameter search grids employed for the competing methods XGBoost (XGB), CatBoost (CAT), DC, CART, and S-IME$_d$. For LDT and PILOT, the maximum tree depth was set to $\lceil log_2 (N_{tr}/(10p + 10)) \rceil$.

(a) **XGB**

| Hyper-parameter | Range |
|---|---|
| n_estimators | $[1, 1000]$ |
| learning_rate | $\{0.001, 0.01, 0.1\}$ |
| max_depth | $\{3, 4, 5, 6, 7\}$ |
| subsample | $\{0.5, 0.75, 1.0\}$ |

(b) **CAT**

| Hyper-parameter | Range |
|---|---|
| n_estimators | $[1, 1000]$ |
| learning_rate | $\{0.001, 0.01, 0.1\}$ |
| max_depth | $\{3, 4, 5, 6, 7\}$ |
| subsample | $\{0.5, 0.75, 1.0\}$ |
| early_stopping_rounds | 100 |

(c) **DNN**

| Hyper-parameter | Range |
|---|---|
| hidden_nodes | $\{32, 64, 128, 256\}$ |
| hidden_layers | $\{2, 3, 4\}$ |
| dropout | $\{0.1, 0.2, 0.3\}$ |
| iterations | $[1, 10000]$ |
| early stopping patiences | 1000 |

(d) **DC**

| Hyper-parameter | Range |
|---|---|
| $\lambda$ | $\{0.01, 0.1, 1, 10, 100\}$ |
| $\rho$ | $\{0.001, 0.01, 0.1\}$ |
| iterations | $[1, 10000]$ |
| early stopping patiences | 1000 |

(e) **S-IME$_d$**

| Hyper-parameter | Range |
|---|---|
| $\beta$ | $\{0.1, 1, 10\}$ |
| $\gamma$ | $\{0.1, 1, 10\}$ |
| $\lambda$ | $\{0.1, 1, 10\}$ |
| iterations | $[1, 10000]$ |
| early stopping patiences | 1000 |

(f) **CART**

| Hyper-parameter | Range |
|---|---|
| max_depth | $[1, \lceil log_2 (N_{tr}/(10p + 10)) \rceil]$ |

Table 5: Hyper-parameter tuning grids for all methods.

## F.3 CG-CLR Hyperparameter Setting

For CG-CLR, several hyperparameters were uniformly set across all datasets, as shown in Table 6. We selected a relatively simple proxy network architecture with three hidden layers (64 neurons each), dropout regularization (rate 0.2), and used Adam optimizer with a learning rate of 0.001. The alignment weight ($\lambda$) was consistently set to 1, reflecting balanced importance between fitting and alignment. Training employed early stopping with patience of 1000 epochs to avoid overfitting.

Table 6: CG-CLR hyper-parameters kept fixed across all datasets.

| Hyper-parameter | Value |
|---|---|
| batch size | 256 |
| proxy network hidden layers | (64, 64, 64) |
| proxy network dropout | 0.2 |
| alignment weight $\lambda$ | 1 |
| learning rate | 0.001 |
| optimizer | Adam |
| iterations | 10000 |
| early stopping patiences | 1000 |

## F.4 Performance Results

Performance results presented in Table 7 systematically compare CG-CLR against both powerful black-box methods and various cluster-based linear regression alternatives. Results are organized into three coverage-based clusters (small, medium, and large) defined by the number of clusters ($K$). Notably, CG-CLR consistently achieves the lowest RMSE among all cluster-based models, and frequently performs comparably to or better than leading black-box models, emphasizing its robust generalization and interpretability across diverse real-world datasets.

Table 7: Test RMSE *mean $\pm$ std.* on seven regression datasets (CONDUCT, HOUSING, BIKE, ELECTRICAL, PLANT, WINE, CONCRETE; Hamidieh, 2018; Kelley Pace & Barry, 1997; Fanaee-T, 2013; Arzamasov, 2018; Tfekci & Kaya, 2014; Cortez & Reis, 2009; Yeh, 1998). **Model blocks.** (i) *Black-box baselines*: RF (Random Forest), XGB (XGBoost; Chen & Guestrin, 2016), CAT (CatBoost; Prokhorenkova et al., 2018), and DNN. (ii–iv) *Cluster-wise models* with three predetermined cluster budgets $K$: (*s*) "small" coverage $K \geq \lceil N_{tr}/(p+1) \rceil$, (*m*) "medium" coverage $K = \lfloor N_{tr}/(5p+5) \rfloor$, and (*l*) "large" coverage $K = \lfloor N_{tr}/(10p+10) \rfloor$. Symbols follow the literature: DC (Siahkamari et al., 2020), MLR* (mixed-linear regression (Pal et al., 2022) with post-hoc neighborhood assignment), CART, PILOT (Raymaekers et al., 2024), LDT (Ahmed et al., 2018), S-IME$_d$ (Ismail et al., 2023), and our proposed CG-CLR variants (CG-CLR$_s$, CG-CLR$_m$, CG-CLR$_l$, and CG-CLR$_{xl}$ where CG-CLR$_{xl}$ sets $K = \lfloor N_{tr}/(20p+20) \rfloor$). Lower RMSE is better; the best score for each block is highlighted in **bold**.

| Model | CONDUCT | HOUSING | BIKE | ELECTRICAL | PLANT | WINE | CONCRETE |
|---|---|---|---|---|---|---|---|
| RF | $9.81 \pm 0.16$ | $0.511 \pm 0.008$ | $56.45 \pm 1.19$ | $0.0121 \pm 0.0002$ | $3.466 \pm 0.099$ | $\mathbf{0.627 \pm 0.025}$ | $5.381 \pm 0.418$ |
| XGB | $9.79 \pm 0.13$ | $0.457 \pm 0.008$ | $45.89 \pm 0.81$ | $0.0080 \pm 0.0001$ | $3.200 \pm 0.093$ | $0.634 \pm 0.025$ | $4.507 \pm 0.340$ |
| CAT | $\mathbf{9.69 \pm 0.14}$ | $\mathbf{0.443 \pm 0.007}$ | $44.99 \pm 0.64$ | $\mathbf{0.0068 \pm 0.0001}$ | $\mathbf{3.162 \pm 0.116}$ | $0.632 \pm 0.024$ | $\mathbf{4.225 \pm 0.360}$ |
| DNN | $10.44 \pm 0.20$ | $0.505 \pm 0.007$ | $\mathbf{42.23 \pm 1.01}$ | $0.0070 \pm 0.0002$ | $3.785 \pm 0.100$ | $0.670 \pm 0.024$ | $4.934 \pm 0.372$ |
| DC | $11.66 \pm 0.59$ | $0.597 \pm 0.024$ | $55.36 \pm 0.73$ | $0.0119 \pm 0.0005$ | $3.746 \pm 0.106$ | $\mathbf{0.651 \pm 0.028}$ | $5.654 \pm 0.427$ |
| MLR$^*_s$ | $34.08 \pm 0.19$ | $1.184 \pm 0.010$ | $178.26 \pm 1.44$ | $0.0348 \pm 0.0005$ | $13.800 \pm 0.170$ | $0.835 \pm 0.018$ | $15.263 \pm 0.445$ |
| S-IME$_{d,s}$ | $12.44 \pm 0.26$ | $0.591 \pm 0.034$ | $54.27 \pm 1.68$ | $0.0094 \pm 0.0007$ | $4.276 \pm 0.091$ | $0.705 \pm 0.023$ | $7.021 \pm 0.436$ |
| **CG-CLR$_s$** | $10.49 \pm 0.11$ | $\mathbf{0.494 \pm 0.009}$ | $40.66 \pm 1.04$ | $\mathbf{0.0061 \pm 0.0002}$ | $3.631 \pm 0.111$ | $0.669 \pm 0.028$ | $\mathbf{4.919 \pm 0.382}$ |
| MLR$^*_m$ | $34.06 \pm 0.20$ | $1.171 \pm 0.009$ | $180.46 \pm 1.47$ | $0.0363 \pm 0.0005$ | $16.393 \pm 0.200$ | $0.868 \pm 0.016$ | $15.248 \pm 0.450$ |
| S-IME$_{d,m}$ | $12.65 \pm 0.26$ | $0.578 \pm 0.068$ | $55.82 \pm 2.33$ | $0.0095 \pm 0.0005$ | $4.172 \pm 0.069$ | $0.703 \pm 0.023$ | $8.553 \pm 0.378$) |
| **CG-CLR$_m$** | $10.50 \pm 0.15$ | $0.494 \pm 0.011$ | $40.85 \pm 1.04$ | $0.0062 \pm 0.0002$ | $3.627 \pm 0.113$ | $0.669 \pm 0.032$ | $5.066 \pm 0.435$ |
| MLR$^*_l$ | $33.86 \pm 0.19$ | $1.164 \pm 0.011$ | $178.24 \pm 1.43$ | $0.0348 \pm 0.0005$ | $13.797 \pm 0.170$ | $0.835 \pm 0.018$ | $15.243 \pm 0.441$ |
| CART | $16.71 \pm 0.33$ | $0.652 \pm 0.014$ | $121.32 \pm 2.09$ | $0.0212 \pm 0.0002$ | $4.261 \pm 0.093$ | $0.745 \pm 0.024$ | $9.569 \pm 0.541$ |
| PILOT$_l$ | $15.52 \pm 0.46$ | $0.821 \pm 0.011$ | $165.87 \pm 1.30$ | $0.0332 \pm 0.0005$ | $4.774 \pm 0.134$ | $0.756 \pm 0.022$ | $14.530 \pm 0.629$ |
| LDT$_l$ | $14.49 \pm 2.41$ | $0.673 \pm 0.149$ | $59.73 \pm 0.62$ | $0.0151 \pm 0.0004$ | $4.119 \pm 0.091$ | $0.708 \pm 0.021$ | $6.264 \pm 0.329$ |
| S-IME$_{d,l}$ | $12.86 \pm 0.38$ | $0.565 \pm 0.011$ | $57.07 \pm 2.00$ | $0.0095 \pm 0.0006$ | $4.221 \pm 0.082$ | $0.704 \pm 0.023$ | $9.182 \pm 0.704$ |
| **CG-CLR$_l$** | $10.56 \pm 0.13$ | $\mathbf{0.491 \pm 0.012}$ | $41.24 \pm 1.00$ | $0.0063 \pm 0.0002$ | $3.624 \pm 0.109$ | $0.664 \pm 0.025$ | $\mathbf{5.365 \pm 0.367}$ |
| **CG-CLR$_{xl}$** | $10.71 \pm 0.13$ | $0.495 \pm 0.009$ | $42.75 \pm 1.04$ | $0.0066 \pm 0.0002$ | $3.632 \pm 0.114$ | $0.671 \pm 0.028$ | $6.311 \pm 0.443$ |

## F.5 Comparison with GAM Baselines

Generalized Additive Models (GAMs) represent an orthogonal approach to balancing accuracy and simplicity: instead of clustering covariates into local linear experts, they rely on additive modeling to achieve strong predictive performance. Among them, Node-GA$^2$M (Chang et al., 2022) is a

state-of-the-art GAM variant that combines tree-based node splitting with additive terms, and thus serves as a competitive baseline distinct from cluster-oriented methods.

To ensure a fair comparison, we evaluate CG-CLR against Node-GA$^2$M across all seven datasets. For each dataset, we report: (i) the minimum number of clusters $K$ at which CG-CLR ties with Node-GA$^2$M (their 95% CIs overlap), and (ii) the minimum $K$ at which CG-CLR strictly outperforms Node-GA$^2$M (its 95% CI lies entirely below Node-GA$^2$M's). If CG-CLR neither ties nor wins at any tested $K$, we denote this by "-".

Table 8: Comparison against Node-GA$^2$M: RMSE (95% CI) and the minimum $K$ required for CG-CLR to tie or outperform.

| Dataset | Node-GA$^2$M: RMSE | Tie: $K$, RMSE | Win: $K$, RMSE |
|---|---|---|---|
| CONDUCT | [12.558, 12.782] | - | 3, [11.738, 11.911] |
| HOUSING | [0.495, 0.503] | 8, [0.502, 0.512] | - |
| BIKE | [53.70, 55.16] | - | 5, [51.05, 52.97] |
| ELECTRICAL | [0.009, 0.009] | 4, [0.009, 0.010] | 5, [0.008, 0.008] |
| PLANT | [3.989, 4.073] | 2, [3.982, 4.059] | 3, [3.808, 3.908] |
| WINE | [0.677, 0.695] | 24, [0.688, 0.709] | 24, [0.652, 0.676] |
| CONCRETE | [4.634, 5.014] | 15, [5.010, 5.449] | - |

The results reveal two distinct regimes. On HOUSING and CONCRETE, CG-CLR already matches Node-GA$^2$M at modest cluster counts, demonstrating comparable accuracy to a strong additive model. On the remaining five datasets (CONDUCT, BIKE, ELECTRICAL, PLANT, and WINE), CG-CLR surpasses Node-GA$^2$M with relatively small $K$, highlighting its ability to reach or exceed state-of-the-art additive modeling performance while controlling complexity through clustering.

### F.6 CASE STUDIES UNDER COMPETITIVE $K$

To better illustrate the interpretability benefits of CG-CLR when it achieves accuracy on par with strong additive models, we provide case studies for the two datasets where CG-CLR matches Node-GA$^2$M at relatively small $K$: HOUSING ($K=8$) and CONDUCT ($K=3$). These examples highlight how competitive accuracy can be attained while still surfacing domain-specific, interpretable cluster patterns.

**HOUSING** ($K = 8$). Tables 9 and 10 report the clusterwise regression coefficients and feature means, respectively. Cluster 1 corresponds to a high-occupancy rental market, with property values driven by the number of bedrooms. Cluster 4 represents affluent suburban districts, where high income and room counts dominate, while additional bedrooms are penalized. Cluster 6 captures peripheral agricultural towns with low income but upward sensitivity to income and population changes. These patterns demonstrate that at $K = 8$, CG-CLR not only ties the performance of Node-GA$^2$M but also yields region-specific linear models that are naturally interpretable.

Table 9: Clusterwise Linear Coefficients Table (HOUSING).

| Cluster idx | MedInc | HouseAge | AveRooms | AveBedrms | Population | AveOccup | Latitude | Longitude |
|---|---|---|---|---|---|---|---|---|
| 0 | -0.05 | 0.017 | -0.06 | -0.05 | -22e-6 | 0.02 | -0.11 | 0.01 |
| 1 | -0.06 | -0.008 | 0.08 | 0.24 | -149e-6 | -0.00 | -0.01 | -0.03 |
| 2 | -0.03 | 0.016 | -0.06 | -0.49 | -150e-6 | -0.04 | -0.01 | 0.10 |
| 3 | 0.14 | -0.011 | 0.11 | -0.32 | -49e-6 | 0.04 | 0.02 | -0.10 |
| 4 | 0.18 | 0.010 | 0.12 | -0.68 | -75e-6 | -0.03 | -0.19 | 0.13 |
| 5 | -0.04 | 0.008 | -0.05 | -0.34 | -50e-6 | -0.03 | 0.08 | -0.10 |
| 6 | 0.18 | 0.001 | 0.04 | -0.07 | 158e-6 | 0.00 | -0.12 | -0.14 |
| 7 | 0.15 | 0.022 | 0.07 | 0.00 | -166e-6 | -0.02 | 0.15 | -0.13 |

**CONDUCT** ($K = 3$). Table 11 presents the estimated coefficients as $K$ increases from 1 to 4. Distinct patterns emerge at small $K$. At $K = 2$, an age-based split appears, separating young and old specimens, with the "Coarse Aggregate" effect flipping sign across groups. At $K = 3$, a clear "Superplasticizer effect" arises, isolating a cluster with exceptionally high superplasticizer weight and confirming its strong interaction with water and cement. At $K = 4$, long-term curing benefits are exposed, as aged specimens form a distinct group with delayed strength gains. These observations show that small $K$ values already uncover meaningful material science insights while achieving accuracy competitive with Node-GA$^2$M.

Table 10: Clusterwise Feature Mean Value Table (HOUSING).

| Cluster idx | MedInc | HouseAge | AveRooms | AveBedrms | Population | AveOccup | Latitude | Longitude |
|---|---|---|---|---|---|---|---|---|
| 0 | 4.17 | 29.18 | 6.08 | 1.21 | 1351.26 | 2.65 | 35.74 | -119.845 |
| 1 | 3.80 | 27.70 | 5.11 | 1.05 | 1571.03 | 3.69 | 35.62 | -119.833 |
| 2 | 3.23 | 26.50 | 4.96 | 1.08 | 1485.59 | 2.86 | 35.91 | -119.829 |
| 3 | 4.39 | 28.28 | 5.34 | 1.05 | 1582.74 | 2.95 | 34.87 | -119.181 |
| 4 | 4.97 | 29.46 | 5.81 | 1.09 | 1325.35 | 2.67 | 35.59 | -119.399 |
| 5 | 4.17 | 30.01 | 5.53 | 1.10 | 1461.98 | 2.90 | 35.77 | -120.057 |
| 6 | 2.64 | 28.06 | 5.23 | 1.11 | 1357.68 | 3.22 | 36.14 | -119.529 |
| 7 | 4.45 | 29.92 | 5.40 | 1.07 | 1369.02 | 3.05 | 35.32 | -119.487 |

Table 11: Qualitative interpretability changes as $K$ increases (CONDUCT).

| Cluster idx ($K$) | Cement | Blast Furnace Slag | Fly Ash | Water | Superplasticizer | Coarse Aggregate | Fine Aggregate | Age |
|---|---|---|---|---|---|---|---|---|
| 1 (1) | 0.100 | 0.093 | 0.069 | -0.230 | 0.253 | 0.003 | -0.005 | 0.118 |
| 1 (2) | 0.105 | 0.068 | 0.008 | -0.175 | 0.884 | -0.035 | -0.029 | 0.093 |
| 2 (2) | 0.054 | 0.073 | 0.024 | -0.148 | 0.616 | 0.098 | 0.009 | 0.108 |
| 1 (3) | 0.033 | 0.026 | -0.094 | -0.221 | -0.234 | -0.048 | -0.086 | 0.035 |
| 2 (3) | 0.029 | -0.005 | 0.058 | -0.155 | 0.743 | -0.068 | -0.054 | 0.046 |
| 3 (3) | 0.031 | 0.060 | 0.016 | -0.241 | 1.685 | -0.000 | -0.067 | 0.096 |
| 1 (4) | 0.033 | -0.018 | -0.064 | 0.047 | 0.875 | -0.074 | 0.034 | 0.075 |
| 2 (4) | 0.047 | -0.024 | 0.047 | -0.252 | 0.476 | 0.033 | -0.066 | 0.021 |
| 3 (4) | 0.073 | 0.079 | -0.049 | -0.237 | 0.304 | -0.034 | -0.036 | 0.133 |
| 4 (4) | 0.031 | 0.040 | 0.051 | 0.070 | 0.790 | -0.460 | -0.072 | 0.041 |

These case studies show that under competitive cluster counts, CG-CLR not only achieves parity with advanced additive models but also yields interpretable, domain-relevant partitions that provide insights into housing markets and materials science.

### F.7 COMPUTATIONAL COST ANALYSIS

To complement the accuracy comparisons, we analyze computational efficiency in terms of wall-clock time and peak memory. We compare our CG-CLR framework against two classical cluster-oriented baselines: (i) **Dense MoE**—a soft-gated Mixture-of-Experts with linear experts (Jacobs et al., 1991), and (ii) **EM-CLR\***—an EM-based clusterwise linear regression (DeSarbo & Cron, 1988) with nearest-centroid routing at test time. All methods use the same expert budget $K = N/(10p+10)$ and identical tuning protocols (20 independent runs per dataset).

**Baseline Performance.** Table 12 reports test RMSE with 95% confidence intervals across seven datasets. CG-CLR consistently achieves the lowest RMSE on every dataset, while maintaining clear cluster-level coefficients (like EM-CLR\*) and markedly lighter computational cost than Dense MoE.

Table 12: Test RMSE with 95% confidence intervals across seven datasets.

| Model | CONDUCT | HOUSING | BIKE | ELECTRICAL | PLANT | WINE | CONCRETE |
|---|---|---|---|---|---|---|---|
| Dense MoE | [12.56, 13.00] | [0.548, 0.561] | [58.91, 67.03] | [0.010, 0.010] | [4.388, 4.535] | [0.694, 0.717] | [7.454, 8.313] |
| EM-CLR\* | [33.85, 34.04] | [1.074, 1.082] | [178.32, 179.69] | [0.035, 0.035] | [13.848, 14.002] | [0.844, 0.861] | [15.356, 15.860] |
| **CG-CLR** | **[10.50, 10.62]** | **[0.485, 0.497]** | **[40.77, 41.71]** | **[0.006, 0.006]** | **[3.573, 3.675]** | **[0.652, 0.676]** | **[5.193, 5.537]** |

**Computational Analysis.** We further benchmark CG-CLR against Dense MoE and EM-CLR\* on three representative datasets (CONDUCT, ELECTRICAL, CONCRETE) covering a 20× span in sample size. Wall-clock time (sec/100 epochs) and peak memory (MiB) were measured over 20 independent runs and reported as 95% confidence intervals. Regressor budgets $K \in \{1, 4, 16, 64, 256, 1024\}$ probe scaling with respect to the number of regressors.

Overall, the results demonstrate three main trends. First, CG-CLR maintains a tight efficiency bound: across all expert budgets, its wall-clock time and memory usage remain within about 1.7× those of EM-CLR\*, while consistently achieving superior accuracy. Second, CG-CLR scales much more gracefully with the number of experts compared to Dense MoE. Because the forward pass uses $O(K)$ argmax routing while the backward pass is $O(1)$ in $K$, the training time increases by only about 15% from $K = 1$ to $K = 256$. By contrast, Dense MoE incurs more than a twofold increase due to its $O(K)$ gradient cost, and this is also reflected in its peak memory consumption. Finally, with respect to dataset size, wall-clock time for CG-CLR grows proportionally to the input scale: moving from 1,030 samples (CONCRETE) to 21,263 samples (CONDUCT) increases runtime by about 10×, which matches the growth in per-sample compute. Memory usage, however, grows only modestly with the number of samples and features, and remains essentially flat with respect to $K$.

Table 13: Scaling w.r.t $K$ — Wall-clock time (sec / 100 epochs) with 95% confidence intervals.

| Dataset / Model | K=1 | K=4 | K=16 | K=64 | K=256 | K=1024 |
|---|---|---|---|---|---|---|
| **CONDUCT** | | | | | | |
| Dense MoE | [11.0, 11.1] | [11.2, 11.3] | [11.4, 12.0] | [15.1, 16.4] | [27.6, 28.9] | [75.1, 77.2] |
| EM-CLR* | [11.1, 11.3] | [10.8, 11.2] | [11.1, 11.3] | [10.6, 11.3] | [11.0, 11.4] | [11.2, 11.5] |
| **CG-CLR** | [11.5, 11.8] | [11.0, 11.3] | [11.3, 11.6] | [11.5, 11.7] | [11.9, 12.1] | [12.5, 12.9] |
| **ELECTRICAL** | | | | | | |
| Dense MoE | [5.4, 5.6] | [5.4, 5.5] | [5.9, 6.3] | [9.7, 10.0] | [19.7, 23.3] | [53.7, 71.4] |
| EM-CLR* | [5.1, 5.3] | [5.0, 5.2] | [5.2, 5.3] | [5.2, 5.3] | [5.2, 5.3] | [5.4, 5.6] |
| **CG-CLR** | [5.5, 5.6] | [5.5, 5.6] | [5.4, 5.6] | [5.3, 5.5] | [5.4, 5.5] | [6.2, 6.4] |
| **CONCRETE** | | | | | | |
| Dense MoE | [1.0, 1.1] | [0.7, 1.0] | [2.0, 2.0] | [4.4, 4.8] | [13.4, 17.2] | [49.7, 68.2] |
| EM-CLR* | [0.7, 0.7] | [0.7, 0.7] | [0.7, 0.7] | [0.6, 0.7] | [0.7, 0.7] | [0.7, 0.7] |
| **CG-CLR** | [0.7, 0.8] | [0.7, 0.8] | [0.7, 0.9] | [0.8, 0.9] | [0.8, 1.0] | [0.8, 0.9] |

Table 14: Scaling w.r.t $K$ — Peak memory (MiB) with 95% confidence intervals.

| Dataset / Model | K=1 | K=4 | K=16 | K=64 | K=256 | K=1024 |
|---|---|---|---|---|---|---|
| **CONDUCT** | | | | | | |
| Dense MoE | [50.0, 50.0] | [50.0, 50.0] | [50.0, 50.0] | [50.3, 50.4] | [52.8, 52.8] | [71.0, 71.1] |
| EM-CLR* | [48.2, 48.2] | [48.2, 48.2] | [48.2, 48.2] | [48.2, 48.2] | [48.2, 48.2] | [48.2, 48.2] |
| **CG-CLR** | [50.0, 50.0] | [50.0, 50.0] | [50.0, 50.0] | [50.0, 50.0] | [50.0, 50.0] | [50.0, 50.0] |
| **ELECTRICAL** | | | | | | |
| Dense MoE | [2.4, 2.4] | [2.4, 2.4] | [2.5, 2.6] | [3.0, 3.1] | [6.8, 7.0] | [25.8, 26.0] |
| EM-CLR* | [0.5, 0.5] | [0.5, 0.5] | [0.5, 0.5] | [0.5, 0.5] | [0.5, 0.5] | [0.5, 0.5] |
| **CG-CLR** | [2.3, 2.3] | [2.3, 2.3] | [2.3, 2.3] | [2.3, 2.3] | [2.3, 2.3] | [2.3, 2.3] |
| **CONCRETE** | | | | | | |
| Dense MoE | [0.6, 0.6] | [0.6, 0.6] | [0.8, 0.8] | [1.9, 2.0] | [6.8, 6.9] | [25.9, 26.0] |
| EM-CLR* | [0.3, 0.3] | [0.3, 0.3] | [0.3, 0.3] | [0.3, 0.3] | [0.3, 0.3] | [0.3, 0.3] |
| **CG-CLR** | [0.4, 0.5] | [0.5, 0.5] | [0.5, 0.5] | [0.5, 0.5] | [0.5, 0.5] | [0.5, 0.5] |

### F.8 HYPERPARAMETER SENSITIVITY

We conducted a sensitivity analysis on the alignment hyperparameter $\lambda$ using CG-CLR with large cluster coverage ($K = \lfloor N_{tr}/(10p + 10) \rfloor$). Results reported in Table 15 illustrate stable performance across varying values of $\lambda$, with optimal or near-optimal performance consistently observed at $\lambda = 1$. This robustness confirms the practicality of the default choice of $\lambda = 1$ and highlights the stability of CG-CLR's predictive accuracy and simplicity balance. Moreover, the observed stability under moderate deviations of $\lambda$ is consistent with our theoretical analysis, which requires sufficiently small $\lambda < 1/(\kappa-1)$ to ensure valid expert assignment.

Table 15: Test RMSE with 95% confidence intervals across seven datasets (CONDUCT, HOUSING, BIKE, ELECTRICAL, PLANT, WINE, CONCRETE; Hamidieh, 2018; Kelley Pace & Barry, 1997; Fanaee-T, 2013; Arzamasov, 2018; Tfekci & Kaya, 2014; Cortez & Reis, 2009; Yeh, 1998). Sensitivity analysis w.r.t $\lambda$ under $K = \lfloor N_{tr}/(10p + 10) \rfloor$.

| Dataset | $\lambda = 0$ | $\lambda = 1$ | $\lambda = 2$ | $\lambda = 3$ |
|---|---|---|---|---|
| CONDUCT | [10.45, 10.60] | [10.50, 10.62] | [10.50, 10.63] | [10.59, 10.73] |
| HOUSING | [0.492, 0.500] | [0.485, 0.497] | [0.491, 0.501] | [0.490, 0.500] |
| BIKE | [41.07, 41.93] | [40.77, 41.71] | [41.23, 42.27] | [41.11, 41.79] |
| ELECTRICAL | [0.006, 0.006] | [0.006, 0.006] | [0.006, 0.006] | [0.006, 0.006] |
| PLANT | [3.583, 3.683] | [3.573, 3.675] | [3.588, 3.682] | [3.568, 3.674] |
| WINE | [0.652, 0.676] | [0.652, 0.676] | [0.654, 0.678] | [0.657, 0.687] |
| CONCRETE | [5.301, 5.731] | [5.193, 5.537] | [5.501, 5.851] | [5.546, 5.908] |

To examine how the stability of learned coefficients varies with the number of clusters $K$, we conduct an empirical study on the CONCRETE dataset in two complementary settings:

**(i) Random Initializations.** For a fixed training split, CG-CLR was trained with $T = 10$ different random seeds. We computed the standard deviation of each assigned coefficient across runs and averaged over all data points, using the following definition:

$$\sigma^j = \frac{1}{N} \sum_{i=1}^{N} \sqrt{\sum_{t=1}^{T} \frac{\left(\tilde{\mathbb{w}}_{z_i}^{j,(t)} - \bar{\mathbb{w}}_i^j\right)^2}{T - 1}}, \quad \left(\text{where } \bar{\mathbb{w}}_i^j = \frac{1}{T} \sum_{t=1}^{T} \tilde{\mathbb{w}}_{z_i}^{j,(t)}\right).$$

This quantifies the variability of learned coefficients under different random initializations.

**(ii) Data Splits (10-fold).** We further trained CG-CLR with 10-fold splits, each time fitting on 90% of the data and evaluating on the full dataset. For the 10-fold splits, the same formula was applied across models trained on each fold. This procedure simulates the effect of sampling variation and reflects robustness in practical deployment.

Table 16 reports the average standard deviations (STD) of coefficients across both settings. For $K = 1$, the coefficient variance is naturally low, as expected for a global linear model. As $K$ increases, the variance remains moderate, indicating that CG-CLR yields stable coefficient estimates both within a fixed dataset and across data splits.

Table 16: Average coefficient variability (STD) across random initializations and 10-fold splits on the CONCRETE dataset.

| K | Cement | Blast Slag | Fly Ash | Water | Superplasticizer | Coarse Agg. | Fine Agg. | Age |
|---|--------|-----------|---------|-------|------------------|-------------|-----------|-----|
| **Random Init. (STD)** | | | | | | | | |
| 1 | 0.034 | 0.038 | 0.050 | 0.122 | 0.186 | 0.029 | 0.038 | 0.002 |
| 2 | 0.027 | 0.030 | 0.042 | 0.153 | 0.357 | 0.025 | 0.027 | 0.048 |
| 3 | 0.019 | 0.037 | 0.043 | 0.094 | 0.460 | 0.020 | 0.014 | 0.048 |
| 4 | 0.022 | 0.034 | 0.051 | 0.083 | 0.228 | 0.035 | 0.020 | 0.053 |
| 5 | 0.024 | 0.032 | 0.052 | 0.096 | 0.366 | 0.030 | 0.025 | 0.057 |
| 6 | 0.017 | 0.034 | 0.052 | 0.094 | 0.330 | 0.035 | 0.026 | 0.040 |
| 7 | 0.023 | 0.033 | 0.059 | 0.082 | 0.337 | 0.037 | 0.024 | 0.037 |
| 8 | 0.018 | 0.035 | 0.052 | 0.098 | 0.411 | 0.039 | 0.026 | 0.034 |
| 9 | 0.023 | 0.034 | 0.056 | 0.115 | 0.336 | 0.042 | 0.030 | 0.034 |
| 10 | 0.019 | 0.037 | 0.060 | 0.123 | 0.390 | 0.040 | 0.027 | 0.032 |
| **10-fold Splits (STD)** | | | | | | | | |
| 1 | 0.016 | 0.019 | 0.029 | 0.088 | 0.350 | 0.018 | 0.021 | 0.010 |
| 2 | 0.036 | 0.037 | 0.047 | 0.117 | 0.388 | 0.037 | 0.034 | 0.092 |
| 3 | 0.020 | 0.038 | 0.053 | 0.109 | 0.396 | 0.031 | 0.021 | 0.060 |
| 4 | 0.018 | 0.027 | 0.056 | 0.079 | 0.343 | 0.023 | 0.024 | 0.068 |
| 5 | 0.018 | 0.032 | 0.058 | 0.109 | 0.396 | 0.026 | 0.024 | 0.038 |
| 6 | 0.020 | 0.037 | 0.056 | 0.113 | 0.405 | 0.038 | 0.023 | 0.042 |
| 7 | 0.019 | 0.032 | 0.055 | 0.107 | 0.405 | 0.041 | 0.026 | 0.037 |
| 8 | 0.020 | 0.040 | 0.056 | 0.109 | 0.370 | 0.041 | 0.028 | 0.036 |
| 9 | 0.022 | 0.032 | 0.058 | 0.105 | 0.358 | 0.039 | 0.029 | 0.033 |
| 10 | 0.020 | 0.037 | 0.057 | 0.103 | 0.366 | 0.045 | 0.027 | 0.035 |

We also examined the **average coefficient magnitudes** (absolute values) to contextualize the scale of learned parameters. Table 17 reports results under both random initialization and 10-fold splits, alongside input feature statistics for reference. Across both settings, magnitudes remain stable as $K$ increases, supporting the robustness of CG-CLR against coefficient variability.

Overall, these results demonstrate that CG-CLR maintains stable coefficient magnitudes and moderate variance even as the number of clusters $K$ increases, supporting the method's robustness and practical reliability.

## F.9 EMPIRICAL STUDY OF ASSUMPTION 1 AND ASSUMPTION 2

To assess whether the regularity conditions in Assumptions 1–2 are reasonable in our experimental regime, we conduct an empirical diagnostic study on the CONCRETE dataset. We track the Jacobian of the proxy network (Assumption 1) and the curvature of the alignment loss (Assumption 2) throughout training.

Table 17: Average coefficient magnitudes (absolute values) for the CONCRETE dataset.

| K | Cement | Blast Slag | Fly Ash | Water | Superplasticizer | Coarse Agg. | Fine Agg. | Age |
|---|--------|-----------|---------|-------|------------------|-------------|-----------|-----|
| **Random Init. (Absolute)** | | | | | | | | |
| 1 | 0.077 | 0.055 | 0.053 | 0.262 | 0.408 | 0.039 | 0.045 | 0.110 |
| 2 | 0.076 | 0.059 | 0.045 | 0.228 | 0.517 | 0.026 | 0.035 | 0.160 |
| 3 | 0.067 | 0.050 | 0.041 | 0.274 | 0.433 | 0.039 | 0.040 | 0.142 |
| 4 | 0.068 | 0.049 | 0.047 | 0.271 | 0.473 | 0.041 | 0.040 | 0.110 |
| 5 | 0.063 | 0.048 | 0.057 | 0.276 | 0.540 | 0.041 | 0.043 | 0.096 |
| 6 | 0.060 | 0.046 | 0.064 | 0.250 | 0.526 | 0.038 | 0.041 | 0.102 |
| 7 | 0.061 | 0.047 | 0.065 | 0.259 | 0.478 | 0.044 | 0.044 | 0.099 |
| 8 | 0.059 | 0.045 | 0.066 | 0.250 | 0.589 | 0.047 | 0.039 | 0.090 |
| 9 | 0.052 | 0.046 | 0.070 | 0.237 | 0.595 | 0.041 | 0.042 | 0.090 |
| 10 | 0.053 | 0.046 | 0.065 | 0.220 | 0.630 | 0.049 | 0.041 | 0.080 |
| **10-fold Splits (Absolute)** | | | | | | | | |
| 1 | 0.074 | 0.051 | 0.042 | 0.272 | 0.409 | 0.028 | 0.036 | 0.108 |
| 2 | 0.071 | 0.053 | 0.037 | 0.265 | 0.418 | 0.031 | 0.038 | 0.157 |
| 3 | 0.071 | 0.054 | 0.049 | 0.266 | 0.395 | 0.028 | 0.035 | 0.121 |
| 4 | 0.071 | 0.055 | 0.062 | 0.247 | 0.399 | 0.036 | 0.038 | 0.126 |
| 5 | 0.067 | 0.053 | 0.063 | 0.257 | 0.467 | 0.034 | 0.039 | 0.104 |
| 6 | 0.060 | 0.050 | 0.070 | 0.257 | 0.534 | 0.039 | 0.047 | 0.095 |
| 7 | 0.061 | 0.048 | 0.063 | 0.272 | 0.533 | 0.043 | 0.043 | 0.102 |
| 8 | 0.060 | 0.047 | 0.064 | 0.258 | 0.523 | 0.046 | 0.043 | 0.103 |
| 9 | 0.056 | 0.044 | 0.066 | 0.239 | 0.572 | 0.045 | 0.042 | 0.092 |
| 10 | 0.054 | 0.046 | 0.066 | 0.237 | 0.608 | 0.050 | 0.044 | 0.093 |

**Assumption 1: Jacobian lower bound and Lipschitz constant.** Every 10 epochs, we sample the proxy network Jacobian with respect to its parameters at all training covariates $\{\boldsymbol{x}_i\}_{i=1}^{N_{\text{tr}}}$. Specifically, for each input $\boldsymbol{x}_i$ and current parameter vector $\boldsymbol{\phi}^{(t)}$, we form the Jacobian

$$J_{\boldsymbol{\phi}^{(t)}}(\boldsymbol{x}_i) = \frac{\partial W_{\boldsymbol{\phi}^{(t)}}(\boldsymbol{x}_i)}{\partial \boldsymbol{\phi}^{(t)}} \in \mathbb{R}^{(p+1) \times d},$$

compute its singular values, and record the smallest and largest, $\sigma_{\min}\big(J_{\boldsymbol{\phi}^{(t)}}(\boldsymbol{x}_i)\big)$ and $\sigma_{\max}\big(J_{\boldsymbol{\phi}^{(t)}}(\boldsymbol{x}_i)\big)$. We then aggregate these over all training points,

$$\hat{m}_x^{(t)} = \min_{1 \le i \le N_{\text{tr}}} \sigma_{\min}\big(J_{\boldsymbol{\phi}^{(t)}}(\boldsymbol{x}_i)\big), \qquad \hat{L}_x^{(t)} = \max_{1 \le i \le N_{\text{tr}}} \sigma_{\max}\big(J_{\boldsymbol{\phi}^{(t)}}(\boldsymbol{x}_i)\big),$$

which serve as empirical estimates of the Jacobian lower bound and the Lipschitz constant in Assumption 1. Storing these statistics every 10 epochs yields a trajectory $\{\hat{m}_x^{(t)}, \hat{L}_x^{(t)}\}_t$; we summarize their distribution over training by histograms in Figures 8.

As shown in Figure 8a, the estimated lower bound $\hat{m}_x^{(t)}$ is consistently larger than 1.0, indicating that the proxy network's Jacobian is numerically well conditioned and bounded away from zero throughout training. Figure 8b shows that the largest singular value $\hat{L}_x^{(t)}$ remains below 25.0, so the Lipschitz constant does not explode and the gradient field stays stable. Taken together, these observations support the view that the Jacobian of the proxy network satisfies a uniform lower bound and a finite Lipschitz constant in the regimes considered in our experiments.

**Assumption 2: strong convexity and smoothness of the alignment loss.** Assumption 2 concerns the curvature of the alignment loss with respect to the codebook parameters. Recall that for cluster $j$, the alignment loss is a least–squares term whose Hessian is proportional to the Gram matrix of the covariates assigned to that cluster. Concretely, let $X_j^{(t)} \in \mathbb{R}^{n_j \times p}$ denote the design matrix of covariates assigned to cluster $j$ at epoch $t$, and let

$$H_j^{(t)} = 2 X_j^{(t)\top} X_j^{(t)}$$

be the corresponding Hessian. Every 10 epochs, we compute the smallest and largest eigenvalues of each $H_j^{(t)}$, and summarize them as

$$\mu^{(t)} = \min_{1 \le j \le K} \lambda_{\min}\big(H_j^{(t)}\big), \qquad L^{(t)} = \max_{1 \le j \le K} \lambda_{\max}\big(H_j^{(t)}\big),$$

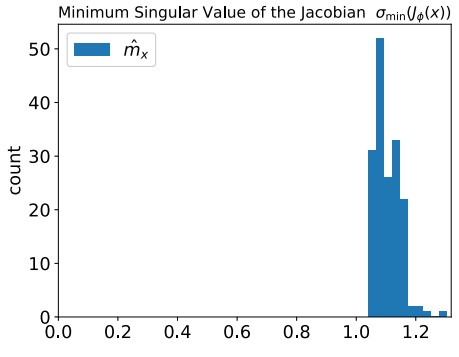
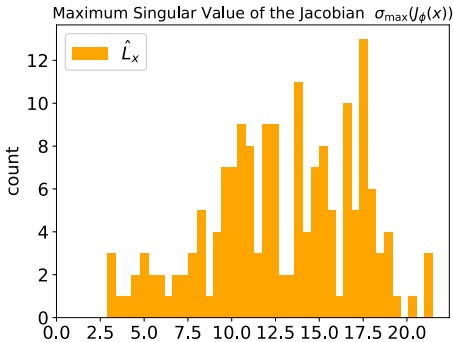

(a) Minimum singular value $\sigma_{\min}\big(J_\phi(\boldsymbol{x})\big)$ $(\hat{m}_x)$.

(b) Maximum singular value $\sigma_{\max}\big(J_\phi(\boldsymbol{x})\big)$ $(\hat{L}_x)$.

Figure 8: Empirical singular–value spectrum of the proxy-network Jacobian on the CONCRETE dataset, collected every 10 epochs over the course of training. The distributions indicate a stable Jacobian lower bound and a non–exploding Lipschitz constant, providing empirical support for Assumption 1.

which provide empirical estimates of the strong–convexity and smoothness constants in Assumption 2. As before, we record these quantities throughout training and visualize their distributions in Figures 9.

Figure 9a shows that the estimated strong–convexity constant $\mu^{(t)}$ is always larger than 6.0, indicating that each cluster's alignment objective is uniformly and numerically strongly convex. At the same time, Figure 9 demonstrates that the smoothness constant $L^{(t)}$ concentrates around values below approximately 1300.0 and does not diverge, confirming that the curvature of the alignment loss remains well controlled. These diagnostics suggest that the alignment loss is both strongly convex and smooth in the empirical regimes where CG-CLR is trained.

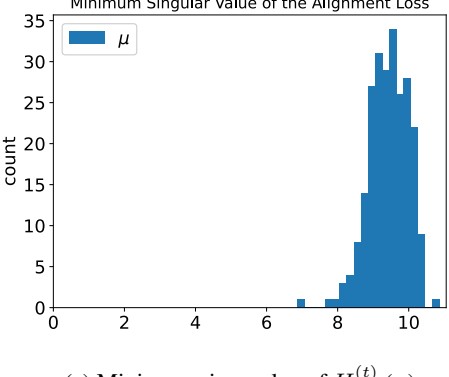
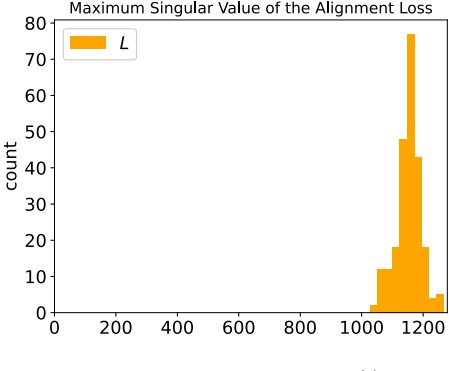

(a) Minimum eigenvalue of $H_j^{(t)}$ $(\mu)$.

(b) Maximum eigenvalue of $H_j^{(t)}$ $(L)$.

Figure 9: Empirical curvature of the alignment loss on the CONCRETE dataset, computed every 10 epochs. The estimated strong–convexity constant $\mu$ remains safely bounded away from zero, while the smoothness constant $L$ stays finite, supporting Assumption 2.

Overall, the Jacobian and curvature statistics in Figures 8–9 indicate that the regularity conditions required by our convergence analysis are not merely abstract assumptions: in a representative real dataset, the proxy network and alignment loss behave in a numerically well–conditioned manner, with bounded Jacobian spectrum and stable strong convexity and smoothness parameters.

### F.10 EMPIRICAL STUDY OF ASSUMPTION 3

Many classical clusterwise linear regression studies—including both traditional and more recent mixed linear regression methods (Kong et al., 2020; Ghosh & Mazumdar, 2024)—rely on the assumption that clusters are well separated, so that each local linear regressor can be correctly recovered. To assess the validity of this assumption in practice, we measure the empirical *cluster-*

*separation gap* after training, defined as

$$\Delta = \min_i \min_{j \neq j'} \left| \mathbf{x}_i^\top (\tilde{\mathbf{w}}_j - \tilde{\mathbf{w}}_{j'}) \right|,$$

which captures the minimum predictive discrepancy between distinct regressors over the training samples. Table 18 reports the 95% confidence intervals of $\Delta$ across 20 independent runs for all seven datasets.

Table 18: Empirical cluster-separation gap $\Delta$ with 95% confidence intervals over 20 runs.

| Dataset | $\Delta$ (95% CI) |
|---|---|
| CONDUCT | $[9 \times 10^{-5},\ 3 \times 10^{-4}]$ |
| HOUSING | $[8 \times 10^{-7},\ 2 \times 10^{-6}]$ |
| BIKE | $[2 \times 10^{-6},\ 6 \times 10^{-4}]$ |
| ELECTRICAL | $[2 \times 10^{-7},\ 5 \times 10^{-7}]$ |
| PLANT | $[1 \times 10^{-5},\ 3 \times 10^{-5}]$ |
| WINE | $[8 \times 10^{-6},\ 3 \times 10^{-5}]$ |
| CONCRETE | $[1 \times 10^{-2},\ 4 \times 10^{-2}]$ |

Across all datasets, the empirical gaps $\Delta$ are extremely small, typically $\Delta \leq 10^{-1}$. Despite this near-zero separation, CG-CLR continues to exhibit monotone descent and achieves strong test RMSE, demonstrating that its convergence and predictive performance do not rely on strong cluster separability. These results indicate that the method remains empirically robust even in the regime where clusters strongly overlap, thereby relaxing one of the key assumptions often imposed in prior work.

## G    LLM USAGE

Large Language Models (LLMs) were used in a limited capacity to aid in the preparation of this manuscript. Specifically, LLMs were employed solely for writing support tasks such as grammar checking, polishing sentence structure, and improving readability of author-written drafts. They were not involved in research ideation, methodological design, data analysis, or result interpretation. The authors take full responsibility for the content of this paper.

