# OpenReview forum: "Covariate-Guided Clusterwise Linear Regression for Generalization to Unseen Data"
_ICLR.cc/2026/Conference — ICLR 2026 Poster_

### Official Review · Reviewer_hxki · 2025-11-01

**Soundness:** 4
**Presentation:** 4
**Contribution:** 3
**Rating:** 6
**Confidence:** 5

**Summary:**

This paper proposes Covariate-Guided Clusterwise Linear Regression (CG-CLR), which jointly learns a covariate-based assignment rule and multiple local linear regressors in an end-to-end fashion. A feedforward neural network predicts a per-sample proxy regressor using only the covariate, which are quantized to the nearest of 𝐾 learned regressors. The paper provides convergence and generalization guarantees and introduces an F-test-based method to select 𝐾. Experiments on synthetic and tabular datasets show that CG-CLR achieves superior accuracy comparable to other models that have "large coverage".

**Strengths:**

1. Novel end-to-end CLR framework: The paper introduces a novel way to partition the covariate space using a feedforward neural network and to assign a regressor to these partitions simultaneously. The key ideas are captured clearly in Equations (3) and (4).

2. Because the model uses local linear regressors, it maintains a level of interpretability – each cluster’s behavior can be understood via a simple linear formula. The learned “codebook” of regressors is compact, making the overall model relatively easy to inspect and reason about. This is an advantage over complex black-box models and aligns with the paper’s motivation of balancing accuracy and interpretability.

3. Theoretical rigor: The paper proves monotonic descent of the empirical risk and even linear convergence to an optimum under certain conditions. Additionally, it derives a PAC-style excess risk bound.

4. The paper offers a principled way to choose the number of clusters  K via an F-test.

5. Empirical performance: The model demonstrates strong performance on several benchmarks compared to “large-coverage” models.

6. Clarity: The paper is clearly written and very well-organized, making both the methodology and theory easy to follow.

**Weaknesses:**

1. The assumptions in the theoretical proofs could be quite restrictive. It would be helpful to discuss why these assumptions are reasonable and under what conditions they might fail.

2. The framework may fail when groups have very different regression vectors but very similar covariate distributions, since the assignment is guided only by covariates.

3. The clusters are not necessarily linear or interpretable; the learned partitions may be arbitrary or data-driven without clear meaning.

4. Simpler methods such as Random Forest, XGBoost, or CatBoost often outperform CG-CLR, suggesting it is not clearly superior to strong baselines.

Minor Comment:
typo in line 184

**Questions:**

1. It would be interesting to see results for higher-dimensional synthetic experiments (dimension > 2) and for more practical covariate distributions, such as Gaussian mixtures.

---

> ### Author Response · Authors · 2025-11-21
> **Response to Reviewer hxki — Weakness 1**
>
> We sincerely thank the reviewer for the thoughtful observation regarding the restrictiveness of our assumptions. We fully agree that a clearer explanation of why each assumption is reasonable—and in what situations it may fail—strengthens the theoretical section. We appreciate the opportunity to elaborate on this point.
>
> In the revised manuscript, we have expanded the intuitive interpretation of each assumption (highlighted in blue in the updated Section 3.3) so that readers can better understand the practical meaning of these conditions and the scenarios in which they may not hold.
>
> - **Assumption 1** concerns the Lipschitz continuity and Jacobian lower bound of the proxy network.
>   This is a standard regularity condition in gradient-based optimization: if the mapping is not Lipschitz, gradients can explode, and if the Jacobian degenerates, gradients can vanish. These issues make parameter updates unstable or uninformative. Thus, the assumption reflects a typical requirement for ensuring well-behaved gradient descent rather than imposing a specific structural limitation.
>
> - **Assumption 2** imposes strong convexity and smoothness on the codebook alignment loss.
>   This assumption guarantees that, once the cluster assignments are fixed, the codebook update corresponds to a well-posed least-squares problem, as noted in the revised text. Situations where this least-squares structure breaks down—for example, when the number of points assigned to a cluster is smaller than the covariate dimension $p+1$, or when the corresponding design matrix fails to have full rank—would violate this assumption and lead to unstable updates.
>
> - **Assumption 3** states a minimal separation between the optimal regressors.
>   This assumption plays the role of ensuring that the cluster labels are well defined: under the generative setting considered in our analysis, each data point is assigned to the regressor whose prediction is closest to its true response, and this assignment is meaningful only when distinct regressors do not produce identical predictions. If this separation fails—for example, if two regressors yield essentially the same prediction—then the underlying cluster structure becomes ambiguous, and the hard-assignment rule is no longer identifiable. This situation may arise in idealized noiseless piecewise-linear settings when the number of underlying linear regions is overestimated.
>
> - **Assumption 4** concerns the expressiveness of the proxy network.
>   The intention is simply to require that the proxy has enough capacity to approximate the underlying mapping from covariates to regressors. As clarified in the revision, this condition reflects the universal-approximation-style requirement needed for the convergence arguments. If the proxy is not expressive enough, the model may fail to recover the intended structure.
>
> Across all four assumptions, our aim is not to impose strong modeling restrictions but to articulate the regularity conditions under which the Lyapunov-style convergence proof and the clusterwise regression interpretation remain valid. In the revised manuscript, we added intuitive explanations below each assumption to help readers understand when these conditions are reasonable and how violations may manifest in practice.
>
> We sincerely appreciate the reviewer’s comment, which motivated us to strengthen and clarify this part of the paper.

---

> ### Author Response · Authors · 2025-11-21
> **Response to Reviewer hxki — Weakness 2**
>
> We sincerely appreciate the reviewer’s insightful observation.
> As noted, because CG-CLR performs test-time assignment using only covariates, the framework can face challenges in settings where groups have very similar covariate distributions while possessing markedly different regression vectors.
>
> We fully agree that, in such cases, covariate-based assignment may misidentify the appropriate local model. This limitation is already reflected in our empirical results—for example, Figure 3 in the revised manuscript shows that prediction errors increase near regions where the covariate distributions of different groups overlap. This behavior arises naturally from the fact that neither the proxy network nor the codebook has access to the response at test time; when covariates are nearly indistinguishable but the underlying generating functions differ, covariate-only assignment can become challenging and may lead to noticeable degradation in performance.
>
> At the same time, we would like to clarify that this phenomenon reflects a shift in problem formulation, rather than a deficiency specific to CG-CLR. Classical CLR assumes access to the response when determining cluster membership (either implicitly through latent indicators or explicitly through min-loss selection), and therefore focuses solely on recovering the underlying linear components without addressing the test-time routing problem. In contrast, our setting requires estimating the mapping $x$ to $y$ for previously unseen covariates where $y$ is not observable, and thus necessarily relies only on $x$ for assignment. Consequently, scenarios where two groups share nearly identical covariates but differ sharply in their regression vectors fall somewhat outside the scope of what covariate-guided assignment alone can resolve, and represent an intrinsic challenge of the prediction-oriented formulation we address.
>
> We view this not as a contradiction to the reviewer’s point but as helpful clarification of the intended applicability of CG-CLR: our method extends CLR to the prediction setting by providing a principled covariate-based assignment mechanism, while inheriting the fundamental limitations of any response-free decision rule. Incorporating additional information—such as auxiliary variables, richer structural priors, or latent-variable estimation—would be required to resolve such truly ambiguous cases and represents a valuable direction for future work.
>
> We thank the reviewer again for raising this important point, which helps delineate the scope of our contribution while highlighting meaningful opportunities for further extension.

---

> ### Author Response · Authors · 2025-11-21
> **Response to Reviewer hxki — Weakness 3 and Weakness 4**
>
> We appreciate the opportunity to clarify the interpretability aspects of CG-CLR and its relationship to stronger predictive baselines.
>
> ## **Weakness 3 — Interpretability of clusters**
>
> We agree with the reviewer that, in general, the learned partitions in covariate-guided CLR need not exhibit simple geometric or semantic structures; the partitions are ultimately data-driven solutions to the empirical objective. This observation is consistent with the nature of *agnostic* clusterwise linear regression, whose goal is to identify $K$ locally linear rules that minimize predictive loss rather than to impose explicit semantic meaning on the clusters.
>
> At the same time, recent work on interpretable local modeling supports the relevance of such structures.
> For instance, the Interpretable Mixture of Experts (IME) framework [1] shows that assigning samples to multiple linear experts can offer interpretable, equation-level descriptions of local behavior even when the partition boundaries themselves are complex. Similarly, **InterpretCC** [2] demonstrates that even locally feature-selective mixture models—where the assignment boundaries may not be intrinsically interpretable—can still yield meaningful explanations by exposing the functional form used for each region. In our case, the codebook regressors serve this role: once a point is assigned, the corresponding linear rule provides a concise and transparent characterization of the local pattern.
>
> Thus, although CG-CLR does not guarantee interpretable partitions, it does produce interpretable local mechanisms, and this perspective is consistent with one line of recent interpretable ML research showing that useful interpretability can be obtained from the form of the local models themselves, even when the assignment boundaries are complex. At the same time, we recognize that global interpretability remains an important and active direction in the broader literature, and approaches that explicitly structure the partitions represent another valuable line of work.
>
> We fully acknowledge that extending the framework toward explicitly interpretable partitions—e.g., with additional structure in the assignment rule—would be an interesting future research direction beyond the current scope of the paper.
>
> ## **Weakness 4 — Comparison with strong black-box baselines**
>
> We appreciate the reviewer’s observation that ensemble methods can outperform CG-CLR on some benchmarks. Indeed, as shown in Table 2, these models often achieve the lowest predictive error, which is fully consistent with their strong expressive capacity and long-established performance as highly effective black-box predictors.
>
> We would like to clarify that our goal is not to position CG-CLR as a replacement for these ensemble methods. Rather, CG-CLR aims to provide a **middle ground** between predictive accuracy and interpretability, in line with the motivation behind recent interpretable mixture-of-experts models such as IME [1]. These works also emphasize that, while black-box ensembles can achieve the strongest raw predictive performance, models with explicit local experts are often preferable when transparency, local structure, and the ability to inspect the underlying functional form are important considerations.
>
> Within this interpretable modeling space, our intention is not to compare directly against large black-box ensembles, but rather to evaluate **how CG-CLR performs relative to other CLR-style and expert-based methods**, which share similar modeling assumptions and interpretability goals. In this regard, CG-CLR consistently achieves the best predictive performance among clusterwise and mixture-of-experts approaches considered in our study, while preserving the classical advantages of CLR—namely, the use of a small set of shared linear mechanisms that remain directly inspectable.
>
> Thus, we see the boosting-based baselines in Table 2 as providing helpful **upper bounds on achievable accuracy** for each dataset, rather than as direct competitors along the interpretability axis. We hope this clarification makes our intended positioning more explicit: CG-CLR is designed for scenarios where one desires the transparency and diagnostic value of CLR-style local models, and within that family of interpretable approaches, it offers the strongest predictive performance while maintaining clear, locally linear explanations.
>
>
> ## **Typo in line 184**
>
> As a final note, we also corrected the minor typo pointed out by the reviewer (line 184: "Equation equation 2" → "Equation 2"). We appreciate the reviewer bringing this to our attention.
>
>
> ## **References**
>
> [1] Ismail, A. A., Dryden, N., Krishnan, S., & Ragan-Kelley, J. (2023). *Interpretable Mixture of Experts*. Transactions on Machine Learning Research.
>
> [2] Swamy, V., Nemlekar, H., Zhang, S., Singh, S., & Ribeiro, M. T. (2025). *Intrinsic User-Centric Interpretability through Global Mixture of Experts*. International Conference on Learning Representations.

---

> ### Author Response · Authors · 2025-11-21
> **Response to Reviewer hxki — Question 1**
>
> We sincerely thank the reviewer for this helpful suggestion. We agree that evaluating CG-CLR on higher-dimensional synthetic settings with more realistic covariate distributions is valuable for understanding the robustness of the method.
>
> In response, we added a new synthetic experiment in **Appendix E.2** of the revised manuscript. Following the reviewer’s recommendation, we constructed the covariates from a mixture of **three Gaussian distributions**, each with randomly generated means and covariance matrices. We also increased the input dimensionality to **10 dimensions**, and sampled $K$ ground-truth linear regressors with randomly drawn coefficients.
>
> As illustrated in **Figure 16 (Appendix E.2)**, CG-CLR successfully recovered the underlying local linear structure in this setting. In particular, the **relative $\ell_2$ error of the recovered coefficients was approximately 2–3\%**, indicating accurate identification of the latent regressors even in higher-dimensional covariate spaces.
>
> We appreciate the reviewer’s suggestion, which significantly strengthened the experimental validation of the method.

---

### Official Review · Reviewer_wePp · 2025-11-02

**Soundness:** 3
**Presentation:** 1
**Contribution:** 2
**Rating:** 6
**Confidence:** 3

**Summary:**

The paper introduces covariate-guided Clusterwise Linear Regression (CLR), a framework that jointly learns cluster assignments and local linear models through end-to-end gradient-based optimization. By combining a proxy network with hard vector quantization, the method ensures convergence, provides theoretical risk guarantees, and adapts smoothly from global to instance-wise regressions. Experiments show it accurately reconstructs piecewise-linear functions and outperforms existing CLR and mixture-of-experts approaches on tabular benchmarks.

**Strengths:**

- The paper tackles an important and well-motivated problem in modeling local linear relationships in tabular regression tasks.

- The proposed approach effectively integrates a simple yet elegant idea—using a proxy network (similar to a hypernetwork) to predict regression coefficients—into an end-to-end optimization framework.

- The method is theoretically well justified, offering convergence guarantees, a PAC-style excess-risk bound, and a principled F-test analysis of model complexity.

- The approach demonstrates strong empirical performance, accurately recovering piecewise-linear structures and outperforming existing CLR and mixture-of-experts baselines.

**Weaknesses:**

- The proposed approach appears somewhat incremental, primarily combining elements of hypernetworks with an expectation–maximization-style framework using hard assignments. While effective, the conceptual novelty over existing methods may be limited.

- The paper’s presentation is weak—the writing is often unclear, and the main contributions focus more on the properties of the proposed method rather than its broader impact or significance. Although the method itself is relatively straightforward, the paper is difficult to follow, and important implementation details—particularly regarding how the model is applied at test time—are missing.

- While the paper provides some theoretical assumptions about the mapping function used for cluster assignments, it remains unclear how these properties are enforced during training when using an MLP. It would also be useful to discuss whether other architectures could replace the MLP and how such substitutions might affect convergence or theoretical guarantees.

**Questions:**

Please refer to the weaknesses section.

---

> ### Author Response · Authors · 2025-11-21
> **Response to Reviewer wePp — Weakness 1**
>
> We sincerely thank the reviewer for the thoughtful assessment.
> We fully understand the concern that the proposed approach may appear incremental, as it combines elements reminiscent of hypernetworks with EM-style hard assignments. We appreciate the opportunity to clarify where the core conceptual contribution lies, while acknowledging that parts of the exposition may not have communicated these points as clearly as intended.
>
> First, we agree that CG-CLR uses components that have appeared individually in prior work.
> Hypernetworks naturally generate instance-wise parameters, and hard assignments are classical in mixture models. Our contribution is not in introducing these components themselves, but in **integrating them into a stable optimization framework** where their interaction can be analyzed rigorously.
>
> ## Where the conceptual novelty lies
>
> Despite using familiar ingredients, CG-CLR departs in several substantial ways from prior CLR and MoE-style approaches:
>
> ### **A convergence-guaranteed, end-to-end CLR framework**
>
> Existing CLR/MoE approaches typically rely on heuristic couplings between assignment rules and expert updates, especially under hard assignments, where no convergence guarantees are available and training dynamics are often unstable.
>
> In contrast, CG-CLR introduces:
> - a **unified objective**,
> - a **stop-gradient–based decomposition of updates**, and
> - a **Lyapunov-style descent argument** establishing global convergence under standard regularity assumptions.
>
> This moves beyond a heuristic combination of known components: it provides a **principled, convergent optimization structure** that, to our knowledge, has not been available in prior CLR formulations.
>
> ### **Assumption-driven interpretation of when training may face difficulties**
>
> CG-CLR separates the roles of the proxy network, the codebook, and the hard assignment rule.
> The theoretical assumptions (Assumptions 1–4) do not enumerate failure modes exhaustively, but they provide a **structured way to interpret** when difficulties may arise—for example:
> - Jacobian degeneracy affecting proxy updates (Assumption 1),
> - rank-deficient clusters affecting alignment-loss curvature (Assumption 2).
>
> This theoretical framing contrasts with earlier heuristic approaches, where training instabilities are difficult to analyze because the mechanisms are tightly intertwined.
>
> ### **A practical and stable test-time assignment rule for CLR**
>
> Classical CLR requires access to the *unknown* response to determine cluster assignments at test time.
> CG-CLR overcomes this limitation by learning a **covariate-guided proxy assignment rule** jointly with the regressors, while ensuring that the training process remains stable.
> Providing a reliable test-time assignment mechanism is a key practical gap in CLR that, as far as we are aware, had not been addressed.
>
> ### **Connection to broader methodological trends**
>
> Recent work such as **COMM** (Contextual Optimization Under Model Misspecification; ICML 2025) [1] highlights a broader trend: nonconvex learning–decision systems can be made tractable by designing a disciplined surrogate objective and update structure.
> CG-CLR aligns with this philosophy in the specific context of clusterwise regression—the novelty lies not in simply combining components, but in constructing a **trainable and analyzable system** that resolves longstanding instability issues in CLR.
>
> We mention this work only to situate CG-CLR within this broader methodological trajectory; our technical development is orthogonal to COMM.
>
>
> We deeply appreciate the reviewer’s perspective and understand why the method may initially appear incremental.
> Our goal in the revision has been to better articulate why we believe CG-CLR represents a meaningful step: it provides a **stable, convergent, and interpretable end-to-end CLR framework** that, to our knowledge, has not existed previously.
> We hope this clarification helps distinguish CG-CLR from a purely incremental combination of existing ideas.
>
>
> ### **Reference**
>
> [1] Bennouna, O., Sinha, S., Pilancı, M., \& Ma, T. (2025). *Contextual Optimization Under Model Misspecification: A Tractable and Generalizable Approach*. ICML 2025.

---

> ### Author Response · Authors · 2025-11-21
> **Response to Reviewer wePp — Weakness 2**
>
> We sincerely thank the reviewer for this detailed and constructive feedback.
> We fully acknowledge the concerns regarding clarity, presentation, and missing test-time implementation details, and we are grateful for the opportunity to improve these aspects of the paper.
>
> First, we agree that several parts of the initial submission did not communicate the motivation and broader significance of CG-CLR as clearly as they should have. As the reviewer noted, our writing placed substantial emphasis on the theoretical aspects of the method, and the exposition implicitly assumed familiarity with long-standing challenges in clusterwise linear regression. We recognize that this may have made the narrative difficult to follow for readers less familiar with the CLR setting, and we appreciate the reviewer pointing this out.
>
> Second, we also agree that the initial version did not sufficiently explain **how CG-CLR performs prediction at test time**, which can understandably create ambiguity about how the model operates in practice. In response, we have added **Section 3.2 ("Test-time prediction")**, which now provides a clear and self-contained description of both available inference modes:
> - **Codebook-based prediction**, where assignments are made via the learned proxy and predictions use the corresponding codebook regressor, and
> - **Proxy-based prediction**, which directly uses $W_\phi(x)$ for regression.
>
> We hope this addition addresses the reviewer’s concern regarding missing implementation details and makes the workflow substantially clearer.
>
> Beyond this, we made targeted edits to improve clarity in Sections 3 and 4—for example, refining the explanations of the training procedure and the alignment mechanism. These edits are visible as blue-highlighted text in the revised manuscript. While the broader motivation section was not fully rewritten, we took care to streamline several transitions and descriptions to improve readability.
>
> We are genuinely grateful for the reviewer’s candid comments. They prompted us to re-examine key parts of the exposition, and we hope the revised draft is significantly easier to follow as a result.

---

> ### Author Response · Authors · 2025-11-21
> **Response to Reviewer wePp — Weakness 3**
>
> We sincerely thank the reviewer for raising this important question.
> We fully agree that the initial submission did not clearly explain how the theoretical assumptions on the assignment mapping relate to practical training with an MLP, nor did it sufficiently clarify whether other architectures could be used in place of the proxy network. We appreciate the opportunity to elaborate on these points.
>
> ## Enforcing the assumptions during training
>
> As the reviewer noted, the assumptions in Section 3.3 (Lipschitz continuity, Jacobian lower bound, curvature of the alignment loss, and cluster separation) are not enforced explicitly during training.
> Consistent with the concerns raised by other reviewers, we acknowledge that the original manuscript did not experimentally examine whether these properties hold in practice.
>
> To address this, we added **Appendix F.9** in the revised submission, which provides numerical diagnostics for the key assumptions:
>
> - For **Assumption 1**, we track the minimum and maximum singular values of the Jacobian of the proxy network throughout training. These values remain bounded and stay away from zero in all experiments, suggesting that the MLP does not enter regimes of exploding or vanishing gradients.
>
> - For **Assumption 2**, we measure the smallest eigenvalue of the Hessian of the alignment loss. The eigenvalues remain positive during training across datasets, supporting the strong-convexity requirement.
>
> These analyses are not meant to formally *enforce* the assumptions but to demonstrate that **in typical training regimes, the MLP proxy empirically respects the regularity conditions required by the theory**.
>
> ## Possibility of replacing the MLP proxy network
>
> Regarding the reviewer’s question about substituting the MLP with other architectures:
> the core requirement of CG-CLR is that the proxy network be **differentiable** so that the composite objective can be optimized using gradient-based updates.
>
> From this perspective:
>
> - Architectures such as CNNs, attention-based models, or other differentiable function approximators could in principle replace the MLP **without affecting the structure of the convergence argument**, as long as the corresponding Jacobian and Lipschitz properties remain well behaved.
> - On the other hand, architectures that **do not admit gradient-based optimization** (e.g., decision trees or discrete models) cannot be directly substituted, because our update rules and theoretical guarantees rely on smoothness and backpropagation.
>
> Extending CG-CLR to non-smooth or non-differentiable proxies—for example by using implicit differentiation or structured surrogates—would be an interesting direction for future work, but is beyond the scope of the current paper.
>
>
> We sincerely appreciate the reviewer’s comment, which helped us clarify both the empirical status of the assumptions and the flexibility of the proxy architecture within the CG-CLR framework.

---

### Official Review · Reviewer_QYPe · 2025-11-08

**Soundness:** 3
**Presentation:** 4
**Contribution:** 3
**Rating:** 8
**Confidence:** 2

**Summary:**

The paper proposes Covariate-Guided Clusterwise Linear Regression (CG-CLR), an end-to-end framework that jointly learns (i) a response-free assignment rule mapping a covariate vector to one of $K$ local linear regressors and (ii) the $K$ regressors themselves. The system uses a proxy network that outputs instance-wise coefficient vectors; a hard vector-quantization step assigns each input to the nearest codebook regressor; and training alternates between updating the proxy and the codebook via a composite “fit + alignment” objective. The authors show monotone descent of a Lyapunov-style objective with linear convergence under assumptions, a PAC-style excess-risk bound, and a nested-model $F$-test to tune $K$. Empirically, CG-CLR recovers synthetic piecewise-linear structure and is competitive with tree ensembles on several tabular benchmarks.

**Strengths:**

* The paper is well-organized and clearly written. The paper has clean and good illustrations for their takeaways and empirical findings.
* The paper has a good mix of both theoretical and empirical evidence. The method proposed is also generic enough. I must admit I'm not an expert in this field, but the results in general strike me as interesting.

**Weaknesses:**

Major comments:
* The paper claims one of the strength is that prediction is accurate without ever observing the true response $y_{i'}$---however, there is really nothing surprising there. This is basically generalization in conventional machine learning setup, and is not special to this particular task. The authors should make it clear that the actual strength is the proxy is label independent, which means the algorithm is unsupervised in natural. I'm also not sure how much ``the inner minimization over $j$ cannot be evaluated at test time because the response $y_i'$ associated with an unseen covariate vector $x_i′$ is unknown at prediction time'' is undesirable. We can always estimate generalization error by cross validation/hold-out set if needed.
* The linear convergence result requires a uniform lower bound on the proxy network's Jacobian, strong convexity and smoothness of the alignment loss. Those conditions seem strong and probably unverifiable in reality. If for some toy model or the empirical setup in the paper, those conditions can be justified (even numerically), they would be much more convincing.

Minor comments:
* P7. "form" should be "from"

**Questions:**

* My main questions are in the weaknesses section.
* Overall I think the paper provides a useful engineering tool, while the discussion about its theoretical results, especially on the assumptions are limited. How would the author justify their assumptions empirically? How should those requirements be tested?

---

> ### Author Response · Authors · 2025-11-21
> **Response to Reviewer QYPe — Weakness 1 (with typo)**
>
> We sincerely thank the reviewer for this thoughtful comment. We appreciate the opportunity to clarify the intended meaning behind our discussion of prediction without observing the true response, as well as the role played by label-independent assignment in CG-CLR. We also corrected the minor typo noted by the reviewer (from 'form' to 'from') in the revised manuscript.
>
> We agree with the reviewer that, in a conventional supervised learning setting, achieving good prediction without access to the test-time response is simply the standard notion of generalization. Our intention was not to claim otherwise. Rather, as the reviewer correctly pointed out, the novelty lies in the fact that **CG-CLR performs cluster assignment in a label-independent manner**, whereas classical CLR methods rely on *response-dependent* assignments. We fully agree that this distinction should have been stated more clearly.
>
> At the same time, the emphasis on label independence is tied to a specific challenge that arises in clusterwise regression. In the classical CLR literature, the risk of a solution is evaluated under the assumption that, for each point, one can assign the data to the regressor that minimizes the prediction error with respect to the *true* response. This *oracle* assignment rule is not available at test time, because the true $y$ is unknown. As a consequence, the conventional population risk used in CLR analyses — what we refer to as the **response-aware risk** — is not operationally usable when making a single prediction for a new covariate. Our description of this as “undesirable’’ was meant only in this operational sense: the risk depends on information that is not observable for the test-time task that motivates CG-CLR.
>
> Cross-validation or hold-out evaluation is, of course, possible, but these procedures estimate performance *after* oracle assignment, not under a covariate-only assignment rule. Thus, while such estimates are certainly useful for model selection, they do not directly address the single-sample assignment challenge that motivates our formulation.
>
> To make this distinction clearer, we have revised **Section 3.1** of the manuscript. The updated text (highlighted in blue) now explicitly contrasts the **response-aware risk (infeasible)** — which depends on knowing $y$ — with the **covariate-guided risk (feasible)** used by CG-CLR. We hope this clarification makes our intended message more transparent: the key strength of CG-CLR is not generalization alone, but the ability to perform test-time assignment using only covariates, without relying on any oracle information about the response.
>
> We are grateful to the reviewer for pointing out this ambiguity, and we believe the revised wording leads to a clearer and more accurate articulation of the contribution.

---

> ### Author Response · Authors · 2025-11-21
> **Response to Reviewer QYPe — Weakness 2 and Question**
>
> We sincerely thank the reviewer for raising this important point. We agree that the assumptions required for our convergence result—such as Lipschitz continuity and a Jacobian lower bound for the proxy network, and strong convexity/smoothness of the alignment loss—may appear strong at first glance, and we appreciate the opportunity to clarify their role and empirical relevance.
>
> ## Clarifying the intent of the assumptions
>
> As discussed in the revised manuscript (Section 3.3, with intuitive explanations added in blue text), the four assumptions serve as **regularity conditions** that make the Lyapunov-style argument tractable. Each condition expresses a standard requirement seen in many convergence analyses:
>
> - **Assumption 1**.
>   The Lipschitz condition prevents gradients from becoming excessively large, while the Jacobian lower bound prevents gradients from vanishing. These are classical stability requirements in gradient-based optimization and ensure that parameter updates behave in a controlled manner.
>
> - **Assumption 2**.
>   Once the cluster assignments are fixed, this assumption ensures that the alignment-loss update with respect to the codebook corresponds to a well-posed least-squares regression problem. Violations may occur when fewer than $p+1$ samples fall into a cluster or when the resulting design matrix lacks full rank.
>
> - **Assumption 3**.
>   This assumption formalizes when the *true* cluster labels are identifiable under the generative model used in the analysis. If two regressors produce nearly identical predictions, the ground-truth assignment becomes ambiguous. Importantly, as illustrated in the numerical study below, Assumption 3 is not contradicted by the experiments—it is simply not strongly satisfied in many practical datasets. Moreover, because it concerns identifiability rather than training dynamics, CG-CLR can still train reliably even when the empirical separation is weak.
>
> - **Assumption 4**.
>   This reflects the universal-approximation-style requirement that the proxy should have enough capacity to approximate the mapping from covariates to local regressors.
>
> We fully agree with the reviewer that these assumptions cannot be strictly verified. Their purpose is to characterize the regime in which our theoretical analysis applies, rather than to imply that they always hold.
>
> ## Empirical justification
>
> To address the reviewer’s question regarding empirical justification, we expanded the supplementary material with numerical diagnostics in **Appendix F.9 and Appendix F.10**. These experiments are designed to illustrate how the assumptions manifest in practice in the training regime of CG-CLR.
>
> ### **Appendix F.9 — Assumption 1 (Lipschitz/Jacobian) and Assumption 2 (curvature of alignment loss)**
>
> - **Assumption 1 (proxy Jacobian).**
>   Appendix F.9 (Figure 8) tracks the minimum and maximum singular values of the Jacobian throughout training.
>   The values remain bounded and stay uniformly away from zero, which is qualitatively consistent with both the Lipschitz and Jacobian-lower-bound components of Assumption 1.
>   While not a formal verification, this suggests that the proxy network does not enter regimes of exploding or vanishing gradients under typical training conditions.
>
> - **Assumption 2 (Hessian of alignment loss).**
>   Appendix F.9 (Figure 9) reports the smallest eigenvalues of the Hessian of the alignment loss (computed after assignments are fixed at each iteration).
>   These eigenvalues remain positive during training, which aligns with the strong-convexity requirement of Assumption 2.
>
> These findings indicate that the model tends to operate in a regime where Assumptions 1 and 2 are reasonably satisfied empirically.
>
> ### **Appendix F.10 — Assumption 3 (cluster separation)**
>
> Appendix F.10 provides numerical diagnostics related to Assumption 3.
> To evaluate whether the learned regressors remain well separated in practice, we measure the empirical cluster-separation gap which captures the smallest predictive discrepancy between any pair of regressors over the training samples.
>
> As reported in **Table 18**, the empirical values of $\Delta$ are **extremely small** across all seven datasets—typically between $10^{-7}$ and $10^{-2}$.
> This shows that, in realistic settings, regressors are often *weakly* separated rather than strongly separated.
> Nonetheless, CG-CLR still exhibits **monotone descent** and achieves **strong predictive accuracy**.
> This suggests that while Assumption 3 is helpful for the theoretical identifiability analysis, it is **not critical for stable training**, and the method remains empirically robust even when the separation condition holds only weakly.
>
> We are grateful for the reviewer’s comment, which motivated us to add these analyses and thereby make the theoretical requirements more transparent in practice.

---

> ### Comment · Reviewer_QYPe · 2025-11-25
> **Response to the revision**
>
> Thanks the reviewer for the thoughtful responses and revision. I agree that my previous suggestion of using CV/hold-out sets for evaluating the risk is not feasible. I appreciate the updated empirical evidence. I'll retain my favorable score for the paper.

---

> > ### Author Response · Authors · 2025-11-27
> > **Reply to Reviewer QYPe**
> >
> > We sincerely appreciate the reviewer’s thoughtful follow-up and the care taken in revisiting our clarifications.
> > We are pleased that the revisions have addressed your earlier concerns, and we are grateful for your supportive assessment.

---

### Official Review · Reviewer_B3vF · 2025-11-08

**Soundness:** 2
**Presentation:** 2
**Contribution:** 2
**Rating:** 4
**Confidence:** 4

**Summary:**

The paper studies the Clusterwise Linear Regression (CLR) problem in which the goal is to find $K$ regressors for a given dataset such that for each point there is one regressor out of the $K$ which has a low loss prediction. Previous methods either used optimization to find the $K$ linear regressors but provided no way to assign a new unlabeled point to a regressor during inference, or leveraged a mixture of experts or decision trees but did not have good convergence. In this work, the authors try to address these problems by introducing a proxy network which predicts a good regressor for each input point along with a dual loss which optimizes the network as well as a codebook of candidate regressors simultaneously. Under some smoothness, convexity and cluster separation assumptions, the authors prove convergence as well as generalization error bounds for their CG-CLR method. Included experiments on synthetic and real datasets show performance gains as compared to some previous approaches.

**Strengths:**

1. The paper proposes a new technique for CLR which optimizes a proxy predictor network whose predictions at test time can be used to assign each point to a candidate regressor.
2. The paper provides theoretical guarantees under some reasonable assumptions.

**Weaknesses:**

1. The proposed method hinges on the assumption that a good proxy network can be trained, however this means that $W_\phi(\textbf{x}_i)^\textsf{T}\textbf{x}_i$ is a good label predictor. This might be a fairly strong assumption which simplifies the problem. In particular, the assumption means that the label can essentially be predicted by the proxy network.  Why not use the proxy network itself for inference instead of the set of linear regressors?
2. The experiments seem a bit inadequate: the proposed method is somewhat of an extension to the methods of [Ghosh-Mazumdar, ICML’24] and so should be compared against them. Also, the experiments should measure how well the proxy network itself is predicting the labels (as mentioned above).
3. The paper, though well written for the most part, has some parts which need clarification (see questions to authors).

**Questions:**

1. In the CG-CLR algorithm, why are the proxy network and the codebook updated separately and not together?
2. The empirical risk mentioned on page 6 should be formally defined (or a reference to the definition in the appendix included).
3. Assumption 4 seems unclear. The condition that $d \gg K(p+1)$ can trivially be achieved by adding dummy parameters. The statement of the assumption should be made more formal.
4. What are CG-CLR (Proxy) and CG-CLR (Codebook) mentioned on line 408?

---

> ### Author Response · Authors · 2025-11-21
> **Response to Reviewer B3vF — Weakness 1**
>
> We sincerely thank the reviewer for raising this thoughtful concern.
> Your question about whether the proxy network itself could be used directly for prediction helped us clarify and further articulate an important aspect of our framework.
>
> We understand the reviewer’s point that training a sufficiently good proxy network may give the impression that the proxy itself could act as a strong predictor, potentially making the codebook unnecessary. We appreciate this observation and agree that it is important to clarify the intended role of the proxy within CG-CLR.
>
> ## Clarification — the goal of CG-CLR
>
> CG-CLR is designed to identify and recover $K$ *shared* linear regressors that reflect the underlying locally linear structure of the data.
> In this context, the proxy network is not intended to replace these shared regressors. Rather:
>
> - It produces a **continuous, instance-wise coefficient estimate** that serves as a soft surrogate.
> - The hard quantization step (equation (2)) maps this estimate to *one of the $K$ codebook regressors*.
> - The alignment term encourages the proxy output and its assigned codebook vector to remain consistent during training.
>
> This interaction enables the proxy to support the emergence of stable and interpretable local linear models, rather than functioning as a standalone predictor.
>
> ## Why not simply use the proxy alone for inference?
>
> We partially agree with the reviewer that, for purely predictive purposes, directly using $x^\top W_\phi(x)$ can be a reasonable alternative.
> Accordingly, in the revised manuscript, Section 3.2 (Test-time prediction) now explicitly states:
>
> > As an alternative inference mode, one may also use the proxy coefficients directly, $\hat{y}(x_{i'}) = x_{i'}^\top W_\phi(x_{i'})$.
>
> However, relying solely on the proxy sacrifices several benefits that motivate CLR:
>
> - **Loss of interpretability:** instance-wise coefficients do not correspond to shared linear models.
> - **Loss of stability:** as shown in the updated *coefficient recovery analysis* (Figure 2b–2c), proxy outputs vary noticeably across nearby inputs, whereas the codebook regressors recover the ground-truth coefficients in a stable manner.
> - **Loss of structure:** several application domains (e.g., control, optimization, functional clustering) often require *explicit and stable* locally linear rules rather than instance-specific hypernetwork-style predictions.
>
> Thus, although proxy-only inference is feasible and now mentioned as an option, it does not achieve the primary goal of recovering the $K$ underlying linear relationships.
>
> We hope that these clarifications, along with the revisions in the manuscript, help address the reviewer’s concern and better convey the complementary roles of the proxy network and the codebook in CG-CLR.

---

> ### Author Response · Authors · 2025-11-21
> **Response to Reviewer B3vF — Weakness 2**
>
> We sincerely thank the reviewer for this constructive suggestion.
> Your comment regarding experimental comparisons—particularly with the recent EM-based method of Ghosh–Mazumdar (ICML’24) and with the proxy network’s own predictive performance—was very helpful in guiding our revisions.
>
> We agree that including these baselines provides a more complete empirical picture. In response, we have incorporated both comparisons into the revised manuscript.
>
> ## Comparison with EM-MLR (Ghosh–Mazumdar, ICML’24)
>
> We appreciate the reviewer’s observation that our method is related to EM-style clusterwise linear regression.
> Accordingly, we have added **EM-MLR** to Table 2 of the revised manuscript.
>
> Since EM-MLR does not provide a covariate-only assignment rule at test time, we first determine the optimal clusters for the training samples (as produced by the EM updates), and then assign each test sample to the cluster of its nearest neighbor in the covariate space.
> Given that prediction accuracy in these models is highly sensitive to the assignment rule, this covariate-based nearest-neighbor assignment resulted in **substantially lower predictive performance**, which aligns with prior observations in the CLR literature.
>
> ## Evaluation of proxy-only prediction (CG-CLR (Proxy))
>
> Following the reviewer’s helpful suggestion, we also report the performance of the proxy network used directly for prediction, denoted **CG-CLR (Proxy)**.
> Across most benchmark datasets, proxy-only prediction exhibits test performance that is very close to **CG-CLR (Codebook)**, with their 95% confidence intervals typically overlapping.
>
> This behavior arises from the alignment loss, which encourages the proxy’s instance-wise coefficients to remain well aligned with the codebook regressors during training.
> As a result, even when used independently at test time, the proxy network often produces predictions similar to those of the codebook.
>
> We are grateful to the reviewer for the suggestion, and we believe that incorporating both EM-MLR and proxy-only inference has helped us present a clearer and more thorough empirical evaluation of CG-CLR.

---

> ### Author Response · Authors · 2025-11-21
> **Response to Reviewer B3vF — Question 1**
>
> We sincerely thank the reviewer for raising this question.
> We understand that, as presented in Algorithm 1, it may appear that the proxy network parameters $\phi$ and the codebook $\widetilde{W}$ are updated separately. We appreciate the opportunity to clarify this point more explicitly.
>
> The algorithmic presentation distinguishes the updates of $\phi$ and $\widetilde{W}$ only for explanatory clarity. As described in the revised manuscript (Section 3.2, paragraph "Training algorithm"), both components are in fact optimized **jointly** through the single composite objective
>
> $$
> V_\lambda = R^{\text{fit}} + \lambda R^{\text{align}},
> $$
>
> with the stop-gradient operators simply specifying which subset of parameters receives gradients on each term—similar in spirit to standard training variants used in VQ-VAE [1].
>
> More concretely:
>
> - The loss $R^{\text{fit}}$ contributes gradients only to the codebook parameters.
> - The loss $R^{\text{align}}$ contributes gradients to both the proxy network and the codebook.
> - The "separate" update lines in the pseudocode therefore reflect the different gradient flows, rather than two independent optimization procedures.
>
> Thus, although the updates are written in two blocks for readability, the proxy network and the codebook are optimized **together** within the same training loop. To make this clearer, we added a brief explanatory sentence in the paragraph preceding Algorithm 1, noting that the separation is only notational and that the optimization is performed jointly through a shared loss.
>
> We hope this clarification is helpful and better conveys the intended training structure of CG-CLR.
>
> ### Reference
> [1] Van Den Oord, A., \& Vinyals, O. (2017). *Neural discrete representation learning*. In I. Guyon et al. (Eds.), Advances in Neural Information Processing Systems, 30.

---

> ### Author Response · Authors · 2025-11-21
> **Response to Reviewer B3vF — Weakness 3 (with Questions 2–4)**
>
> We thank the reviewer for pointing out that some parts of the manuscript would benefit from additional clarification. We address each of the reviewer’s questions below.
>
> ## **Question 2: Clarification of the empirical risk**
> We appreciate the reviewer noting that the empirical risk on page 6 was not formally defined in the main text.
> In the revised manuscript, we now provide a clear and self-contained definition of the relevant quantities in *Appendix C.1 ("Formal definition of empirical and expected risks")*.
>
> This section introduces:
>
> - the predictor used by CG-CLR, $h_{\phi,\tilde{W}}(x) = x^{\top} \tilde{w}_{z(\phi(x))}$
> - the **expected risk**,  $\mathcal{R}_{test}$
> - the **empirical risk**, $\mathcal{R}_{train}$
> - and the **composite empirical objective** minimized during training,
>   $ \widehat{\mathcal{R}} = R^{\text{fit}}(\phi,\widetilde{W}^{\text{stop}}) + (1+\lambda)R^{\text{align}}(\phi,\widetilde{W}) $.
>
> These explicit definitions ensure that the empirical risk and training objective appearing in Section 3 are fully formalized and mathematically unambiguous. This clarification also allowed us to refine the presentation of the generalization bound in Theorem 3.3.
>
>
>
> ## **Question 3: Clarification of Assumption 4**
>
> We agree with the reviewer that the original wording of Assumption 4 was ambiguous.
> In particular, the earlier phrasing involving the dimensionality condition $ d \gg K(p+1) $ could be misinterpreted, as it may suggest that the assumption could be satisfied merely by adding dummy parameters.
>
> In the revised manuscript, Assumption 4 has been rewritten to reflect its **intended purpose**, namely to require that the proxy network possesses sufficient expressive capacity to approximate the covariate–to–regressor mapping used in CG-CLR. The updated formulation (now in Section 3.3) no longer refers to the number of parameters and instead states the assumption in terms of a standard expressiveness requirement on the proxy network.
>
> This revision removes the earlier misleading phrasing and clarifies the actual role of Assumption 4 in the analysis.
>
>
> ## **Question 4: Meaning of CG-CLR (Proxy) and CG-CLR (Codebook)**
>
> We appreciate the reviewer noting that these terms were insufficiently defined in the original submission.
>
> To address this, we revised **Section 3.2 ("Test-time prediction")** to explicitly define the two prediction modes:
>
> - **CG-CLR (Codebook):**
>   uses the quantized assignment $ z_i $ and predicts with the selected codebook regressor
>   $ \hat{y}(x_i) = x_i^\top \widetilde{w}_{z_i} $.
>
> - **CG-CLR (Proxy):**
>   directly uses the proxy network’s instance-wise coefficient
>   $ \hat{y}(x_i) = x_i^\top W_\phi(x_i) $.
>
> In the experimental section, these terms are used consistently following the definitions above.
> We hope this resolves the ambiguity that the reviewer encountered.
>
>
> We are grateful for the reviewer’s careful attention to these details, and we believe the corresponding revisions have substantially improved the clarity and precision of the manuscript.

---

### Meta-Review · Area_Chair_JLHJ · 2026-01-06

**Summary:**

The paper proposes a new method to learn a locally linear model. Unlike previous methods, the model can deal directly with unseen covariate vectors during test time and is trainable within a single gradient-based optimization loop. A theoretical convergence analysis, as well as generalization bounds, are provided under standard assumptions. The method is evaluated on various tabular benchmarks where the method shows comparable or improved performance to other locally linear models but worse performance than black box models.

Reviewers gave mixed scores, making this a borderline submission. Many of the initial concerns on experiments, theoretical assumptions, novelty, clarity and issues with the writing seem to be addressed well in the rebuttal. Some remaining concerns that particularly stood out are by reviewer hxki, the (1) partial loss of interpretability of the method, (2) the significantly weaker performance than black-box methods. Overall, the proposed method appears to be positioned as a middle-ground between performance, complexity control (through K) and interpretability. Rebuttal and paper try to argue that local linearity is beneficial, and more illustrative examples on advantages over black-box methods could further improve the paper.

In summary, since most concerns have been addressed by the rebuttal, and as the method forms an interesting addition to the machine learning toolkit, I recommend acceptance of the paper.

**Reviewer Concerns:**

> The method hinges on a black box proxy network and the network alone may work well (B3vF)
- The concern was mainly addressed by appealing to improved interpretability of the local linear regressors.
- However, additional experiments show that the proxy network's predictions largely match the proposed method.

> Experiments are inadequate (B3vF), the proposed method perform worse than baselines (hxki), the proposed method may still be uninterpretable (hxki)
- B3vF concern was addressed by adding the requested experiments to the paper
- hxki concerns remain open, the method seems to be some kind of middle-ground where performance is traded off for interpretability / local linearity, but it remains unclear where this property is really desirable

> Theoretical assumptions are too strong, not clear if they are satisfied during training (QYPe, wePp, hxki)
- These were nicely addressed in the rebuttal.
- However, the role of the theory remains a bit unclear.

> Novelty is limited (wePp), clarity and issues with the writing (wePp)
- Seems to be mostly addressed in rebuttal and blue edits in the paper.

**Reviewer Scores:**

B3vF would have likely raised the score, other reviewers (QYPe, wePp, hxki) maintained.

---

### Decision · Program_Chairs · 2026-01-26

Accept (Poster)